



# Substrate potential of last interglacial to Holocene permafrost organic matter for future microbial greenhouse gas production

**Janina G. Stapel**[1], **Georg Schwamborn**[2], **Lutz Schirrmeister**[2], **Brian Horsfield**[1], and **Kai Mangelsdorf**[1]

[1]GFZ, German Research Centre for Geoscience, Helmholtz Centre Potsdam, Organic Geochemistry,
Telegrafenberg, 14473 Potsdam, Germany
[2]Alfred Wegener Institute, Helmholtz Centre for Polar and Marine Research, Department of Periglacial Research,
Telegrafenberg, A43, 14473 Potsdam, Germany

**Correspondence:** Kai Mangelsdorf (k.mangelsdorf@gfz-potsdam.de)

**Abstract.** In this study the organic matter (OM) in several permafrost cores from Bol'shoy Lyakhovsky Island in NE Siberia was investigated. In the context of the observed global warming the aim was to evaluate the potential of freeze-locked OM from different depositional ages to act as a substrate provider for microbial production of greenhouse gases from thawing permafrost. To assess this potential, the concentrations of free and bound acetate, which form an appropriate substrate for methanogenesis, were determined. The largest free-acetate (in pore water) and bound-acetate (organic-matrix-linked) substrate pools were present in interstadial marine isotope stage (MIS) 3 and stadial MIS 4 Yedoma permafrost deposits. In contrast, deposits from the last interglacial MIS 5e (Eemian) contained only a small pool of substrates. The Holocene (MIS 1) deposits revealed a significant bound-acetate pool, representing a future substrate potential upon release during OM degradation. Additionally, pyrolysis experiments on the OM allocated an increased aliphatic character to the MIS 3 and 4 Late Pleistocene deposits, which might indicate less decomposed and presumably more easily degradable OM. Biomarkers for past microbial communities, including those for methanogenic archaea, also showed the highest abundance during MIS 3 and 4, which indicated OM-stimulated microbial degradation and presumably greenhouse gas production during time of deposition. On a broader perspective, Arctic warming will increase and deepen permafrost thaw and favor substrate availability from older freeze-locked permafrost deposits. Thus, the Yedoma deposits especially showed a high potential for providing substrates relevant for microbial greenhouse gas production.

## 1 Introduction

The northern areas of the Eurasian landmass are underlain by permafrost, which is defined as ground that remains colder than 0 °C for at least 2 consecutive years (Washburn, 1980). These areas contain a large reservoir of organic carbon freeze-locked in the permafrost deposits (French, 2007; Zimov et al., 2009). Hugelius et al. (2014) estimated that about 1300 Pg (1 Pg = $10^{15}$ g = 1 Gt) of soil organic carbon is stored in the upper 0–3 m in the northern circumpolar permafrost region, which is highly vulnerable to climate warming (Grosse et al., 2011; Schmidt et al., 2011; Mu et al., 2014). Today, Arctic summer temperatures are higher than in the past 400 years (Chapin III et al., 2005). Thus, as a consequence of Northern Hemisphere warming an increase in ground temperature, changes in soil drainage, deepening of the active layer (seasonally thawed surface layer), spatial retreat of permafrost, and changes in vegetation have already been reported for the Arctic (Davidson and Janssens, 2006; Anisimov, 2007; Romanovsky et al., 2010; Mueller et al., 2015). During permafrost formation low temperatures, anoxic soil conditions, and low rates of organic matter (OM) decomposition (Schimel and Schaeffer, 2012) result in high rates of OM accumulation (Kuhry et al., 2009; Zimov et al., 2009; Schirrmeister et al., 2011a). The currently observed thawing of permafrost promotes the accessibility of accumulated and freeze-locked OM and nutrients for microbial turnover. This results in increased microbial activity and, consequently, in increased OM decomposition rates (Dutta et al., 2006; Schmidt et al., 2011). Previous studies on permafrost samples from Holocene (marine

isotope stage (MIS) 1) and Late Pleistocene (LP) Yedoma deposits, which represent widespread ice-rich paleosol formations in NE Siberia, have shown that microbial degradability of the freeze-locked OM seems to depend on the amount and composition of organic carbon rather than on the age of the deposits (Knoblauch et al., 2013; Strauss et al., 2015; Stapel et al., 2016). As observed in incubation experiments on permafrost samples of different ages (Waldrop et al., 2010; Lee et al., 2012; Lipson et al., 2012; Knoblauch et al., 2013; Schadel et al., 2014; Walz et al., 2017), degradation of this OM can lead to enhanced microbial production and release of greenhouse gases such as carbon dioxide and methane to the atmosphere (Wagner et al., 2003; Schuur et al., 2008; McGuire et al., 2009; Knoblauch et al., 2013). This release is expected to have strong feedback on global warming and with that on further permafrost degradation (Schuur et al., 2008). Thus, due to the carbon–climate feedback cycle the permafrost region bears the risk of acting as a self-reinforcing system exacerbating global warming.

NE Siberian permafrost formation had already started in the late Pliocene, e.g., at today's coasts and islands along the Dmitry Laptev Strait (Arkhangelov et al., 1996). These deposits provide a unique paleoenvironmental archive with stratigraphic patterns of long-lasting accumulation periods of permafrost during glacial periods, as well as permafrost degradation features during interglacial periods (Andreev et al., 2004, 2009; Wetterich et al., 2009, 2011). Permafrost deposits were accumulated under continental cold climate conditions, accompanied by syngenetic ice-wedge growth (Wetterich et al., 2011) during glacial periods, e.g., the middle Pleistocene (Saalian) and Late Pleistocene (Weichselian; Yedoma deposits) (Andreev et al., 2004; Schirrmeister et al., 2013). In contrast, during the Eemian (MIS 5e) and Holocene interglacial periods, extensive thawing of ice wedges and permafrost deposits led to the formation of thermokarst depressions, as well as of thermo-erosional valleys and small rivers (Andreev et al., 2004; Ilyashuk et al., 2006; Wetterich et al., 2009). According to pollen and insect data, the climate of the Eemian resulted in an open grass and grass–sedge tundra similar to the modern situation (Kienast et al., 2008). The mid-Eemian environment was characterized by 4–5 °C higher summer temperatures than today, with greater seasonal temperature variations in the Northern Hemisphere (Andreev et al., 2004; Dahl-Jensen et al., 2013).

These described environmental changes are expected to have a significant impact, not only on the amount, but also on the composition of the accumulated OM (Strauss et al., 2015; Stapel et al., 2016). After thawing, compositional differences in the formerly freeze-locked OM might strongly affect OM biodegradability and therefore the microbial production of greenhouse gases. To obtain deeper insights into the potential of permafrost OM from different ages to act as a substrate provider for intense microbial degradation, we examined characteristic OM parameters and exemplarily low-molecular-weight organic acid (LMWOA) concen-

trations. LMWOAs such as acetate are important and easily convertible substrates for microbial metabolism (Ganzert et al., 2007). Acetate is a well-known substrate for methanogenesis (Chin and Conrad, 1995). Thus, acetate concentrations provide valuable information on the greenhouse gas production potential of the respective OM (Stapel et al., 2016). On the one hand, acetate can be dissolved in pore water and cryostructures (e.g., segregated ice) of permafrost deposits. This acetate represents a free-substrate pool, which is directly bioavailable for microorganisms. On the other hand, acetate can be bound to the organic matrix (e.g., by ester linkage) forming a future substrate pool upon liberation via geochemical or microbial alteration of the OM (Glombitza et al., 2009a, b; Stapel et al., 2016).

In addition, microbial biomarkers such as phospholipid fatty acids (PLFAs) and glycerol dialkyl glycerol tetraethers (GDGTs) were analyzed to examine the interaction of present and past microbial communities with the OM accumulated in the past. Phospholipid esters are essential membrane components of living bacterial cells (Zelles, 1999) and are hydrolyzed rapidly after cell death (White et al., 1979; Logemann et al., 2011). Therefore, their fatty acid side chain inventories (PLFAs) are used as an indicator for viable microorganisms in sediments (Haack et al., 1994; Bischoff et al., 2013). In contrast, biomarkers such as GDGTs and archaeol represent membrane lipids of dead microbial biomass since they are already partly degraded as indicated by the loss of their head groups (Pease et al., 1998). While archaeol and GDGTs with isoprenoid tetraether bridges (isoGDGTs) represent archaeal biomass, GDGTs with branched tetraether bridges (brGDGTs) derive from bacteria (Weijers et al., 2006a, b). However, it should be mentioned that the brGDGT biomarkers only represent part of the bacterial community (Weijers et al., 2006a, b; Schouten et al., 2013), while the archaeal markers cover most of the past archaeal community (Pancost et al., 2001; Koga and Morii, 2006). In permafrost regions and peatlands archaeol is used as a biomarker for methanogenic archaea (Bischoff et al., 2013; Pancost et al., 2011).

The feedback between climate warming and microbial greenhouse gas generation from thawing permafrost is a topic of large global interest and intensive scientific debate (Zimov et al., 2006; Koven et al., 2011; Schuur et al., 2015). In this context the contribution from thawing OM of different depositional ages to the carbon–climate feedback cycle is still unclear. In order to learn more about this interrelation, we conducted a study on Bol'shoy Lyakhovsky Island in the Laptev Sea (NE Siberia). Samples from this region provided an excellent opportunity to investigate permafrost OM that was deposited from the last interglacial to the Holocene. The aims of our study were (1) to assess and compare the stored OM potential for microbial greenhouse gas production in permafrost deposits from different glacial and interglacial periods and (2) to assign this substrate potential to charac-

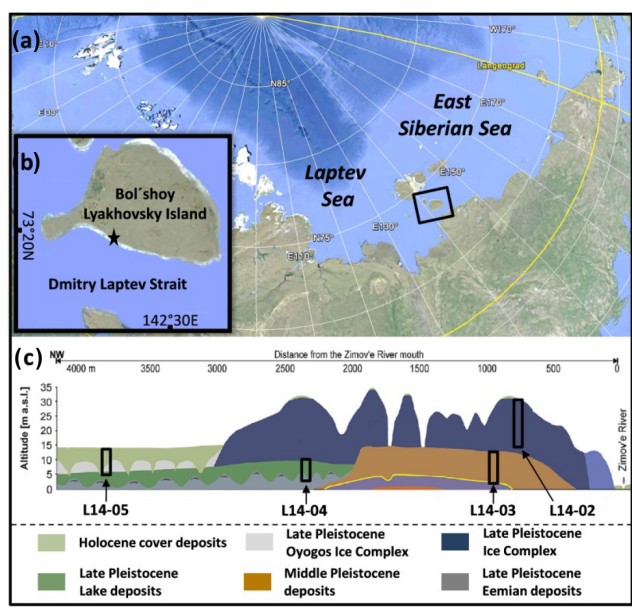

**Figure 1. (a)** Position of Bol'shoy Lyakhovsky Island in the Siberian Arctic. **(b)** Study site on Bol'shoy Lyakhovsky Island indicated by a black star and **(c)** location of the drilled cores comprising different age intervals (L14-05, L14-02, L14-03, and L14-04) modified after Wetterich et al. (2009, 2014) and Schwamborn and Wetterich (2015).

teristic OM parameters and depositional paleoenvironmental conditions.

## 2 Study area and material

Bol'shoy Lyakhovsky Island is located between the Laptev and East Siberian seas as the southernmost part of the New Siberian Archipelago (Fig. 1a). During Pleistocene periods of low sea level the island was part of west Beringia, an unglaciated landmass stretching from NE Siberia to Alaska (Hubberten et al., 2004; Andreev et al., 2009). The area is part of the northern tundra zone with an active layer (AL) thickness of 30–40 cm and a permafrost thickness of 500–600 m (Andreev et al., 2004). The study site is located west of the Zimové River mouth on the south coast of Bol'shoy Lyakhovsky Island along the Dmitry Laptev Strait (Fig. 1b). The southern coast is characterized by exposed permafrost deposits, while the hinterland is formed by gradually sloping terrain intersected by rivers and valleys developed through thermo-erosion. Based on previous studies, the stratigraphy and regional setting are well known in the current study area (Andreev et al., 2004, 2009; Ilyashuk et al., 2006; Kienast et al., 2008; Wetterich et al., 2009, 2014). Therefore, the drill sites (Fig. 1c) were chosen to maximize stratigraphic coverage and age with the aim to obtain a permafrost record from the Holocene (MIS 1) back to the Eemian interglacial (MIS 5e; Russian: Kazansevo). Eemian deposits

form a paleo-equivalent to the Holocene and are otherwise rather difficult to assess in this area. According to Wetterich et al. (2014) and studies referenced therein, the cores investigated in this study can be integrated into an already described environmental and climatic history. It must be mentioned that similar interglacial deposits at Bereg Oyogos-Yar CE1 from the mainland coast opposite to Bol'shoy Lyakhovsky Island have recently been dated. These deposits reveal infrared optical stimulated luminescence (IR-OSL, $112.5 \pm 9.6$ kyr) ages that are slightly younger than Eemian (Opel et al., 2017). However, since it is not yet clear whether both deposits really represent the same age window, we follow the interpretation by Andreev et al. (2004) based on the Bol'shoy Lyakhovsky deposits.

The field work was conducted in April 2014 as part of the joint Russian–German research project CarboPerm (Schwamborn and Wetterich, 2015). Four cores were drilled using a KMB-3-15M (rotary) drill rig. The drilled core segments were kept frozen and transported in the frozen state for further processing to Potsdam, Germany. In our home laboratory sampling was conducted in a climate chamber at $-10\,°C$. With the exception of intervals in which ice wedges were encountered, 40 inner core samples distributed throughout the cores were taken. Samples were investigated for microbial biomarkers, free-acetate (pore-water) and bound-acetate concentrations, and OM characteristics such as total organic carbon (TOC), TOC-to-total nitrogen (TOC / TN) ratio CE2 hydrogen index (HI), and compositional OM analysis using open-pyrolysis gas chromatography (Py-GC).

## Core descriptions

Cores are described stratigraphically from younger to older deposits. Core L14-05 (Fig. 1c) is 7.89 m long and consists of silty finely grained sediments with scattered organic remains. Overall this core possesses lens-like cryostructures which occur between 1.00 and 2.45 m as well as between 6.71 and 7.89 m of CE3 core depth. According to prior studies by Andreev et al. (2009) and Wetterich et al. (2009), the upper core section approximately down to 5.5 m consists of a Holocene (MIS 1) unit (Table 1), while the deeper core section is of MIS 3 age (Russian: Kargin). According to previous paleoenvironmental interpretations the MIS 1 deposits represent Alas deposits, where early Holocene lake sediments have accumulated on top of a MIS 3 surface. During the late Holocene ($< 3.7$ ka BP) the site drained and froze (Andreev et al., 2009).

Core L14-02 (Fig. 1c) is 20.02 m in length. The upper 11.26 m consists of CE4 silty finely grained sediments with macroscopic organic remains and an alternation of horizontal, vertical, and reticulated ice veins and lens-like cryostructures. Below 11.26 m of depth the core consists of an ice wedge, and no samples were taken from this part. The core material is of LP age (Table 1) and was deposited under subaerial conditions during the last interstadial MIS 3 (Wetterich

**Table 1.** Schematic overview of assigned ages of the core material drilled on Bol'shoy Lyakhovsky Island showing the different cores with their drilling positions in relation to different age periods (interval names, marine isotope stages (MIS) and Russian terminology (RT) for the respective time periods). The age assignment is based on information obtained from Andreev et al. (2004, 2009) and Wetterich et al. (2009, 2014).

| Cores | Drill site | Age | | |
|-------|-----------|-----|-----|-----|
| | | **Period** | **MIS** | **RT** |
| L14-05 | 73.34994°N 141.24156°E | Holocene (interglacial) | 1 | Holocene |
| L14-02 | 73.33623°N 141.32761°E | Late Pleistocene glacial period (glacial) — interstadial | 3 | Kargin |
| L14-03 | 73.33464°N 141.32822°E | Late Pleistocene glacial period (glacial) — stadial | 4 | Zyryan |
| L14-04 | 73.34100°N 141.28587°E | Eemian (interglacial) | 5e | Kazansevo |

et al., 2014). The deposits represent the infill of an ice-wedge polygon with a succession of paleosols.

The upper 4.90 m of core L14-03 (15.49 m in length; Fig. 1c) is comparable in its sedimentology and cryostructures to the silty finely grained sediments of core L14-02. Below 4.90 m of depth the sediment has more sandy portions. The sediments between 4.90 and 8.45 m have visible organic remains and similar cryostructures. Below 8.45 m of depth the sediments are characterized by only scattered organic remains but cryostructures similar to those described above. Below 10.90 m of depth the deposits mainly consist of sand and gravel; the lowermost 40 cm is gravel. Cryostructures are partly formed as vertically aligned centimeter-thick ice veins. In earlier studies at the same site these deposits were interpreted to represent MIS 4 (Russian: Zyryan; Table 1) deposits (Andreev et al., 2009).

Core L14-04 (Fig. 1c) is 8.10 m long and consists of silty finely grained sediments with visible organic remains and cryostructures, comparable to core L14-02. Between 4.24

and 4.89 m the core consists of massive ice. The upper 6 m was probably deposited during the MIS 4 stadial period (Table 1). Sediments below 6 m of depth were deposited during the Eemian (MIS 5e; Russian: Kazansevo; Table 1) and appear to represent thermokarst lake sediments (Andreev et al., 2004).

## 3 Methods

### 3.1 OM parameters

After freeze-drying and grinding, samples for TOC analysis were decalcified with 0.1 N HCL. TOC and TN (wt %) were determined by a carbon–nitrogen–sulfur elemental analyzer (Vario EL III, Elementar) with a device-specific accuracy of ±0.1 wt %. For further information on characteristic OM parameters the HI was determined with Rock-Eval pyrolysis using a Rock-Eval 6 instrument (Behar et al., 2001). Therefore, 17 freeze-dried and ground samples of different TOC content covering all time intervals were analyzed. Measurements were conducted by Applied Petroleum Technology AS (Kjeller, Norway). To obtain additional information on the macromolecular structure of the OM, 10 mg from these samples was used for open-system pyrolysis after Horsfield et al. (1989). After the free biomolecules (bitumen) were thermally removed (300 °C), the macromolecular organic matrix was pyrolyzed with temperatures between 300 and 600 °C. The pyrolysates were trapped (liquid $N_2$) and finally measured on a pyrolysis gas chromatograph (Agilent GC 6890A chromatograph) equipped with a flame ionization detector (Py-GC-FID). For peak quantification of the detected pyrolysate products, $n$-butane was used as the external standard. The areas of the detected pyrolysate peaks were integrated and calculated using the Agilent ChemStation software. For the Eglinton diagram (Eglinton et al., 1990) we integrated o-xylene, 2,3-dimethylthiophene, and $n$-nonene; for the Horsfield diagram (Horsfield et al., 1989) $C_1$-$C_5$ $n$-alkanes, $C_6$-$C_{14}$-$n$-alkanes, and $n$-alkenes as well as $C_{15}$ and longer $n$-alkanes and $n$-alkenes were integrated. For further details on these methods see Horsfield et al. (1989) and Stapel et al. (2016).

### 3.2 Low-molecular-weight organic acid analyses

After slow thawing of a subset of the frozen samples at about 4 °C, the pore water within the samples was separated by centrifugation (Sigma, laboratory centrifuge 6K15, 1398 × $g$, 20 °C, 10 min). Free LMWOAs such as acetate were detected in pore water samples using ion chromatography with conductivity detection (ICS 3000, Dionex) and measured as milligrams per liter of acetate. Furthermore, LMWOAs bound via ester bonds to the complex OM were analyzed by conducting an alkaline ester cleavage approach developed by Glombitza et al. (2009a) on pre-extracted sediment samples. Details are described in Stapel et al. (2016).

### 3.3 Microbial lipid biomarker analysis

Approximately 30–50 g of the freeze-dried and ground samples was extracted using a flow blending system modified after Bligh and Dyer (1959) as described in Stapel et al. (2016). Subsequently, the obtained sediment extract was separated into four different fractions of increasing polarity (low polar lipids, free fatty acids, glycolipids, and polar lipids) following a method described by Zink and Mangelsdorf (2004). Finally, all four fractions were evaporated to dryness and stored at $-20\,^\circ\text{C}$ until analysis. After a fatty acid cleavage procedure described in Müller et al. (1998), the PLFAs within the polar-lipid fraction were measured using gas chromatography mass spectrometry (GC-MS). For PLFA quantification an internal standard (1-myristoyl-d27-sn-glycero-3-phosphocholine) was used. PLFAs were identified by means of mass spectrum interpretation and comparison with a standard mixture comprising a series of saturated, unsaturated, branched, and cyclo-propyl-fatty acid methyl esters in the range from $C_{12}$ to $C_{20}$. Details on instrument settings are described in Stapel et al. (2016).

After asphaltene precipitation the low polar lipid fraction was separated into aliphatic, aromatic, and hetero-compound (containing nitrogen, sulfur, and oxygen components; NSO) fractions using a medium-pressure liquid chromatography system (Radke et al., 1980). An aliquot of the NSO fraction was investigated for tetraether lipids (GDGTs) and archaeol using a Shimadzu LC20AD HPLC instrument coupled to a Finnigan TSQ 7000 triple quadrupole MS instrument with an atmospheric pressure chemical ionization interface. An external archaeol standard was used for quantification. Details on instrument settings are described in Stapel et al. (2016). The branched vs. isoprenoid tetraether (BIT) index was calculated after Hopmans et al. (2004). The data on individual GDGTs are provided in the Supplement (Tables S1 and S2).

### 3.4 Statistical approaches

For statistical analysis of the measured parameters, the Pearson correlation coefficient ($R^2$) was computed using the MATLAB R2015b software environment. In addition, $p$ values were also calculated with the same software and only correlations with $p \leq 0.05$ were evaluated for this study.

### 4 Results

Characteristic OM parameters (TOC, TOC / TN, and HI), biomarkers for living microbial communities (PLFA) and for past bacterial (brGDGTs) as well as archaeal communities (isoGDGT-0 and archaeol), and the concentration of free and bound acetate are presented in Fig. 2 for all four cores from Bol'shoy Lyakhovsky Island. Every core includes at least one sample (core L14-03 has two samples) representing the overlaying soil as part of the AL above the permafrost deposits from MIS 1, 3, and 4. Due to the stratigraphic settings at

the study site on Bol'shoy Lyakhovsky Island, ALs containing OM from MIS 2 and MIS 5e could not be obtained in the field. Additionally, the results from open-system pyrolysis are shown in Fig. 3.

### 4.1 Characteristic OM parameters

**Active layers**

In the ALs TOC concentrations were above 2 wt %, except in core L14-05 with 1.5 wt % (Fig. 2a). The TOC / TN ratios ranged between 8.2 and 11.1 TS1 (Fig. 2b). Rock-Eval analysis of a representative AL sample (L15-05, 10 cm) revealed a HI value of 236 mg hydrocarbons (HC) $\text{g}^{-1}$ of TOC (Fig. 2c). Overall, the AL samples showed significantly higher PLFA concentrations (84.2, 149.3, 86.3, and 37.9 $\mu\text{g}\,\text{g}^{-1}$ of sediment) compared to the underlying permafrost deposits (Fig. 2d). The PLFA inventory was composed of saturated FAs ranging from $C_{12}$ to $C_{24}$; *iso*-branched FAs from $C_{13}$ to $C_{19}$; *anteiso*-branched FAs from $C_{13}$ to $C_{17}$; other saturated branched FAs from $C_{17}$ to $C_{20}$; a saturated mid-chain branched FA 10-Me16 : 0; a series of monounsaturated FAs such as 14 : 1ω5, 15 : 1, 16 : 1ω5*cis/trans*, 16 : 1ω7*cis/trans*, 17 : 1ω7*cis/trans*, 18 : 1ω7*cis/trans,* and 18 : 1ω9*cis/trans*; FAs with a cyclopropyl ring, cycl-17 : 0 and cycl-19 : 0; and some monounsaturated branched FAs from $C_{15}$ to $C_{17}$. The sum of all brGDGT (past bacterial markers) varied between 27.1 and 834.1 $\text{ng}\,\text{g}^{-1}$ of sediment, while concentrations of the isoGDGT-0 with 0.8 to 3.1 $\text{ng}\,\text{g}^{-1}$ of sediment and archaeol (both past archaeal markers) with 2.0 to 17.7 $\text{ng}\,\text{g}^{-1}$ of sediment were much lower in each core (Fig. 2e to g). The free-acetate concentrations (Fig. 2h) were rather low (0.9 to 16.7 $\text{mg}\,\text{L}^{-1}$ of acetate) compared to the rest of the cores. In contrast, the bound-acetate concentrations were comparatively high at 44.5 to 64.2 $\text{mg}\,\text{L}^{-1}$ (Fig. 2i).

**Marine Isotope Stage 1 (MIS 1, Holocene interglacial)**

In the MIS 1 permafrost section of core L14-05, the TOC, TOC / TN, and HI profiles (Fig. 2a to c) all correlated (TOC : TOC / TN, $R^2 = 0.9$; $p = 0.047$; TOC : HI, $R^2 = 0.79$; $p = 0.015$). TOC concentrations varied between 0.2 and 1.8 wt %, TOC / TN ratios between 2.4 and 8.2, and HI data between 71 and 194 mg HC $\text{g}^{-1}$ of TOC (H1 to H5; Fig. 2c). Overall, the concentration of PLFAs was lower than in the AL, ranging between 11.7 and 27.0 $\mu\text{g}\,\text{g}^{-1}$ of sediment (Fig. 2d). Also, the diversity of detected PLFAs decreased with depth. The PLFA profile revealed some similarities to the TOC data, but no overall correlation was found. Concentrations of past bacterial markers varied between 22.5 and 370.0 $\text{ng}\,\text{g}^{-1}$ of sediment (Fig. 2e). Past archaeal markers ranged between 1.0 and 15.0 $\text{ng}\,\text{g}^{-1}$ of sediment for isoGDGT-0 and between 5.4 and 41.1 $\text{ng}\,\text{g}^{-1}$ of sediment for archaeol (Fig. 2f to g). The brGDGT profile correlated well with TOC ($R^2 = 0.9$; $p = 0.015$). Free-acetate concen-

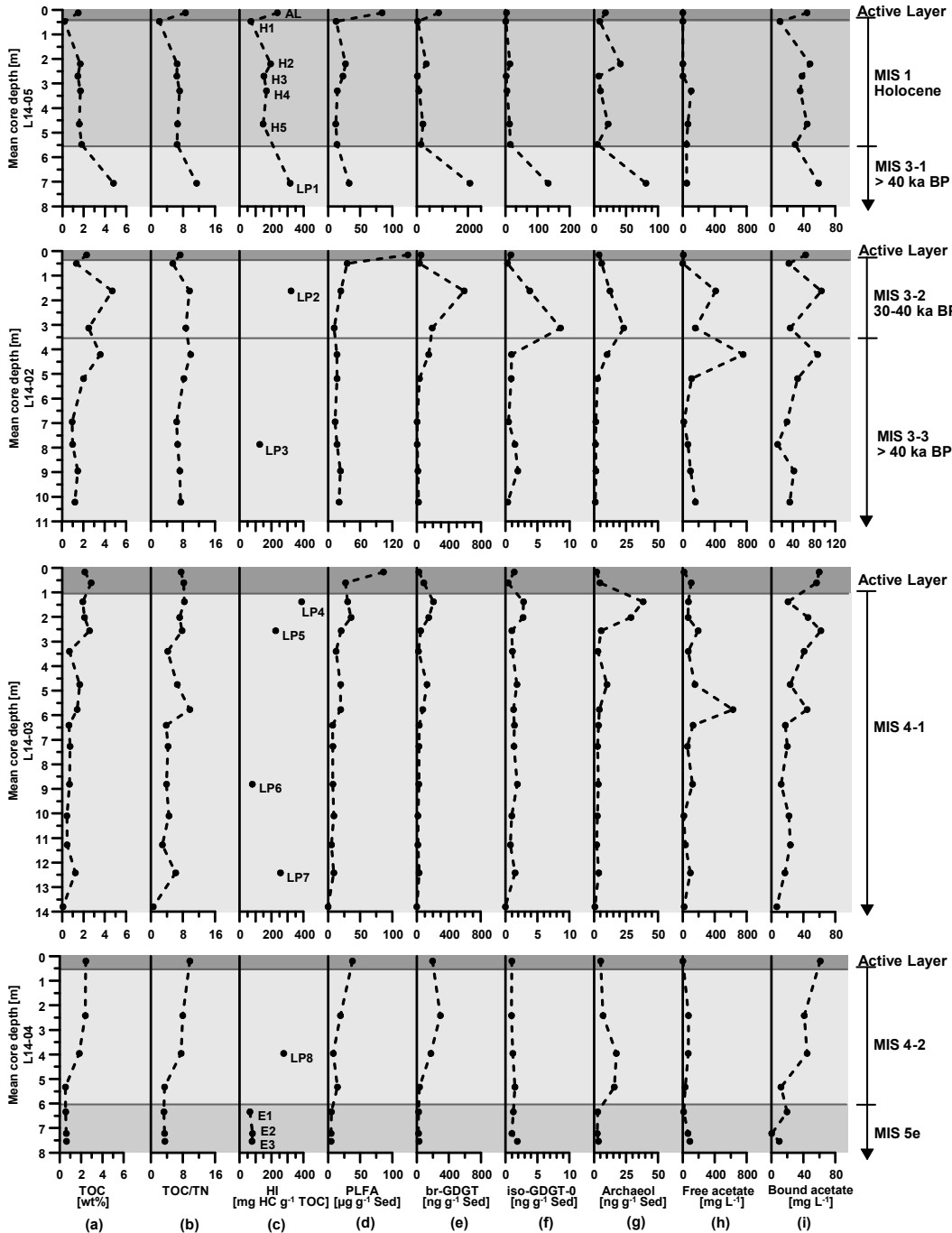

**Figure 2.** Bio- and geochemical parameters of permafrost cores L14-05, L14-02, L14-03, and L14-04 from Bol'shoy Lyakhovsky Island, northern Siberia, presented with respect to core depth (left axis) as well as stratigraphic and age units (right column). The vertical profiles show (note partly different $x$ axes) **(a)** the total organic carbon (TOC) content in wt %, **(b)** the ratio of TOC and total nitrogen (TN), **(c)** the hydrogen index (HI) in mg HC g$^{-1}$ of TOC, **(d)** the concentration of phospholipid fatty acids (PLFAs) in µg g$^{-1}$ of sediment, **(e)** the concentration of branched glycerol dialkyl glycerol tetraethers (brGDGTs), **(f)** of isoGDGT-0, and **(g)** of archaeol, all in ng g$^{-1}$ of sediment, and **(h)** the concentration of free acetate and **(i)** bound acetate, both in mg L$^{-1}$. Active layer samples are shown in dark grey, interglacial periods (MIS 1 and MIS 5e) in grey, and the last glacial period (MIS 3 and MIS 4) in light grey. According to age, stratigraphy, and core segments, the MIS 3 unit is subdivided into core segments MIS 3-1, 3-2, and 3-3 and the MIS 4 unit into segments MIS 4-1 and 4-2. Sample labels within the HI profile correspond to core samples of different ages (H: Holocene; LP: Late Pleistocene glacial period; E: Eemian).

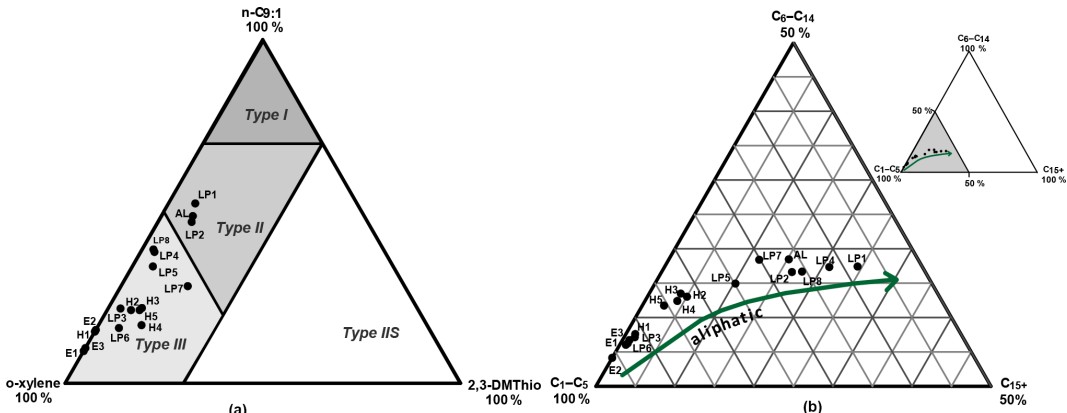

**Figure 3.** Triangular plots derived from organic matter (OM) open-system pyrolysis. **(a)** Eglinton diagram showing the classification of the kerogen type (type I/II: aquatic and marine; type III: terrestrial; type IIS: enriched sulfur content) due to the relative abundance of 1,2-dimethylbenzene (ortho-xylene), $n$-nonene ($n$-C9 : 1), and 2,3-dimethylthiophene (2,3DMThio) in the OM after Eglinton et al. (1990). **(b)** Horsfield diagram showing the composition of the OM according to the chain length distribution of short- ($C_1$-$C_5$), intermediate- ($C_6$-$C_{14}$), and long-chain ($C_{15+}$) $n$-alkanes and $n$-alk-1-enes after Horsfield et al. (1989). The arrow indicates an increasing aliphatic proportion in the OM of the investigated samples. Sample labels correspond to core samples of different ages (H: Holocene; LP: Late Pleistocene glacial period; E: Eemian) with different total organic carbon (TOC) and hydrogen index (HI) values (Fig. 2a and c).

trations (Fig. 2h) of up to $2.9\,\mathrm{mg\,L^{-1}}$ were detected between 0.5 and 2.7 m, followed by an increase to $106\,\mathrm{mg\,L^{-1}}$ at 3.3 m of depth. Bound-acetate concentrations (Fig. 2i) correlated with TOC ($R^2 = 0.8$; $p = 0.046$) and were comparatively low ($10.7\,\mathrm{mg\,L^{-1}}$) at 0.5 m of depth, but were much higher, with concentrations between 29.8 and $48.2\,\mathrm{mg\,L^{-1}}$, for the rest of the section.

**Marine Isotope Stage 3 (MIS 3, last glacial interstadial)**

MIS 3 comprised the core sections MIS 3-1 from core L14-05 as well as MIS 3-2 and MIS 3-3 from core L14-02 (Table 1, Fig. 2). Overall, the TOC contents and TOC / TN ratios (Fig. 2a, b) correlated ($R^2 = 0.8$; $p = 0.003$). The sample from core section MIS 3-1 showed the highest TOC content (4.8 wt %) and TOC / TN ratio (12.8). Samples from core section MIS 3-2 were characterized by TOC concentrations ranging from 1.4 to 4.7 wt % with a maximum at 1.6 m of depth and TOC / TN ratios between 6.2 and 11.8. Within core section MIS 3-3, TOC varied from 1.0 to 3.6 wt % and TOC / TN ratios from 8.1 to 11.9. HI values of 316 (LP1), 322 (LP2), and 126 mg HC g$^{-1}$ of TOC (LP3) were detected for the MIS 3 interval (Fig. 2c). Figure 2d shows PLFA concentrations of $32.7\,\mathrm{\mu g\,g^{-1}}$ of sediment for core section MIS 3-1, a decreasing trend from 35.6 to $11.2\,\mathrm{\mu g\,g^{-1}}$ of sediment in core section MIS 3-2, and concentrations between 13.0 and $23.0\,\mathrm{\mu g\,g^{-1}}$ of sediment in core section MIS 3-3. In the MIS 3 interval no correlation between PLFA concentration and TOC was observable. In contrast, past bacterial markers (Fig. 2e) generally correlated with TOC ($R^2 = 0.9$; $p = 0.016$). Past bacterial and archaeal biomarkers (Fig. 2e to g) showed higher abundances in the same depth interval (0.5 to 4.2 m), but reached their maximum at different

depths ($R^2 = 0.8$; $p = 0.021$). In the MIS 3-1 section all past markers were significantly higher ($2070.4\,\mathrm{ng\,g^{-1}}$ of sediment for brGDGTs, $133.4\,\mathrm{ng\,g^{-1}}$ of sediment for isoGDGT-0, and $81.1\,\mathrm{ng\,g^{-1}}$ of sediment for archaeol) compared to the other MIS 3 deposits. MIS 3-2 was characterized by concentrations between 34.3 and $591.2\,\mathrm{ng\,g^{-1}}$ of sediment for the brGDGTs, 0.3 and $8.6\,\mathrm{ng\,g^{-1}}$ of sediment for isoGDGT-0, and 5.9 and $23.2\,\mathrm{ng\,g^{-1}}$ of sediment for archaeol. Much lower concentrations were observed for MIS 3-3 with values between 3.7 and $147.7\,\mathrm{ng\,g^{-1}}$ of sediment for the brGDGTs, 0.3 and $1.9\,\mathrm{ng\,g^{-1}}$ of sediment for isoGDGT-0, and 1.0 and $10.2\,\mathrm{ng\,g^{-1}}$ of sediment for archaeol. The sample from MIS 3-1 showed a free-acetate concentration (Fig. 2h) of $51.3\,\mathrm{mg\,L^{-1}}$. In MIS 3-2 the free-acetate concentration rose to $412.0\,\mathrm{mg\,L^{-1}}$ at 1.6 m of depth. In MIS 3-3 the highest concentration of up to $755.5\,\mathrm{mg\,L^{-1}}$ was measured at 4.2 m of depth, followed by decreasing concentrations to $13.8\,\mathrm{mg\,L^{-1}}$, and a subsequent rise to $160.0\,\mathrm{mg\,L^{-1}}$. The bound-acetate concentrations (Fig. 2i) correlated with TOC ($R^2 = 0.8$; $p = 0.049$) and showed a value of $58.9\,\mathrm{mg\,L^{-1}}$ in MIS 3-1. In the MIS 3-2 and MIS 3-3 core sections bound-acetate concentrations ranged between 11.7 and $93.7\,\mathrm{mg\,L^{-1}}$ with maxima at 1.6 and 4.2 m of depth.

**Marine Isotope Stage 4 (MIS 4, last glacial stadial)**

MIS 4 comprised the core sections MIS 4-1 from core L14-03 and MIS 4-2 from core L14-04. The TOC contents and TOC / TN ratios (Fig. 2a, b) correlated within the MIS 4 interval ($R^2 = 0.8$; $p = 0.009$). In the upper 2.6 m of core section MIS 4-1, TOC concentrations ranged between 1.9 and 2.6 wt % and TOC / TN ratios between 8.6 and 9.7. Below 2.6 m TOC concentrations were $\leq 1.7$ wt % and TOC / TN

ratios usually $< 6.1$. Exceptions occurred at 4.8 and 5.8 m of depth with values of 7.8 and 10.5, respectively. Core section MIS 4-2 was characterized by a decreasing TOC trend from 2.4 to 0.5 wt % and TOC / TN ratios from 9 to 4.6. Four HI values were measured in MIS 4-1 with values of 388 (LP4), 226 (LP5), 80 (LP6), and 256 (LP7) mg HC g$^{-1}$ of TOC. In the MIS 4-2 section one HI value was determined with a value of 276 (LP8) mg HC g$^{-1}$ of TOC (Fig. 2c). In the upper 5.7 m of MIS 4-1 the PLFA concentrations (Fig. 2d) resembled the TOC contents ($R^2 = 0.7$; $p = 0.002$) with concentrations between 12.4 and 35.8 µg g$^{-1}$ of sediment and lower concentrations, between 5.3 and 9.0 µg g$^{-1}$ of sediment, below 6 m. Core section MIS 4-2 revealed low PLFA concentrations between 8.0 and 19.2 µg g$^{-1}$ of sediment. In MIS 4-1 past microbial biomarker profiles (Fig. 2e to g) correlated with each other ($R^2 = 0.8$; $p = 0.031$) and with TOC (brGDGTs vs. TOC: $R^2 = 0.7$, $p = 0.047$; archaeol vs. TOC: $R^2 = 0.7$, $p = 0.039$). The concentrations of bacterial markers generally ranged from 13.7 to 48.9 ng g$^{-1}$ of sediment with an increase to 208.0 ng g$^{-1}$ of sediment at 1.4 m and to 128.0 ng g$^{-1}$ of sediment at 4.7 m of depth. For the archaeal markers, concentrations varied between 0.8 and 2.8 ng g$^{-1}$ of sediment for isoGDGT-0 as well as between 2.0 and 38.3 ng g$^{-1}$ of sediment for archaeol with maxima at 1.4 and 4.8 m of depth. In the MIS 4-2 interval the bacterial GDGT concentrations decreased from 294.0 to 34.2 ng g$^{-1}$ of sediment and correlated with TOC ($R^2 = 0.9$; $p = 0.018$). The archaeal marker concentrations ranged from 1.0 to 1.5 ng g$^{-1}$ of sediment for isoGDGT-0 and between 7.0 and 17.4 ng g$^{-1}$ of sediment for archaeol with a maximum at 4 m of depth. The archaeal marker concentrations did not correlate with the profile of brGDGT markers or with TOC. Within core section MIS 4-1 free-acetate concentrations (Fig. 2h) were generally below 150.1 mg L$^{-1}$. However, maxima occurred at 2.6 m (193.2 mg L$^{-1}$) and 5.8 m of depth (628.5 mg L$^{-1}$). In core section MIS 4-2 the free-acetate concentrations ranged from 31.0 to 72.0 mg L$^{-1}$. The bound-acetate concentrations (Fig. 2i) of the MIS 4 interval correlated with TOC ($R^2 = 0.7$; $p = 0.018$). The concentrations varied between 6.7 and 61.8 mg L$^{-1}$ with maxima at 2.6 and 5.8 m in the MIS 4-1 section and decreasing concentrations from 41.2 to 12.1 mg L$^{-1}$ in the MIS 4-2 section.

**Marine Isotope Stage 5e (MIS 5e, Eemian interglacial)**

In the MIS 5e (Eemian) interval from core L14-04, TOC contents and TOC / TN ratios correlated ($R^2 = 0.99$; $p = 0.018$) and showed low TOC contents of about 0.6 wt % and TOC / TN ratios from 3.7 to 5.0 (Fig. 2a, b). Samples from MIS 5e showed low HI values (Fig. 2c) of 65 (E1), 81 (E2), and 78 (E3) mg HC g$^{-1}$ of TOC. The PLFA concentrations (Fig. 2d) were between 4.6 and 5.1 µg g$^{-1}$ of sediment, which is quite low compared to the other intervals and resembled the low TOC profile. All past microbial marker profiles (Fig. 2e to g) were low. The concentrations of past bacterial markers varied between 21.8 and 27.1 ng g$^{-1}$ of sediment, while the concentrations of the archaeal markers ranged from 1.0 to 1.9 ng g$^{-1}$ of sediment for isoGDGT-0 and from 2.5 to 3.4 ng g$^{-1}$ of sediment for archaeol. The free-acetate concentrations (Fig. 2h) increased from 12.5 to 89.5 mg L$^{-1}$ with depth, while the bound-acetate concentrations (Fig. 2i) scattered between 0.5 and 19.6 mg L$^{-1}$.

## 4.2 Open-system pyrolysis GC

Results provided by open-system pyrolysis experiments on 17 representative samples (high and low TOC) enabled a deeper insight into the OM characteristics. Figure 3a (after Eglinton et al., 1990) classifies the deposited OM into aliphatic-, aromatic-, or sulfur-rich OM. All samples from the Holocene (H1, H2, H3, H4, H5) and Eemian (E1, E2, E3) units and two samples from the LP unit (LP3, LP6) fell within the range of OM type III (terrestrial OM type). LP samples (LP4, LP5, LP7, LP8) corresponding to higher HI values showed a mixture of OM types III and II (increased aliphatic character). Two LP samples (LP1, LP2) and the AL sample, all displaying the highest HI values, fell within the range of OM type II. All samples, especially the samples from the Eemian (MIS 5e), showed only a very low abundance of sulfur compounds generated by pyrolysis, indicating sulfur-poor OM (2,3-dimethylthiophene).

Figure 3b (after Horsfield et al., 1989) suggests different aliphatic characters for the selected samples, indicating an increasing aliphatic character with higher HI and TOC. Samples from the Eemian unit (E1, E2, E3) and the Holocene sample (H1) as well as two samples from the LP unit (LP3, LP6) all with low HI ($< 130$ mg HC g$^{-1}$ of TOC) and low TOC ($< 1$ wt %) revealed the weakest aliphatic character. In comparison, the samples from the Holocene unit (H2, H3, H4, H5) with intermediate HI values (140–200 mg HC g$^{-1}$ of TOC) and TOC contents (up to 1.7 wt %) showed a slightly increased aliphatic character. All these samples were characterized by OM type III (Fig. 3a). Most of the LP samples (LP1, LP2, PL4, PL5, PL7, PL8) and the AL sample revealed the strongest aliphatic character corresponding to higher HI values $> 200$ mg HC g$^{-1}$ of TOC and TOC contents of up to 4.8 wt %. These samples resembled a mixture of OM types III and II (Fig. 3a, b).

## 5 Discussion

When permafrost thaws, formerly freeze-locked OM becomes bioavailable again (Wagner et al., 2007; Lee et al., 2012). In order to assess the impact of this OM on future climate evolution it is of the utmost interest to learn more about the degradability of the thawing OM, especially with regard to its potential to release greenhouse gases. OM degradability in soils can be influenced by the molecular structure of the source OM and the decomposition processes that affected

this material during deposition. Thereby, decomposition is affected by environmental factors (temperature, water saturation causing oxic or anoxic conditions, and adsorption onto the mineral soil matrix) and biological controls concerning the microbial ecosystem involved (Schmidt et al., 2011). In this paper the focus is placed on the OM composition and its specific characteristics. However, environmental and biological controls on OM degradation and accumulation are also considered.

## 5.1 OM characteristics

The composition of permafrost OM is mainly a mix of different terrestrial sources and the result of early diagenetic degradation processes during OM deposition in the past (White, 2013). We applied pyrolysis techniques (Rock-Eval pyrolysis and open-system pyrolysis GC-FID) to the OM to gain a deeper insight into the structural composition and to define specific characteristics for OM of different depositional ages.

The highest accumulation of OM, up to 4.9 % TOC, was found in the interstadial deposits of MIS 3 (core sections 3-1 and 3-2), which is in the range also observed in other studies investigating Yedoma deposits (Schirrmeister et al., 2011a; Strauss et al., 2015). The measured TOC / TN ratios of 5 to 12 are within the range of terrestrial permafrost deposits reported for the NE Siberian Arctic (Wetterich et al., 2009; Schirrmeister et al., 2011a; Strauss et al., 2015). Open-system pyrolysis data confirm, as expected, that the permafrost deposits on Bol'shoy Lyakhovsky Island are mainly dominated by terrestrial OM (type III OM) (Fig. 3a), although Holocene (MIS 1) and Eemian (MIS 5e) samples are deposited in a thermokarst environment (Wetterich et al., 2009). Pyrolysis experiments using TOC-rich samples from the AL (core L14-05) and the LP glacial period (comprising MIS 3 and MIS 4, Table 1) revealed high HI values (Fig. 2c). The Eglinton triangular plot indicates for these samples a type III OM with a tendency to type II OM (Fig. 3a). This is mainly caused by the higher aliphatic character of these samples as shown by the Horsfield triangular plot (Fig. 3b). Thus, this TOC-rich OM can be interpreted as terrestrial OM with higher proportions of aliphatic structural moieties. One origin of this aliphatic material could be soil algae OM. Algae are usually rich in aliphatic structural units (Kolattukudy, 1980) and cyanobacteria and green algae are reported for permafrost soils and deposits (Vishnivetskaya, 2009). Thus, these samples might indicate OM accumulation during intervals of increased soil moisture, which would be favorable for algae growth. Periods of increased soil moisture have been reported by Sher et al. (2005) for the LP glacial interval. The LP glacial period (Yedoma deposits) was influenced by climate variations, which resulted in alternating wetter or drier environmental conditions in NE Siberia (Andreev et al., 2009). The generally cool climate and anaerobic soil conditions (due to water-saturated soils) during the LP glacial period slowed rates of soil OM decomposition (Dutta et al.,

2006) and increased the accumulation of OM (Andreev et al., 2011; Schirrmeister et al., 2011a; Wetterich et al., 2014). The TOC-rich Yedoma deposits (comprising MIS 3 and partly MIS 4) with their higher aliphatic proportion (Figs. 3a and b) are indicative of such depositional environments. However, those LP glacial period samples with lower TOC contents and HI values (LP3, LP6; Fig. 2a, c) revealed a minor aliphatic character, which likely reflects a change to a drier depositional environment with less-water-saturated soils during this time interval.

In contrast, the low TOC concentrations and HI values of the Eemian samples (MIS 5e, Table 1) point to less OM accumulation and/or an increased level of OM decomposition. This could reflect the warmer and drier climate of the Eemian period in NE Siberia (Andreev et al., 2004; Wetterich et al., 2014, 2016), which might have supported intense aerobic microbial degradation of OM due to drier soil conditions (Andreev et al., 2009). The last interglacial was characterized by higher summer temperatures compared to the LP glacial period and even to the Holocene (Bond et al., 2001; Shackleton et al., 2003; Kienast et al., 2008, 2011) and was accompanied by permafrost thawing and draining, thermokarst formation, and thermal erosion (Table 2) (Andreev et al., 2009). The open-system pyrolysis data (Fig. 3) for the Eemian deposits (E1, E2, E3) showed a more pronounced type III OM and a less-aliphatic character compared to the Yedoma samples. Comparing the Eemian with the Holocene interglacial deposits (both are interpreted to have comparable vegetation – Kienast et al., 2008), the Holocene samples were characterized by higher TOC contents, HI values, and aliphatic proportions. This might indicate less-decomposed OM during deposition and therefore more favorable characteristics for OM degradation in future than for the Eemian deposits.

In addition to the environmental conditions in our study, the OM characteristics might also have been influenced by the depositional settings. Both the Holocene and Eemian OM were deposited in thermokarst lake environments (Wetterich et al., 2009). Comparing the TOC contents from our study with those from thermokarst lakes in the Kolyma region further to the east (Peterse et al., 2014), it becomes clear that the Holocene and Eemian deposits on Bol'shoy Lyakhovsky Island show much lower TOC contents (Holocene: 0.21 to 1.81 % TOC; Eemian: 0.57 to 0.64 % TOC) than the Holocene thermokarst deposits from the Kolyma region (Holocene: 4.8 to 22.6 % TOC). However, Wetterich et al. (2009) reported that Holocene and Eemian lake deposits can vary between lacustrine (around 2 % TOC) and boggy (around 10 to 20 % TOC) characteristics. Thus, the Holocene and Eemian samples in our study seem to reflect lacustrine while those in the study of Peterse et al. (2014) represent boggy conditions. Overall, the thermokarst conditions did not lead to higher accumulation of OM in the recovered deposits on Bol'shoy Lyakhovsky Island. In contrast, warmer conditions during the Holocene and Eemian might have caused a higher OM decomposition. This would suggest

**Table 2.** Schematic summary compiling the assigned ages (time period and marine isotope stage (MIS) classification) of the core material from Bol'shoy Lyakhovsky Island based on Andreev et al. (2004, 2009) and Wetterich et al. (2009, 2014), the paleoenvironmental information on the different time intervals ([1] Schirrmeister et al., 2002; [2] Andreev et al., 2009, [3] Grosse et al., 2007; [4] Sher et al., 2005; [5] Schirrmeister et al., 2011a; [6] Wetterich et al., 2014), the core sections, and the assessment of the organic matter (OM) from different ages to act as a substrate provider for microbial greenhouse gas production (present (free acetate) and future (bound acetate) substrate potential). The substrate potential was classified into very good (++), good (+), and poor (−) based on the results of the current study. TS2

| Age | | Paleoenvironment | Core sections | | Substrate potential | |
|---|---|---|---|---|---|---|
| Period | MIS | | | | present | future |
| Holocene (interglacial) | 1 | - climate warming[1], permafrost thawing[2]<br>- moisture increase, thermokarst formation[2]<br>- unstable environmental conditions[2]<br>- dissected landscape influenced by local hydrology[3] | L14-05 | 1 | - | + |
| Late Pleistocene glacial period (glacial) — interstadial | 3 | - increased temperature and soil moisture[4]<br>- high organic matter accumulation[5]<br>- optimum: warm and dry (48 to 38 ka BP)[6]<br>- warmer summers, open vegetation[2] | L14-02 | 3-1 | - | ++ |
| | | | | 3-2 | ++ | ++ |
| | | | | 3-3 | + | + |
| Late Pleistocene glacial period (glacial) — stadial | 4 | - cold and dry climate[6]<br>- harsh climate conditions[2]<br>- thin snow cover, low precipitation[2] | L14-03 | 4-1 | + | + |
| | | | | 4-2 | - | + |
| Eemian (interglacial) | 5e | - warmer climate, open-grass tundra similar to today[2]<br>- permafrost thawing[2]<br>- optimum: 4-5 °C higher summer temperatures than today, shrub tundra[2] | L14-04 | 5e | - | - |

that, for our sample set, environmental factors had a stronger impact on OM accumulation than did depositional characteristics. More investigations on depositional settings that differ not only with depth and sediment age but also on a regional scale must be conducted to improve our insights into the role of different depositional settings in the OM characteristics in permafrost regions.

## 5.2 Signals of present and past microbial communities in permafrost deposits

In order to investigate whether the freeze-locked OM had already stimulated a diverse bacterial and archaeal community during deposition in the past, biomarkers for past microbial communities were examined. Intervals with increased abundance of biomarkers characteristic of methanogenic archaea are of particular interest. This will provide information on how the deposited OM of different ages had already stimulated microbial greenhouse gas production in the past, which will help to assess the potential of the OM for green-

house gas production upon future permafrost thaw. Since past microbial biomarkers could also be a degradation product of the presently living microbial community, the Bol'shoy Lyakhovsky samples were also screened with regard to microbial life markers to compare both biomarker records.

As life markers we used phospholipids with ester bound fatty acids (PLFAs) since these bacterial cell membrane components are rapidly degraded after cell death (Logemann et al., 2011). In contrast, intact polar lipids with ether-bound moieties (diether side chain or tetraethers) have only a restricted potential to act as life markers for bacteria or archaea due to their significantly higher stability (Logemann et al., 2011). Thus, since microbial communities generally consist of both bacteria and archaea, we used the PLFAs here as a general indicator for intervals of increased present microbial life.

The detection of PLFA life markers indicated the occurrence of living bacterial communities in all investigated cores from Bol'shoy Lyakhovsky. While the PLFA signals were low in the permafrost sequences, all ALs contained higher

concentrations of PLFAs (Fig. 2d). This indicates a larger microbial community in the surface layers and presumably also increased microbial activity, at least during the summer season. According to Knoblauch et al. (2013), permafrost surface layers contain a mix of newly produced and old OM; this OM can stimulate microbial activity during unfrozen periods. Signals of microbial life in permafrost deposits are strongly decreased compared to those seen in the AL. It has been suggested that the life marker signals in the permafrost section most likely represent living successors of the microbial community incorporated into the sediments during time of deposition (Bischoff et al., 2013). Thus, the permafrost preserved the microbial community of the past. Different studies have shown that the microbial cells in permafrost can be reactivated upon permafrost thaw, after which they are able to produce greenhouse gases (e.g., Knoblauch et al., 2013; Schuur et al., 2015; Treat et al., 2015; Walz et al., 2017).

GDGTs and archaeol represent past microbial biomass (Stapel et al., 2016). GDGTs and archaeol are the cores of former membrane lipids, which are already partly degraded as indicated by the loss of their head group moieties. However, the core lipids are very stable over geological timescales (Pease et al., 1998; Schouten et al., 2013) and can be found in many different habitats (Bischoff et al., 2013; Schouten et al., 2013). Past bacterial markers (brGDGTs – Weijers et al., 2006a, b) and archaeal markers (isoGDGTs and archaeol – Koga et al., 1993; Pancost et al., 2001) provide information on the abundance of a past microbial community and might provide indirect information about microbial activity during time of deposition. IsoGDGT-0 (no cyclopentyl rings in the tetraether alkyl chains) and archaeol are used as markers for methanogenic communities in permafrost and peatland environments (Pancost et al., 2011; Bischoff et al., 2014), whereas their relative proportion varies between different methanogenic genera (Koga and Mori, 2006). PLFA life marker profiles indicated abundant present microbial life only for the ALs and did not correlate with the past biomarkers. Thus, the data suggest that in the permafrost sequence the past markers represent a paleo-signal (Stapel et al., 2016).

The results on past bacterial and archaeal biomarkers (Fig. 2e to g) showed that intervals with increased concentrations often corresponded to increased OM contents (TOC, Fig. 2a) with higher aliphatic character (higher HI values). This could especially be observed in the Yedoma deposits with the core sections MIS 3-1 and 3-2 and the upper part of MIS 4-1 and 4-2 (Fig. 2c: LP1, LP2, LP4, and LP8). The archaeol profile (Fig. 2g) suggested the presence of methanogenic communities during these intervals and methane production from this kind of OM in the past. Thus, the microbial past markers indicate that the OM-rich Yedoma deposits supported an abundant microbial life including methanogenic archaea during time of deposition. Comparable trends (with some deviations when TOC contents are quite low) can be observed when relating the past

biomarkers to gTOC (grams of TOC) (Fig. S1 in the Supplement).

A slight increase in permafrost temperatures is expected to influence not only the soil-moisture content but also the abundance and diversity of the microbial community (Wagner et al., 2007). Thus, intervals with increased past biomarker concentrations in permafrost regions might reflect increased soil moisture during time of deposition forming favorable living conditions for anaerobic bacteria (Weijers et al., 2006a) and archaea (Wagner et al., 2007). According to Wetterich et al. (2014), the MIS 3 interstadial optimum occurred between 48 and 38 ka BP on Bol'shoy Lyakhovsky Island and was characterized by warmer conditions of tundra environments with water-saturated ALs (Meyer et al., 2002; Hubberten et al., 2004; Andreev et al., 2011). This link between warmer and wetter conditions and increased abundance of microbial biomarkers during the interstadial MIS 3 was already observed by Bischoff et al. (2013) on Kurungnakh Island and Stapel et al. (2016) on Buor-Khaya peninsula. In their studies, comparable or even higher concentrations of brGDGTs, archaeol, and isoGDGTs were detected for the Yedoma intervals deposited during warmer and wetter environmental conditions. In contrast, both studies indicated relatively high concentrations of archaeol (up to approximately $80 \, \mathrm{ng \, g^{-1}}$ of sediment and $250 \, \mathrm{ng \, g^{-1}}$ of sediment, respectively) in the AL sequence, which cannot be observed in the present study. Methanogenic archaea require anaerobic soil conditions. Thus, the biomarker data suggest lower soil-moisture conditions in the ALs of the drilled cores on Bol'shoy Lyakhovsky Island. An explanation for these lower soil-moisture conditions is that all cores were drilled on high-centered polygons, supporting drier surface conditions and therefore a deeper penetration of oxygen into the soils.

As outlined above, the Holocene (MIS 1) and Eemian (MIS 5e) successions in this study were deposited in a thermokarst lake environment (Andreev et al., 2004, 2009; Wetterich et al., 2009, 2014). Peterse et al. (2014) reported much higher concentrations of brGDGTs (ranging between 7037 and $47\,676 \, \mathrm{ng \, g^{-1}}$ of sediment as well as 146 and $211 \, \mathrm{\mu g \, g^{-1}}$ of TOC) for Holocene surface samples of thermokarst lakes in the Kolyma region than concentrations observed in the current study (20 to $834 \, \mathrm{ng \, g^{-1}}$ of sediment and 1.4 to $56 \, \mathrm{\mu g \, g^{-1}}$ of TOC, Tables S1 and S2). The main reason for this seems to be that larger amounts of OM stimulate a larger microbial community in the thermokarst lakes from the Kolyma region.

## 5.3 Microbial substrate potential for greenhouse gas generation

To assess the potential of the OM from different depositional ages to provide substrates for the production of greenhouse gases, acetate is used as an appropriate substrate for microbial metabolism (Ivarson and Stevenson, 1964; Sørensen

and Paul, 1971; Sansone and Martens, 1981; Balba and Nedwell, 1982). Acetate is the terminal electron acceptor for methanogens in cold-temperate environments (Chin and Conrad, 1995; Wagner and Pfeiffer, 1997), especially for ace-toclastic methanogens (Thauer, 1998). They are ubiquitous methanogenic archaea in anoxic environments and in permafrost sediments (Kobabe et al., 2004).

In this study two acetate pools were investigated: (1) the free-acetate pool within the pore water, representing an easily and quickly accessible substrate source for microbial metabolism, and (2) the bound-acetate fraction, which is still linked to the OM. The latter constitutes a future substrate source upon degradation (Glombitza et al., 2009b). The concentrations of bound acetate (Fig. 2h) in the investigated samples correlated well with the amount of TOC and also often with the HI values. Overall, the largest future substrate potential for microbial turnover is associated with MIS 3 (mean concentrations of $\sim 48.9$ mg L$^{-1}$), followed by MIS 4 ($\sim 33.26$ mg L$^{-1}$) and MIS 1 ($\sim 30.05$ mg L$^{-1}$) (Table 2). In contrast, the bound-acetate concentration in the Eemian deposits suggests a depleted and possibly already altered bound-substrate pool because the concentration is considerably lower ($\sim 9.98$ mg L$^{-1}$) than in all other deposits.

In the AL samples the very low concentrations of free acetate together with the elevated concentrations of PLFA life markers suggest a higher microbial consumption of free acetate by an active microbial community (Lee et al., 2012; Knoblauch et al., 2013; Stapel et al., 2016). This activity is most likely stimulated by, for example, warmer temperatures (thawing conditions) and the input of newly produced and old OM during the thawing period. AL deepening due to global warming especially increases the accessibility of formerly freeze-locked OM. This old OM has been reported to be particularly sensitive to temperature-induced microbial decomposition (Knorr et al., 2005; Davidson and Janssens, 2006) and, therefore, is considered to be an important substrate source for future microbial turnover.

In the present study, the highest PLFA concentration was detected in the AL of the core sequence containing MIS 3 deposits (core L14-02). This may reflect the high potential of the MIS 3 Yedoma OM to serve as a substrate provider for a living microbial community upon thaw. However, local environmental differences may also affect the PLFA concentration. For example, the core containing MIS 3 deposits was drilled on a stable tundra surface (core L14-02) with relatively stable AL conditions. In contrast, the other cores were either drilled in a geomorphologically dynamic terrace position with thermo-erosion (Schirrmeister et al., 2011a, b; Grosse et al., 2011) and a seasonal supply of sediment and water, or in a drained and refrozen Holocene thermokarst basin (core L14-05), which is characterized by lower ice contents and shallower ALs (Schwamborn and Wetterich, 2015). Nevertheless, the increased PLFA concentrations in all ALs indicate, to a certain extent, that the permafrost OM, at least from MIS 3, 4, and 1, can serve as a good substrate provider

in a future permafrost thawing scenario. For MIS 5e OM this could not be evaluated due to the lack of MIS 5e deposits with an AL on top.

In contrast to the bound-acetate concentrations, the free-acetate substrate pool only partly correlated with the TOC contents in the individual cores (in all cores: $R^2 < 0.5$). The reason for this might be that the free-acetate pool in permafrost pore waters is not only the result of acetate released from the organic source material but also of other factors influencing the free-acetate signal such as lateral and vertical diffusion promoted by capillary pressure (Parlange, 1971), thawing and freezing processes, and microbial production and consumption. However, similar trends among acetate (free and bound), TOC, and HI were observed at several depth intervals mainly within the MIS 3 and 4 deposits (e.g., core L14-02 at 1.5 and 4.2 m and core L14-03 at 2.5 and 5.8 m; Fig. 2). Here, the mean concentrations ($\sim 93.6$ and 82.1 mg L$^{-1}$) of free acetate are at least 2 to 3 times higher than those identified in the Holocene (46.1 mg L$^{-1}$) or Eemian (24.1 mg L$^{-1}$) interglacial periods. The smaller free-acetate pool in the Holocene deposits may be the result of intense microbial consumption during OM deposition in the past, as has been observed for the modern ALs (see above). This could have been supported by deeper and prolonged thawing of past ALs starting with the onset of the early Holocene, when warming resulted in unstable environmental conditions, especially later during the Holocene optimum (Andreev et al., 2004; Wagner et al., 2007; Wetterich et al., 2008). As a consequence, this warming could have increased active microbial acetate consumption (Xue et al., 2016). Similarly, the low concentrations of free acetate in the Eemian deposits may also be the result of increased microbial consumption due to warmer environmental conditions. The Eemian is another period associated with intensive permafrost thaw, which again likely altered the free-acetate concentration in the sediments due to the lateral transport of water and/or sediment (Andreev et al., 2009). Based on the results obtained in this study it can be noted that sediments from the interglacial periods contain reduced amounts of free acetate compared to those from the LP glacial period. Although the free-acetate pools in the Holocene and Eemian deposits are similarly low, the Holocene deposits contain at least a considerable bound-acetate future substrate potential (Fig. 2i; Table 2).

Overall, the Yedoma deposits (MIS 3 and MIS 4) contain the largest free- and bound-acetate substrate pools (Fig. 2h, i). They are rich in OM and are characterized by the highest abundance of past microbial biomarkers, including those resembling methanogenic communities in wetlands (Fig. 2e to g). The Yedoma organic-rich material often shows high HI values, assigning an increased aliphatic character to this OM (Fig. 3a, b). Thus, in contrast to simply using the TOC content, the HI values seem to represent a promising parameter for assessing the potential of permafrost OM to act as appropriate source material for microbial OM degradation.

OM with high HI is considered to contain a higher proportion of more easily degradable aliphatic molecular structures, whereas OM with a low HI contains a higher proportion of less easily degradable aromatic structures (Hedges et al., 2000).

Analog results were obtained for Yedoma deposits in a previous study on Buor Khaya Peninsula about 400 km SW from Bol'shoy Lyakhovsky Island (Stapel et al., 2016). Considering the thickness of the LP Yedoma deposits on Bol'shoy Lyakhovsky Island (20 m; Schennen et al., 2016) and across Siberia (10–60 m; Dutta et al., 2006), as well as the extension of these deposits across Russia (about 1 028 264 km$^2$; Grosse et al., 2013), it becomes obvious that the freeze-locked Yedoma deposits represent a significant substrate potential for future microbial greenhouse gas generation. Ongoing warming in the Arctic will increase AL depth, rendering deeper and older OM bioavailable for microbial decomposition.

## 6 Conclusions

The potential of the permafrost OM to provide organic substrates for microbial greenhouse gas production strongly varies within the investigated deposits from the Eemian to the present time and is mainly controlled by environmental and climatic conditions. Overall, the strongest present and future substrate potential appears to be stored within the Yedoma OM deposits from the last interstadial (MIS 3) and stadial (MIS 4) periods, which are characterized by increased HI values and a higher aliphatic character. Thus, this currently frozen Yedoma OM is likely to have a strong impact on the greenhouse-gas-driven climate–carbon feedback cycle upon thaw. In contrast, the interglacial periods (Holocene and especially Eemian) show lower substrate potentials, which might point to stronger microbial degradation during time of deposition. The Eemian deposits reveal both low present and low future substrate pools. However, the Holocene deposits at least contain a significant future substrate pool, which may become available when recycled in the AL.

*Data availability.* https://www.pangaea.de/TS3 (follows after acceptance and includes all shown datasets).

**The Supplement related to this article is available online at https://doi.org/10.5194/bg-15-1-2018-supplement.** TS4

*Author contributions.* JGS conducted subsampling of the core material; the laboratory analyses and data interpretations were guided by KM and BH. GS and LS planned and coordinated the fieldwork as well as drilled and processed the core material. JGS wrote the paper, which was commented on by all co-authors.

*Competing interests.* The authors declare that they have no conflict of interest.

*Acknowledgements.* This research was supported by the German Ministry of Education and Research as part of the bilateral CarboPerm Project between Germany and Russia (grant no. 03G0836B). We thank all Russian and German participants of the drilling expedition, especially Mikhail N. Grigoriev (Melnikov Permafrost Institute, Yakutsk, Russia) for his leadership. We thank the three reviewers for their thoughtful and very constructive comments and suggestions on our paper. Furthermore, a previous version of the paper benefited from valuable comments and English language correction from Candace O'Connor (Fairbanks, Alaska, USA).

The article processing charges for this open-access publication were covered by a Research Centre of the Helmholtz Association.

Edited by: Jack Middelburg
Reviewed by: R. Sparkes and two anonymous referees

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

## Remarks from the language copy-editor

## Remarks from the typesetter