# Peer review of "Substrate potential of last interglacial to Holocene permafrost organic matter for future microbial greenhouse gas production"

_Biogeosciences, 2017_

## Referee Comment (RC1) · Anonymous Referee #1 · 13 Apr 2017

Stapel et al have studied OM properties and microbial biomarkers throughout a permafrost section from Siberia going back to MIS5, to identify the lability of OM, and thus the availability of potential substrate for greenhouse gas generation due to microbial decomposition upon permafrost thaw. The authors find that OM deposited during glacials and interstadial periods likely contain most substrate, and thus contribute most to greenhouse gas emission upon thaw. The manuscript is generally well written, although the English and the use of punctuation marks is getting a bit sloppy towards the end. I have a few concerns about the manuscript, which I would like to see addressed before I can recommend this work for publication in Biogeosciences:

Introduction: The link between the general introduction on permafrost thaw, its consequences, and the scope of this study can use some improvement. For example, Bol'shoy Lyakhovsky Island is suddenly mentioned on P2, L15. Later it appears to be the study site, but it needs some more context in the introduction. Similar for the 'Eemian deposits in this study...' (P2, L30). Also include acetate, and the difference between free and bound acetate.

Stratigraphy: The composite core consists of different lithologies, i.e. lacustrine (MIS1 and MIS5), floodplain deposits (MIS4), and also contains cryostructures. I miss a discussion on how different sources may influence OM paramters, lipid abundance and distribution, or acetate availability? For example, GDGTs in lacustrine (MIS1 and MIS5) and floodplain deposits (MIS4) may have a mixed soil and aquatic origin. This may influence your results, especially when considering that an earlier study has shown that Siberian thermokarst lakes (probably comparable to the MIS5 deposits in this study?) can contain >200 times the concentration of branched GDGTs compared to Yedoma (Peterse et al., 2014, JGR-B). Similarly, isoGDGTs, archaeol, and PLFA concentrations may be influenced too. Another way to check sources could be to calculate BIT index values (Hopmans et al., 2004, EPSL). Peterse et al., 2014 (JGR-B) found that Yedoma has a significantly lower BIT index than in soils. Similar changes should be visible throughout the composite core studied here.

Methods: P4, L30: were samples decalcified prior to determining TOC? Otherwise total carbon is reported instead of TOC. P5, L5: was there any pre-treatment of the sample material prior to pyrolysis?

Discussion: P9, L3: How/in what figure/parameter is the contribution of aquatic OM reflected? Similar for the input of aquatic OM during MIS3 (P9, L32). Please clarify. Again P12, L10-11: what indicates the link to moist depositional conditions?

P.9, L27: I can't follow this sentence. Check grammar/order of words. Also: if glacial conditions would slow down degradation, how does it influence its production?

P.10, L15 and following: How can you be certain that GDGTs reflect past microbial

biomass? In this study, only the core lipids are analyzed, whereas part of the GDGT pool may present as IPL, and thus derive from living biomass. Other studies have reported an IPL contribution of >30% to the GDGT pool in OC-rich soils (e.g. Peterse et al., 2011 Org Geochem). Furthermore, the manuscript refers to both microbial biomass and methanogens, of which the latter is of course more specific. Please go through the manuscript and check which level of specificity is relevant.

Fig. 2: Is there a reason why archaeol is plotted together with isoGDGTs? They do not necessarily share the same source. Instead, it would be more logical to plot archaeol next to e.g. GDGT-0/cren, which are both indicators of methanogenesis. These data can then also be used to compare with the acetate data.

L 20: Can you explain how exactly GDGT concentration data provides information on the activity of microbial biomass in the past? L23: Given that GDGT concentrations and TOC seem to covary (i.e. both higher in Yedoma?), it makes sense to normalize GDGT concentrations on TOC to distinguish between high GDGT concentrations due to high TOC content and actual elevated microbial biomass. Do the trends and conclusions still hold?

The choice of 'excellent' as description for substrate (in abstract and P10, L31) seems odd. In my opinion something can turn out to be, or has proven to be an excellent substrate, but you can not select something as an excellent substrate if it isn't compared to anything else.

The discussion on microbial activity in the active layer leads to the conclusion that MIS1, 3, 4 provide most substrate upon thaw. However, it is important to mention that there are only few active layer sediments included in this study, and that there are no active layer samples from MIS2 and MIS5 permafrost are included. There should be a few words on how representative this sample selection is. I furthermore miss the link between substrate availability and OM quality or composition and microbial biomarker abundance. I also miss a comparison with data from the literature. I have already mentioned Peterse et al 2014, JGR-B (Branched glycerol dialkyl glycerol tetraether in Arctic lake sediments: sources and implications for paleothermometry at high latitudes), but there are more biomarker papers on Siberian (or Arctic in general) permafrost soils, and there must be on microbial community composition and OM properties, too. To name a few: Bischoff et al 2013, GBC. Response of methanogenic archaea to Late Pleistocene and Holocene climate changes in the Siberian Arctic. Knoblauch et al., 2013, GCB, Predicting long-term carbon mineralization and trace gas production from thawing permafrost of Northeast Siberia. Blaud et al., 2015, Res in Microb. Arctic soil microbial diversity in a changing world. And papers citing those.

P11, L31: Can you support the drawn similarities between TOC and free acetate with statistics? Is their relation stronger during MIS3 and 4 compared to during MIS1 and 5?

P12, L2-5: check the grammar of this sentence, I can't follow the reasoning. I think 'favoured' needs to be replaced with 'a result of' ….caused by the onset of the Holocene, when a warming climate caused unstable environmental conditions…. P12, L5-9: This sentence also seems to lack punctuation marks, verbs, logical order of words. Please check.

Specific comments: P1, Line 15: delete excellent P3, L5: abbreviation (PLFA) seems out of place here. P4, L30: replace grounding by grinding. P5, L2: past tense of grind is ground, not grounded P5, L9, replace were by was P5, L15: replace grounded by ground P.5, L17: polarity fractions, or fractions of increasing polarity P5, L20: what does PL stand for? P10, L23: increased P11, L31: similarities….are… P11, L34: …the OM composition… P12, L5: reason P12, L10: replace 'implied' by 'caused' P12, L16: 'moist' or 'more moist'. 'Moister' does not exist. P12, L20: …gas generation is accessible within permafrost. P12, L20: delete 'an'.

Terminology: P.11, L20: what is old freeze-look permafrost? P. 11, L 27: what is a thermos terrace?

---

## Referee Comment (RC2) · Anonymous Referee #2 · 8 May 2017

General and specific comments

The authors describe a set of permafrost cores from Bol'shoy Lyakhovsky Island in the north-east Siberian Arctic that cover a time span from Eemian to Holocene, with the aim to assess the potential of organic matter from different periods as a substrate for microbial greenhouse gas production. However, I cannot fully follow the approach. The authors investigate organic matter quality using pyrolysis-GC-MS and Rock Eval, past and present microbial biomass using GDGTs and PLFAs, as well as concentrations of free and organic matter-bound acetate:

(1) I recommend to be careful with conclusions on organic matter degradability based on organic matter chemistry (see e.g. Schmidt et al., 2011, Nature).

[Figure]

(2) I do not understand the importance of past and present microbial biomass for this research question. There is no doubt about the presence of a living, active microbial decomposer community in the active layer, and the data are not interpreted to more detail. I would also like to point out that I am not aware of studies testing how fast PLFAs are degraded in continuously frozen soils, and that we therefore do not know for sure if PLFAs in permafrost really represent the living microbial community.

(3) As the authors themselves acknowledge (page 11, lines 12-14), acetate concentrations say little about acetate availability as this depends on the production rates of acetate from organic matter. I am therefore not sure about the value of this parameter in this context.

(4) I would appreciate more details on the applied methods that are also not contained in the cited previous publication (Stapel et al., 2016). In particular, what compounds were detected with pyrolysis-GC-MS and how were they evaluated to generate Figure 3? What PLFAs were detected and used to quantify total PLFAs? Were only bacterial or also fungal markers considered?

(5) The authors further present some interesting correlations between individual parameters, and the statistical approach should be described in the methods section. The underlying correlation matrix could also be presented in a separate table in the manuscript to give the reader a better overview.

(6) The authors use European terminology to describe the glacial cycles that is technically not correct for Siberia. I do not object in general since the European terms are well known and the authors also use Marine Isotope Stages to identify these periods, but I suggest to add at least a comment on the terminology to the text.

(7) If I understood correctly, the authors imply that old organic matter from deep, continuously frozen permafrost deposits might move upwards into the active layer and stimulate microbial activity there (e.g., page 10, lines 10-13; page 11, lines 20-24; Conclusions). While it is correct that an influx of additional organic carbon can stimulate microbial activity in soils ("priming effect"), I do not see by what mechanism organic compounds could move upwards from frozen into non-frozen parts of the soil.

(8) The manuscript contains many grammatical mistakes and would overall profit from some language polishing (some sentences are very long and difficult to understand).

(9) As my last general comment, I want to mention that I think the authors do a good job in keeping overview of the different depositional ages across the different cores. This is not an easy task.

Technical corrections

Page 1, line 19: What do you mean with "representing at least a future substrate potential upon release during OM degradation"?

Page 1, lines 28-30: I find it difficult to follow the causalities here, please rephrase.

Page 2, line 1: The upper 0-3 m.

Page 2, lines 2-4: I suppose you mean that the freeze-locked OM might thaw and/or be converted into $CO_2$ or $CH_4$, potentially inducing a positive feedback to global warming. Since the consequences of permafrost thaw are described in more detail later in the paragraph anyway, I suggest deleting this sentence. If you want to keep the sentence, please be more concrete.

Page 2, line 6: What do you mean with "drastic changes in the ecosystem"?

Page 2, lines 8-10: The sentence is not clear to me. The term "re-mobilization" usually refers to the export of previously frozen OM or nutrients into aquatic systems (e.g., in the cited Vonk et al. reference), but this is not the cause for increasing decomposition rates or accessibility of OM for microbial degradation. Rather, permafrost thaw leads to increased microbial activity and consequently increased decomposition rates, as well as to increased export into aquatic systems.

Page 2, line 10: "preserved".

Page 2, line 10: Delete "the".

Page 2, lines 12-14: This sentence mainly repeats the statements in the sentences before, I suggest to include the experimental information there.

Page 2, line 28: Delete "of".

Page 3, lines 9-10: Reference missing.

Page 3, lines 12-13: What do you mean with "feedback effects on permafrost deposits"?

Page 3, lines 15-16: I don't understand, please rephrase.

Page 4, line 30: "Grinding" instead of "grounding". See also page 5, lines 2 and 15.

Page 4, lines 30-31: Were samples acidified before TOC analysis?

Page 5, line 9: Change "were" to "was".

Page 5, line 23: Change to either "using medium-pressure liquid chromatography" or "using a medium-pressure liquid chromatograph".

Page 6, line 1: I suggest using "includes" instead of "consists".

Page 6, line 10: Change "marker" to "markers".

Page 6, line 19: Do the PLFA and TOC concentrations also correlate?

Page 6, line 23: Change "is" to "are".

Page 6, line 23: "Significantly": Was this statistically tested, and if yes, how? Otherwise, I would use another word.

Page 7, line 2: What do you mean with a partial correlation between past bacterial and archaeal markers?

Page 7, line 6: Should be plural, "concentrations".

Page 7, line 7: "Rise".

Page 7, line 23: Do p-value and R2 refer to correlations of both bacteria and archaea with TOC?

Page 8, line 8: "Scatters".

Page 8, line 23: I suggest using "weakest" instead of "smallest". Same for line 26 ("strongest" instead of " highest").

Page 8, line 30: What do you mean with "assigned"? How does OM degradability depend on the amount of OM? And how do you distinguish OM composition and OM quality?

Page 9, line 10: The main mechanism by which TOC/TN decreases during OM decomposition is the faster loss of C than N due to microbial respiration.

Page 9, line 13: Change to "... while OM with low HI contains ..."

Page 9, line 19: Please add a reference here.

Page 9, lines 27-29: Are you referring to the last interstadial here? I also noticed that Table 2 suggests both a dry climate and moist soils during that period, this seems rather strange. There is also something wrong with the grammar in the first part of the sentence.

Page 9, lines 30-31: "Moisture increased depositional settings": please rephrase.

Page 9, line 33: Change "minor" to "lower".

Page 10, line 11: The Fontaine paper is not about permafrost.

Page 10, lines 11-12: I do not understand.

Page 10, line 12: Which is not surprising considering that the active layer is seasonally thawed and the deeper permafrost is continuously frozen.
Page 10, lines 13-14: PLFAs inform about microbial biomass, not activity.

Page 10, line 20: Please add a reference for GDGTs as indicators of microbial activity. Also, the data presented show TOC concentrations, not accumulation rates.

Page 10, line 23: "increased".

Page 10, lines 23-25: The observed coincidence of high GDGT and OM concentrations does not necessarily imply certain environmental conditions. Microbial biomass is often correlated with OM concentrations since most microorganisms use OM as substrates.

Page 10, line 32: I suppose you mean microbial metabolism, not turnover.

Page 11, line 15: What do you mean with "soil biogeochemistry composition"?

Page 11, lines 16-17: Speculation.

Page 11, line 19: It is true that input of fresh OM by plants might additionally stimulate the microbial community in active layers, but I would expect that the main reason for the higher microbial biomass is the fact that the active layer is thawed in summer (i.e., provides liquid water).

Page 11, lines 20-21: What do you mean with the incorporation of frozen permafrost carbon into the active layer? I also think it should be "freeze-locked".

Page 11, line 25: Change to "... differences also affect the PLFA concentration."

Page 11, line 26: "thermos"?

Page 11, line 27: Change to "continuous".

Page 11, line 31: Change "is" to "are".

Page 11, line 34: What do you mean with "OM composition deposited"?

Page 12, line 1: Microbial consumption in the active layer is an ongoing process and not restricted to the "time of deposition".

Page 12, line 2: What do you mean with "stronger pronounced"? I'm afraid I cannot follow the entire sentence.

Page 12, line 5: Singular: "reason".

Page 12, line 10: I am not sure "implied" is the right word.

Page 12, line 11: What do you mean with "microbial acetate consumption on a regional scale"?

Page 12, line 20: I suggest changing to ". . . a significant substrate pool for future microbial greenhouse gas generation might become accessible within thawing permafrost."

Page 12, lines 22-26: I think this sentence is incomplete. I certainly cannot follow the grammar.

Page 12, line 27: "appears".

Page 12, line 32: "a strong impact".

Page 13, line 8: "guided".

Page 27, line 6: "Eglinton"

Page 29: How were the plus and minus signs assigned?

---

## Referee Comment (RC3) · R. Sparkes (Referee) · 11 Jun 2017

R. Sparkes (Referee)

r.sparkes@mmu.ac.uk

The authors have produced an interesting study investigating the material present within a permafrost core, and describing both the possibility that it has been degraded over time, and the potential for future degradation as climate warms. This is important work, preparing the ground for estimates of $CO_2$ release from thawing permafrost.

Major comments

Methods section – there is no explicit mention of the technique used to measure acetates, or it's not mentioned clearly. Since this is the main purpose of the paper it

should be obvious what has been done to measure the acetate compounds.

A lot of the molecular concentrations are reported as per gram sediment, but this leads to depth profiles that mostly correlate with OC content. Reporting molecular concentrations per gram carbon may lead to more interesting comparisons along and between cores.

Permafrost soils and Yedoma can have very different biomarker compositions. For example, GDGTs are being used as microbial biomarkers, but Sparkes et al., Biogeosciences, 2015 showed that GDGT concentrations are low in Yedoma sediments. Bacteriohopanepolyols may be better tracers of microbial activity in this region (see for example Bischoff et al., Biogeosciences, 2016). When linking timescales to substrate potential, the different sediment types within each core need to be shown in figures and discussed as well, since there could be a combination of climatic and sedimentological controls on OM quality and substrate potential.

P9 Line 1 – it is asserted that the permafrost deposits are dominated by terrestrial OM. Since GDGTs were measured, the BIT index could be used to confirm this.

P1 Line 23 – The GDGTs seem to correlate with TOC in the core sections referenced here. Relative increases in these molecules may support increased bacterial productivity, but if the biomarkers are changing with TOC then it may just represent variations in preservation. Once more, other markers for microbial activity would add value to the study.

Minor comments

P5 Line 24 - Was an internal GDGT standard used? The synthetic C46 GDGT standard is commonly used to provide accurate quantification of GDGTs, correcting for biases in response factor.

P5 Line 24 – The GDGT biomarker molecules being measured are not defined.

P 9 Line 15 – Absence of a particular biomarker does not necessarily mean that it has

decomposed, it may never have been present.

P10 Line 10-15. This section is hard to understand, rephrasing may help

Typos

P2 Line 10 – preserved

P2 Line 11 – gases

P3 Line 32 – no comma needed after intervals

P4 Line 30, and elsewhere – grinding rather than grounding; gorund rather than grounded

P6 Line 21 – sentence does not make sense, especially "whereas"

P6 Line 23 – is should be are

P8 Line 4 – commas required after 'are' and 'sediment'

P9 Line 23 – "nonene" rather than "nonen"?

P9 Line 27 – 'have' does not make sense

P11 Line 20 – freeze-locked

P12 Line 11 – comma required after 'transportation'

Figure 3 caption – Eglinton not Eglington

---

## Author Comment (AC1) · 7 Jul 2017

Responses to the reviewer's comments: "Substrate potential of Eemian to Holocene permafrost organic matter for future microbial greenhouse gas production"

By Janina G. Stapel et al.

We thank the reviewer for his/her thoughtful and very constructive comments and suggestions on our manuscript which will improve the clarity and the quality of the paper. Below, we will address all listed issues, which are relevant for discussion.

Reviewer #1 To Introduction:

1) The link between the general introduction on permafrost thaw, its consequences, and the scope of this study can use some improvement. For example, Bol'shoy Lyakhovsky Island is suddenly mentioned on P2, L15. Later it appears to be the study site, but it needs some more context in the introduction. Similar for the 'Eemian deposits in this study...' (P2, L30). Also include acetate, and the difference between free and bound acetate.

We agree with the reviewer and will improve the first part of the general introduction by rephrasing, condensing and/or combining sentences, and better outlined the relation between global warming, permafrost thawing, OM availability, greenhouse gas production and it feedback on global warming and permafrost thaw. We agree to describe why Bol'shoy Lyakhovsky Island was selected as study site earlier in the introduction and why Eemian deposits are of specific interest. We will add some sentences on acetate as a quality indicator for greenhouse gas production and the difference between free and bound acetate into the introduction. Finally, we will improve the context of our scientific question: "The feedback between climate warming and microbial greenhouse gas generation from thawing permafrost is a topic of intensive modern scientific debate (Zimov et al., 2006; Koven et al., 2011; Schuur et al., 2015). Especially the contribution of OM from thawing permafrost deposits of different ages to the climate carbon feedback cycle is still an open question. Therefore, the aims of this study are (1) to compare the stored potential for microbial greenhouse gas production in permafrost deposits from different glacial/ interglacial and stadial/ interstadial periods, and (2) to assign the substrate potential of different permafrost units to characteristic OM parameters and palaeoenvironmental deposition conditions. Furthermore, the Eemian deposits in this study are used as model for an interglacial period containing information on how an ongoing warming climate in the Arctic may affect permafrost OM degradation."

To Stratigraphy:

2) The composite core consists of different lithologies, i.e. lacustrine (MIS1and MIS5), floodplain deposits (MIS4), and also contains cryostructures. I miss a discussion on
how different sources may influence OM parameters, lipid abundance and distribution, or acetate availability? For example, GDGTs in lacustrine (MIS1 and MIS5) and floodplain deposits (MIS4) may have a mixed soil and aquatic origin. This may influence your results, especially when considering that an earlier study has shown that Siberian thermokarst lakes (probably comparable to the MIS5 deposits in this study?) can contain >200 times the concentration of branched GDGTs compared to Yedoma (Peterse et al., 2014, JGR-B). Similarly, isoGDGTs, archaeol, and PLFA concentrations may be influenced too.

Our pyrolysis data indicate that the OM is mainly of terrestrial origin with some variations in the aliphatic character. Mainly samples deposited during the Late Pleistocene (LP) glacial period show this higher aliphatic character. We already discussed these differences in the aliphatic character as a result of different environmental conditions (higher soil moisture favorable for algae growth vs dry conditions, anaerobic soil conditions/reduced OM decomposition, cold vs warmer conditions etc.) rather than of different depositional settings (see chapter 5.1), which seems sometimes confusing. For instance, Holocene as well as Eemian samples were both deposited in a kind of thermokarst lakes, but they show significant differences in the amount and guality of the OM. Thus, the influence of the depositional settings on the OM parameters applied here is not clear yet and environmental impact on OM production and degradation might play a more important role. We will add a few sentences on this into the manuscript. In our study we cannot really see that the Holocene GDGTs or archaeol concentrations (deposited in thermokarst lakes) are significantly more abundant than in the LP glacial deposits. The data are in the same range in all time intervals. However, we will add a sentence on this into the manuscript. PLFAs are generally high in active layers and significantly lower in permafrost sequences. Thus, the PLFA signal is mainly determined by the thaw-front in summer and not so much from the depositional environment.

3) Another way to check sources could be to calculate BIT index values (Hopmans et
al., 2004, EPSL). Peterse et al., 2014 (JGR-B) found that Yedoma has a significantly lower BIT index than in soils. Similar changes should be visible throughout the composite core studied here.

We calculated the BIT index (according to Hopmans et al., 2004) and will integrated it into the text as well as provide the data in the table in the supplement (S1). The BIT index is usually 1 and only decrease in four MIS 3 samples (and in one AL of MIS 4) to 0.91 and, therefore, it supports the overwhelming terrestrial soil character. Our study found that the BIT in the Yedoma deposits (MIS 3 and 4) did not much differ from the BIT values indicated in the active layers or in MIS 1 and 5e. The results presented by Peterse et al. (2014), indicate a much lower BIT value of 0.82 for Yedoma deposits from the Duvannyi Yar cliff. According to Strauss et al. (2012), the Yedoma at Duvanny Yar is of polygenetic origin formed on a floodplain of the Kolyma River, where alluvial and fluvial processes were the controlling processes. On the other hand, the Yedoma deposits on Bol'shoy Lyakhovsky Island were not influenced by dynamic fluvial processes during deposition (except for the lowermost 40 cm of core L14-03, where gravel in the sediments might indicate stronger fluvial influence during deposition). Furthermore, the investigated permafrost deposits are not floodplain deposits and therefore are less influenced by alluvial processes (Andreev et al., 2009). That explains why - other than in Peterse et al., 2014 - we observe a higher BIT index in the Yedoma deposits at Bol'shoy Lyakhovsky Island and do not see relevant differences in the BIT index by comparing the individual deposits.

To Methods:

4) P4, L30: were samples decalcified prior to determining TOC? Otherwise total carbon is reported instead of TOC. P5, L5: was there any pre-treatment of the sample material prior to pyrolysis?

Samples for TOC analyses were decalcified before the measurement. We will add this information into the text. For both Rock-Eval pyrolysis and open-system pyrolysis
sample material was freeze-dried and ground (as indicated in the text). There was no additional pre-treatment of the samples before the measurement. Free-biomolecules were thermally removed before pyrolysis of the macromolecular organic matrix. We will add this information into the text, too.

To Discussion:

5) P9, L3: How/in what figure/parameter is the contribution of aquatic OM reflected? Similar for the input of aquatic OM during MIS3 (P9, L32).

Generally, based on our results the deposited OM in all investigated cores is of terrestrial origin. However, the results from Rock-Eval analysis (Hydrogen Index) indicate that the samples have varying proportion of a hydrogen-rich component (Fig. 3a). Opensystem pyrolysis shows that the samples with a higher HI reveal a higher aliphatic character (Fig. 3b). In contrast to aromatic compounds aliphatic compounds are assumed to be better degradable, thus, samples with higher HI (more aliphatic rich) seem to have a better OM quality in terms of biodegradability. This material also contains higher concentrations of potential substrates (e.g. acetate). A source of more aliphatic-rich OM could be algae, living in surface ponds or water saturated soils. Periods of increased soil-moisture for the Yedoma deposits were already indicated by Sher et al. (2005). Thus, higher HI point to a higher aliphatic character in permafrost OM which may reflect a higher proportion of aquatic OM. (We will add this to the text.) Figure 3a (after Eglinton et al., 1990) classifies the OM by different types and provides hints on the origin of the OM; whereby OM type I and II are characteristic for aquatic or marine OM. Especially, Yedoma samples (Fig. 3a; e.g. AL, LP1 and LP2) show a stronger proportion of hydrogen-rich OM (type II OM). The higher aliphatic character seems to be related to increased soil moisture favorable for algae growth during time of deposition or as it is observed today in the active layer during the thawing period. Figure 3b assigns the higher aliphatic character to the LP glacial samples. We will add some further information on the parameters and figures indicating "aquatic OM" to the discussion.

BGD
6) Please clarify. Again P12, L15-16: what indicates the link to moist depositional conditions?

We will delete "moist depositional conditions" from the text here, since it was not important at this position. As described under point 5) above the interpretation on moist depositional environment derived from the increased aliphatic character and from literature information for the LP glacial period.

7) P.9, L27: I can't follow this sentence. Check grammar/order of words. Also: if glacial conditions would slow down degradation, how does it influence its production?

We will rephrase this sentence as suggested.

8) P.10, L15 and following: How can you be certain that GDGTs reflect past microbial biomass? In this study, only the core lipids are analyzed, whereas part of the GDGT pool may present as IPL, and thus derive from living biomass. Other studies have reported an IPL contribution of >30% to the GDGT pool in OC-rich soils (e.g. Peterse et al., 2011 Org Geochem).

IPL with an ether bond moiety are less suitable to act as life markers due to their significantly increased stability (Logemann et al., 2011). Due to the similar structural moieties something similar can be suggested for the br-GDGTs and iso-GDGTs. In contrast IPLs with ester moieties are known to rapidly decrease after cell death and thus, since microbial communities usually consist of both bacteria and archaea, we use the PLFAs as life markers for the microbial community in general here. Since life marker and past marker do not match we interpret that past marker signal is a paleo-signal. We will add this information to the discussion in chapter 5.2.

9) Furthermore, the manuscript refers to both microbial biomass and methanogens, of which the latter is of course more specific. Please go through the manuscript and check which level of specificity is relevant.

Methanogens are more relevant when discussing acetate as a substrate for green-
house gas production. We will go through the manuscript to be more precise.

10) Fig. 2: Is there a reason why archaeol is plotted together with isoGDGTs? They do not necessarily share the same source. Instead, it would be more logical to plot archaeol next to e.g. GDGT-0/cren, which are both indicators of methanogenesis. These data can then also be used to compare with the acetate data.

The reason was to have one parameter for the past archaeal biomass and to safe some vertical space for the figure. Iso-GDGTs are mainly dominated by iso-GDGT-0 (supplement table S1) and both iso-GDGT-0 and archaeol are used for the presence of methanogens in permafrost regions (Bischoff et al., 2014; Pancost et al., 2011), although their ratio differs throughout different methanogenic genera (Koga and Mori, 2006). We will add some sentences on this to chapter 5.2. Thus, we would like to keep this combined archaeal parameter in Fig. 2, but provide all GDGT data in the supplement table S1.

11) L 20: Can you explain how exactly GDGT concentration data provides information on the activity of microbial biomass in the past?

Sure the GDGTs are not a direct measure of microbial activity. However, we can assume that if microorganisms can be found that they have been active. Thus, the presence of past markers concomitantly suggests the activity of these past microbial communities (when they formed the active layer community). To avoid further confusions, we will delete this part of the sentence from the text.

12) L23: Given that GDGT concentrations and TOC seem to covary (i.e. both higher in Yedoma?), it makes sense to normalize GDGT concentrations on TOC to distinguish between high GDGT concentrations due to high TOC content and actual elevated microbial biomass. Do the trends and conclusions still hold?

Yes they do. By normalizing the GDGT and archaeol concentrations to TOC, the same trends within the depth profiles of the cores are visible. Depths of increased TOC and
OM quality (e.g. core L14-02 at about 1.5 m with a HI of 246 mg HC/ g TOC and TOC of 4.7 wt%) correspond with increased concentrations of br-GDGTs. On the opposite, depths of decreased TOC but increased HI (e.g. core L14-03 at about 1.4 m with a HI of 254 mg HC/ g TOC and TOC of 1.9 wt%) reveal increased br-GDGT and archaeol concentrations indicating that the concentration of living microorganisms in the past not only depends on the amount of TOC but also on the quality of the OM (HI values).

13) The choice of 'excellent' as description for substrate (in abstract and P10, L31) seems odd. In my opinion something can turn out to be, or has proven to be an excellent substrate, but you cannot select something as an excellent substrate if it isn't compared to anything else.

Acetate is well-known to be intensively used as a substrate for microbial metabolism. Thus, based on this background knowledge acetate was classified as being "an excellent substrate" especially for methanogenesis (thus, compared to what we know from literature). However, we will replace "excellent" by "appropriate".

14) The discussion on microbial activity in the active layer leads to the conclusion that MIS1, 3, 4 provide most substrate upon thaw. However, it is important to mention that active layer sediments included in this study, and that there are no active layer samples from MIS2 and MIS5 permafrost are included. There should be a few words on how representative this sample selection is.

Reading this comment, we agree that this sentence can be interpreted in this way. However, what we wanted to say is that there are already "natural" examples that old material can stimulate microbial life again. We are aware that we cannot say anything about the potential of MIS 5e material, since there was no active layer for the MIS 5e available in the field. We will add the following sentences:"...representing the overlaying soil as part of the active layer above the permafrost deposits from MIS 1, 3 and 4. Due to the stratigraphic settings at the study site on Bol'shoy Lyakhovsky Island, active layers containing OM from MIS 2 and MIS 5e could not be obtained in the field".

BGD
15) I furthermore miss the link between substrate availability and OM quality or composition and microbial biomarker abundance. I also miss a comparison with data from the literature. I have already mentioned Peterse et al 2014, JGR-B (Branched glycerol dialkyl glycerol tetraether in Arctic lake sediments: sources and implications for paleothermometry at high latitudes), but there are more biomarker papers on Siberian (or Arctic in general) permafrost soils, and there must be on microbial community composition and OM properties, too. To name a few: Bischoff et al 2013, GBC. Response of methanogenic archaea to Late Pleistocene and Holocene climate changes in the Siberian Arctic. Knoblauch et al., 2013, GCB, Predicting long-term carbon mineralization and trace gas production from thawing permafrost of Northeast Siberia. Blaud et al., 2015, Res in Microb. Arctic soil microbial diversity in a changing world. And papers citing those.

We will include a paragraph on the relation of OM quality or composition and microbial abundance, and a sentence on OM quality and substrate availability. Furthermore we will add a paragraph were we compare our results with literature data.

16) P11, L31: Can you support the drawn similarities between TOC and free acetate with statistics? Is their relation stronger during MIS3 and 4 compared to during MIS1 and 5?

The statistical similarity of the bound acetate fraction with TOC (average  $R^2$  for all cores= 0.7-0.8) is much better than for the free acetate (for all cores  $R^2

when a warming climate caused unstable environmental conditions ....

We will correct the sentence according to the reviewer's suggestions.

18) P12, L5-9: This sentence also seems to lack punctuation marks, verbs, logical order of words. Please check.

We will rewrite and split this sentence.

Specific comments:

We thank the reviewer for his/her comments and will follow all his/her suggestions.

Terminology:

P.11, L20: what is old freeze-look permafrost; - it has to be freeze-locked permafrost

P. 11, L 27: what is a thermos terrace?; - It has to be "thermo terrace" which is also known as "thermo-erosional valley" (Schirrmeister et al., 2011).

---

## Author Comment (AC2) · 7 Jul 2017

Responses to the reviewer's comments: "Substrate potential of Eemian to Holocene permafrost organic matter for future microbial greenhouse gas production"

By Janina G. Stapel et al.

We thank the reviewer for his/her thoughtful and very constructive comments and suggestions on our manuscript which will improve the clarity and the quality of the paper. Below, we will address all listed issues, which are relevant for discussion.

Reviewer #2

**General comments**

1) I recommend to be careful with conclusions on organic matter degradability based on organic matter chemistry (see e.g. Schmidt et al., 2011, Nature).

In our understanding Schmidt et al. 2011 claimed that the persistence of soil OM primarily not depend on the molecular properties of the OM itself but on the physicochemical and biological properties of the surrounding environment. In permafrost regions the main factor controlling the ecosystem is the cold temperature, which causes reduced OM decomposition and therefore increased OM accumulation. Thus, permafrost can contain OM which is highly vulnerable to microbial degradation as it was not fully degraded due to the low environmental temperature conditions (short periods of microbial activity during summer season). Although microbial degradation might be impeded by the conditions in permafrost areas, microorganisms will not degrade OM randomly upon thaw. Specific structural moleties are better degradable (less energy demanding) than others for the microbial community involved in the degradation processes (aromatics vs. aliphatics). These structural differences in the OM depend on the OM source and the level of microbial degradation before freeze-locking. In the current paper we addressed whether we can (i) trace these structural differences in the deposited OM by pyrolysis and (ii) characterize a stored potential substrate pool for microbial turnover. Thus, we apply these two parameters as quality indicators for the stored OM to act as a substrate provider for microbial degradation upon permafrost thaw. The detected microbial abundance and activity in the active layers on top of the permafrost (Fig. 2) as well as incubation experiments of old permafrost material indicate that old OM indeed can act as a substrate provider. Thus, our approach is more to discuss the OM quality in terms of potential substrate provision as outlined in the final conclusions.

2) I do not understand the importance of past and present microbial biomass for this research question. There is no doubt about the presence of a living, active microbial decomposer community in the active layer, and the data are not interpreted to more

**BGD**
detail. I would also like to point out that I am not aware of studies testing how fast PLFAs are degraded in continuously frozen soils, and that we therefore do not know for sure if PLFAs in permafrost really represent the living microbial community.

In order to investigate whether the freeze-locked OM stimulated already a microbial community during its deposition in the past, biomarkers for past microbial communities were examined. Since past microbial biomarkers could also be a product of microbial degradation by the presently living microbial community, the Bol'shoy Lyakhovsky samples were also screened with regard to microbial life markers to compare both biomarker records. PLFA life marker profiles indicate abundant microbial life in the active layers compared to the permafrost deposits and do not correlate with the past markers profiles. Thus, the data suggest that in the permafrost sequence the past marker represent a paleo-signal. The significant difference in PLFA concentration between the active layer and the permafrost deposits suggest that PLFAs also in permafrost environments can be used as a life marker. Other studies have shown that microorganisms can survive in deep permafrost sediments (e.g. Gilichinsky and Wagener, 1995; Rivkina et al., 2004; Bischoff et al., 2013) and incubation experiments have measured microbial produced CO2 after thawing permafrost sediments (e.g. Knoblauch et al., 2013; Walz et al., 2017), indicating living microbial cells in frozen permafrost sediments, which can be "re-activated" with permafrost thaw and then consume the stored OM (We will add this to the text). Thus, the low numbers of PLFA in the permafrost seems indeed represent living microorganisms in the permafrost sequence. For comparison also see Stapel et al., 2016. Overall, PLFA life marker profiles only indicate abundant microbial life for the active layers and do not correlate to the past markers in the permafrost sequence. Thus, the data suggest that in the permafrost sequence the past marker represent a paleo-signal (see chapter 5.2).

3) As the authors themselves acknowledge (page 11, lines 12-14), acetate concentrations say little about acetate availability as this depends on the production rates of acetate from organic matter. I am therefore not sure about the value of this parameter

**BGD**
in this context.

This was misunderstood; free acetate is an easily consumed substrate for microbial metabolism. Thus, free acetate concentration is a good tool to assess the potential of the OM to provide substrates for microbial turnover. The same is valid for the bound acetate concentrations indicating the future potential of the OM for substrates release upon degradation. In the respective sentence we wanted to explain why the free acetate concentration might be more different from the TOC values than the bound acetate concentration. We will re-phrase this part.

4) I would appreciate more details on the applied methods that are also not contained in the cited previous publication (Stapel et al., 2016). In particular, what compounds were detected with pyrolysis-GC-MS and how were they evaluated to generate Figure 3?

As external standard we used n-butane and the pyrolysate products were identified with the aid of reference chromatograms. The peak areas were integrated and calculated using the AGILENT ChemStation software. For the Eglinton-diagram (Eglinton et al., 1990) o-xylene, 2,3-dimethylthiophene and n-nonene and for the Horsfield-diagram Horsfield et al. (1989) C1-C5 gases, C6-C14-n-alkanes and n-alkenes as well as C15 and longer n-alkanes and n-alkenes were integrated. We will integrate this to the methods chapter.

5) What PLFAs were detected and used to quantify total PLFAs? Were only bacterial or also fungal markers considered?

Further information on the quantification of the PLFAs was added in the sub-chapter "microbial lipid biomarker analysis". Bacterial PLFAs from 14:0 to 21:0 with corresponding iso- and anteiso-FAs as well as br- and unsaturated-FAs were considered. We will integrate this to the text.

6) The authors further present some interesting correlations between individual pa-
rameters, and the statistical approach should be described in the methods section. The underlying correlation matrix could also be presented in a separate table in the manuscript to give the reader a better overview. We will add an additional sub-chapter (3.4 Statistical approaches) to describe how the statistical parameters were calculated. All shown data will be available on https://www.pangaea.de for free download; therefore we decided not to include extra tables to present the underlying correlation matrix for every single calculation.

7) The authors use European terminology to describe the glacial cycles that is technically not correct for Siberia. I do not object in general since the European terms are well known and the authors also use Marine Isotope Stages to identify these periods, but I suggest to add at least a comment on the terminology to the text.

We totally agree to also add the Russian terminology into the text and into table 1.

8) If I understood correctly, the authors imply that old organic matter from deep, continuously frozen permafrost deposits might move upwards into the active layer and stimulate microbial activity there (e.g., page 10, lines 10-13; page 11, lines 20-24; Conclusions). While it is correct that an influx of additional organic carbon can stimulate microbial activity in soils ("priming effect"), I do not see by what mechanism organic compounds could move upwards from frozen into non-frozen parts of the soil.

This was also a misunderstanding. What is meant here is that old OM gets into the active layer with increasing active layer depth (deepening of thaw front) and therefore is incorporated into the active microbial carbon-cycle again during the thawing period. We will re-write the respective paragraphs making clear that the OM becomes available again due to permafrost thaw.

9) The manuscript contains many grammatical mistakes and would overall profit from some language polishing (some sentences are very long and difficult to understand).

We will revise the manuscript thoroughly.

**BGD**
10) As my last general comment, I want to mention that I think the authors do a good job in keeping overview of the different depositional ages across the different cores. This is not an easy task.

We thank the reviewer for the positive feedback.

Technical corrections:

Page 1, line 19: What do you mean with "representing at least a future substrate potential upon release during OM degradation"?

The investigated bound acetate is a not directly available substrate pool and can only be available upon liberation from the OM via geochemical or microbial alteration of the OM. We will add "(present substrate pool)" and "(future substrate pool upon degradation)" into the abstract. Additionally, to improve the understanding of the idea of this parameter sentences will be added into the introduction.

Page 2, lines 2-4: I suppose you mean that the freeze-locked OM might thaw and/or be converted into CO2 or CH4, potentially inducing a positive feedback to global warming. Since the consequences of permafrost thaw are described in more detail later in the paragraph anyway, I suggest deleting this sentence. If you want to keep the sentence, please be more concrete.

We will remove this sentence part, but explain the climate carbon feedback later.

Page 2, line 6: What do you mean with "drastic changes in the ecosystem"; - we will replace it by "changes in vegetation".

Page 2, lines 8-10: The sentence is not clear to me. The term "re-mobilization" usually refers to the export of previously frozen OM or nutrients into aquatic systems (e.g., in the cited Vonk et al. reference), but this is not the cause for increasing decomposition rates or accessibility of OM for microbial degradation. Rather, permafrost thaw leads to increased microbial activity and consequently increased decomposition rates, as well as to increased export into aquatic systems.
We were not aware that the term "re-mobilization" can only be used in the way the reviewer claimed. Here, we wanted to say that freeze-locked OM and nutrients are again taking part in the carbon cycling upon permafrost thaw. To avoid confusion we will replace this term by "Thawing of permafrost promotes the accessibility of the formally preserved OM and nutrients for microbial turnover again, which results in increased microbial activity and consequently in increased OM decomposition rates (Dutta et al., 2006; Schmidt et al., 2011)."

Page 3, lines 9-10: Reference missing.

We will add the references "Weijers et al., 2006; Schouten et al., 2013" and "Pancost et al., 2001; Koga and Morii".

Page 3, lines 12-13: What do you mean with "feedback effects on permafrost deposits"?

We will rephrase the whole sentence: "... the contribution of thawing permafrost deposits of different ages to the carbon-climate cycle is still an open question"

Page 4, lines 30-31: Were samples acidified before TOC analysis?

Yes they were. We will add this information to the text.

Page 6, line 19: Do the PLFA and TOC concentrations also correlate?

There are some similarities between PLFA profile and TOC (especially MIS 1, MIS 5e) but they show no overall correlation. We will add this information to the text.

Page 7, line 2: What do you mean with a partial correlation between past bacterial and archaeal markers?

Past bacterial and archaeal biomarker show higher abundances in the same depth interval, but the curves do not directly correlate.

Page 7, line 23: Do p-value and R2 refer to correlations of both bacteria and archaea

**BGD**
with TOC?

The p-value and  $R^2$  referred only to the correlation of the br-GDGTs with TOC. We will rephrase the sentence to make that more clear.

Page 8, line 30: What do you mean with "assigned"? How does OM degradability depend on the amount of OM? And how do you distinguish OM composition and OM quality?

We will add a paragraph at the beginning of each discussion chapter to introduce into the following discussion (chapter 5.1, 5.2, 5.3). Thus, the respective part will be rephrased becoming part of the starting paragraph: "When permafrost thaws, formerly freeze-locked OM becomes bioavailable again. In this context, it is of utmost interest for the assessment of the impact of this OM on future climate evolution not only to determine the abundance but also to learn more about the quality of the OM with regard to its potential degradability. For instance, terrestrial OM (more aromatic rich) is considered to be more recalcitrant than aquatic OM (more aliphatic rich) (Hedges et al., 2000). Thus, the quality of the OM in permafrost deposits is determined by its source and, therefore, structural composition as well as by its alteration due to early diagenetic degradation processes during its deposition in the past (White, 2013). In the following chapter we apply pyrolysis techniques (Rock-Eval pyrolysis and open-system pyrolysis GC-FID) of the OM to get a deeper insight into the structural composition and to establish a new tool for OM quality assessment as introduced in Stapel et al. (2016)."

Page 9, line 10: The main mechanism by which TOC/TN decreases during OM decomposition is the faster loss of C than N due to microbial respiration.

We agree with the reviewer and will add this to the sentence.

Page 9, line 19: Please add a reference here;- we will add "Andreev et al., 2009 " as reference.

Page 9, lines 27-29: Are you referring to the last interstadial here? I also noticed that
Table 2 suggests both a dry climate and moist soils during that period, this seems rather strange. There is also something wrong with the grammar in the first part of the sentence.

In this sentence we are referring to the Late Pleistocene (LP) glacial period in general, which is characterized by cold-climate conditions with anaerobic soil conditions. These slowed down OM decomposition rates and increased the accumulation of OM during this period. In this study, we present permafrost deposits from a stadial and interstadial period within the LP glacial. Subordinated different climate conditions (e.g. dry and wet) during the overall cold-climate conditions during the LP glacial period characterize this interstadial and stadial period. We will rephrase the sentence.

Page 10, line 11: The Fontaine paper is not about permafrost; - we removed this reference and replaced it by Knoblauch et al., 2013.

Page 10, lines 11-12: I do not understand. + Page 10, line 12: Which is not surprising considering that the active layer is seasonally thawed and the deeper permafrost is continuously frozen.

We will delete this here and shift it to chapter 5.3, where it will be rephrased to: "However, the increased PLFA concentrations in all active layers indicate to a certain extent that the permafrost deposits at least from MIS 3, 4 and 1 can serve as good substrate providers when thawed. For MIS 5e this could not be evaluated due to the lack of MIS 5e deposits with an active layer on top."

Page 10, lines 13-14: PLFAs inform about microbial biomass, not activity. + Page 10, line 20: Please add a reference for GDGTs as indicators of microbial activity. Also, the data presented show TOC concentrations, not accumulation rates.

That is true, but high abundance of PLFAs in a surface near seasonally thawed deposit highly suggest that the living microbes are somehow active, especially if comparing with the PLFA data from the deeper permafrost deposits. We will rewrite the whole
paragraph: "According to Knoblauch et al. (2013), permafrost surface layers contain both fresh organic material and old OM (within the permafrost), which can stimulate microbial activity. The high concentrations of PLFAs in the active layer suggest that not only the abundance of microbial life is increased in this layer, but also the microbial activity." Also GDGTs provide no direct measure on the activity, but the high abundance of these past markers suggest that they have been active in the past during time of deposition. We will rephrase this sentence to: "...and indirectly their abundance might say something about their activity during time of deposition."

Page 10, lines 23-25: The observed coincidence of high GDGT and OM concentrations does not necessarily imply certain environmental conditions. Microbial biomass is often correlated with OM concentrations since most microorganisms use OM as substrates.

By normalizing the GDGT and archaeol concentrations to TOC, the same trends within the depth profiles of the cores are visible. Depths of increased TOC and OM quality (e.g. core L14-02 at about 1.5 m with a HI of 246 mg HC/ g TOC and TOC of 4.7 wt%) correlate with increased concentrations of br-GDGTs. On the opposite, depths of decreased TOC but increased HI (e.g. core L14-03 at about 1.4 m with a HI of 254 mg HC/ g TOC and TOC of 1.9 wt%) reveal increased br-GDGT and archaeol concentrations indicating that the concentrations of living microorganisms in the past not only depend on the amount of TOC but also on the quality of the OM (HI values). As the HI indicates a higher aliphatic character representing increased soil-moister conditions (Stapel et al., 2016), conclusions on changes in the past soil moisture can be derived. We will rewrite the paragraph about the past microbial markers.

Page 11, line 15: What do you mean with "soil biogeochemistry composition"?

We will rephrase this sentence to: "The reason for this might be that free acetate pool in permafrost pore waters is not only the result of acetate released from the OM, but also can be influenced by other factors e.g. lateral and vertical diffusion promoted by capillary pressure (Parlange, 1971), thawing and freezing processes as well as
microbial production and consumption."

Page 11, lines 16-17: Speculation.

The sentence will be rephrased to: "The very low concentrations of free acetate and elevated concentrations of PLFA life markers detected in the investigated active layer samples suggest a higher microbial consumption of free acetate by an active microbial community (Lee et al., 2012; Knoblauch et al., 2013; Stapel et al., 2016)."

Page 11, line 19: It is true that input of fresh OM by plants might additionally stimulate the microbial community in active layers, but I would expect that the main reason for the higher microbial biomass is the fact that the active layer is thawed in summer (i.e., provides liquid water).

Sure the reason for the activity in the surface layer is the fact that this part is thawing during summer, but we think also the kind of OM is important for microbial degradation (what is more easily degradable for a microorganisms). We will rephrase the paragraph and consider the reviewer's comment.

Page 11, lines 20-21: What do you mean with the incorporation of frozen permafrost carbon into the active layer?

Due to increasing active layer thickness, more old freeze-locked permafrost carbon is integrated in the microbial degradation processes in the active layer during the thawing period. We will extend the sentence.

Page 11, line 26: "thermos"?

It has to be "thermoterrace" which is also known as "thermo-erosional valley" (Schirrmeister et al., 2011). We will replace it and add the reference to the text.

Page 12, line 1: Microbial consumption in the active layer is an ongoing process and not restricted to the "time of deposition".

This was a misunderstanding. It was meant when the Holocene deposits were part of
the active layer in the past. We will rephrase this to: "The minor free-acetate pool in the Holocene deposits may be the result of the OM composition or of intense microbial consumption during OM deposition in the past as has been proposed for the modern active layer."

Page 12, line 2: What do you mean with "stronger pronounced"? I'm afraid I cannot follow the entire sentence.

We will rephrase the sentence and replace "stronger pronounced" by "deeper and longer thawed"

Page 12, line 11: What do you mean with "microbial acetate consumption on a regional scale"?

Permafrost thaw enables lateral and vertical transportations of water, OM and sediment which can redistribute substrates. These redistributed substrates will probably be microbially consumed on/in another place (e.g. in a near located soil or lake). In other words, on a more regional scale (this study site only represents a small spot in the Siberian Arctic) lateral and vertical transportation of substrates probably will end up in microbial consumption. We will add further information into the text.

Page 29: How were the plus and minus signs assigned?

To visualize the OM quality and the substrate potential a relative scaling based on the results of this study was applied.

We thank the reviewer for his/her comments and will follow all his/her suggestions on the other (minor) technical corrections (not listed here).

BGD

---

## Author Comment (AC3) · 7 Jul 2017

Responses to the reviewer's comments: "Substrate potential of Eemian to Holocene permafrost organic matter for future microbial greenhouse gas production"

By Janina G. Stapel et al.

We thank R.Sparkes for his/her thoughtful and very constructive comments and suggestions on our manuscript which will improve the clarity and the quality of the paper. Below, we will address all listed issues, which are relevant for discussion.

Reviewer: R.Sparkes

[Figure]

Major comments

1) Methods section – there is no explicit mention of the technique used to measure acetates, or it's not mentioned clearly. Since this is the main purpose of the paper it should be obvious what has been done to measure the acetate compounds.

The method is described in chapter "3.2 Low molecular weight organic acids (LM-WOAs) analyses".

2) A lot of the molecular concentrations are reported as per gram sediment, but this leads to depth profiles that mostly correlate with OC content. Reporting molecular concentrations per gram carbon may lead to more interesting comparisons along and between cores.

Relation to gTOC provides information on the abundance of a parameter relative to TOC. As outlined above (reviewer 1, comment 12) there was not much change in the observed trends. Since we want to show the total abundance of microorganisms and substrates we would like to keep the gSed relation.

3) Permafrost soils and Yedoma can have very different biomarker compositions. For example, GDGTs are being used as microbial biomarkers, but Sparkes et al., Biogeosciences, 2015 showed that GDGT concentrations are low in Yedoma sediments. Bacteriohopanepolyols may be better tracers of microbial activity in this region (see for example Bischoff et al., Biogeosciences, 2016). When linking timescales to substrate potential, the different sediment types within each core need to be shown in figures and discussed as well, since there could be a combination of climatic and sedimentological controls on OM quality and substrate potential.

Well, bacteriohopanepolyols have not been investigated. The appeal of using GDGTs and archaeol is that you get information on bacteria and archaea, which is not the case for bacteriohopanepolyols. Also we are aware that GDGTs and archaeol do not represent all microorganisms, we at least get some insights of bacterial and archaeal

variations over time. As outlined in our response to comment 2 (reviewer 1) the depositional effect on the OM quality is not clear yet. However, we will add this into the discussion.

4) P9 Line 1 – it is asserted that the permafrost deposits are dominated by terrestrial OM. Since GDGTs were measured, the BIT index could be used to confirm this.

Please see comment 3 reviewer#1.

5) P1 Line 23 – The GDGTs seem to correlate with TOC in the core sections referenced here. Relative increases in these molecules may support increased bacterial productivity, but if the biomarkers are changing with TOC then it may just represent variations in preservation. Once more, other markers for microbial activity would add value to the study.

GDGTs are already degraded biomarkers from intact polar lipids. The core lipids (GDGTs) are regarded as relatively stable. Thus, we suggest that they represent the past abundance of microbial communities, although preservation aspects cannot fully be ruled out.

Minor comments

P5 Line 24 - Was an internal GDGT standard used?

An external archaeol standard is used for quantification. We will added this to the method chapter.

P5 Line 24 – The GDGT biomarker molecules being measured are not defined.

We will add a supplement table where all GDGTs are listed (table S1).

P9 Line 15 – Absence of a particular biomarker does not necessarily mean that it has decomposed; it may never have been present.

This sentence will be rephrased to: "Thus, the low TOC and TOC/TN values (< 5), in

addition to the low HI in the Eemian samples (MIS 5e, Table 1) may point to less favorable conditions for OM accumulation and/or an increased degree of OM decomposition and therefore to a reduced OM quality."

P10 Line 10-15. This section is hard to understand, rephrasing may help

This will be rephrased and shifted (see comments above) to: "However, the increased PLFA concentrations in all active layers indicate to a certain extent that the permafrost deposits at least from MIS 3, 4 and 1 can serve as good substrate providers when thawed. For MIS 5e this could not be evaluated due to the lack of MIS 5e deposits with an active layer on top." Typos

We thank R.Sparkes for his/her comments and will follow all his/her suggestions on the other minor comments (not listed here).

---

## Author Response (AR1)

[revised manuscript text omitted]

30 As study area in NE Siberia Bol´shoy Lyakhovsky Island in the Laptev Sea was selected, since it provides the excellent opportunity to investigate permafrost OM deposited from last interglacial to Holocene time. The last interglacial deposits have been interpreted as Eemian deposits with, based on pollen data, 4-5 °C higher summer temperatures than today (Andreev et al., 2004). Especially, these Eemian deposits forming a paleo-equivalent to the Holocene interval are otherwise rather difficult to assess. According to prior studies by Wetterich et al. (2014) and references therein, the cores investigated

in this study can be integrated into an already described environmental and climatic history. It has to be mentioned that similar interglacial deposits at Oyogoss Yar from the mainland coast opposite to Bol'shoy Lyakhovsky Island have recently been dated and reveal younger latest infrared optical stimulated luminescence (IR-OSL) ages than Eemian (Opel et al., 2017). However, since it is not clear yet whether both deposits really represent the same age window, we stay here with the interpretation based on the Bol'shoy Lyakhovsky deposits by Andreev et al. (2004). To access information on quality in terms of biodegradability of the freeze-locked OM, we examined characteristic OM parameters (amount and quality) and low molecular weight organic acids (LMWOAs). LMWOAs such as acetate are important and easily convertible substrates for microbial metabolism (Ganzert et al., 2007) and are therefore used as a quality indicator in terms of future microbial degradability of the sedimentary OM (Glombitza et al., 2009; Strauss et al., 2015; Stapel et al., 2016). Acetate is a well-known substrate for methanogenesis (Chin and Conrad, 1995) and, thus its concentration provides information on the greenhouse gas production potential of the respective OM (Stapel et al., 2016). Acetate can either be dissolved in pore water and cryostructures (e.g. segregated ice) of permafrost deposits as free substrate being directly bioavailable for microorganisms or it can be bound to the organic matrix (e.g. by ester-linkage) forming a future substrate pool upon liberation via geochemical or microbial alteration of the OM (Glombitza et al., 2009; Stapel et al., 2016). In addition, investigations of microbial biomarkers such as phospholipid fatty acids (PLFAs) and glycerol dialkyl glycerol tetraethers (GDGTs) are used to examine present and past microbial communities in the context of modern and past environmental conditions. Phospholipids are essential membrane components of living cells (Zelles, 1999) and are hydrolysed rapidly after cell death (White et al., 1979; Logemann et al., 2011), therefore their fatty acid side chain inventories are used as an indicator for viable microorganisms in sediments (Haack et al., 1994). In contrast, GDGTs and archaeol represent membrane lipids of past microbial biomass, since they are already partly degraded as indicated by the loss of their head groups (Pease et al., 1998). While archaeol and GDGTs with isoprenoid tetraether bridges (iso-GDGTs) represent archaeal biomass, GDGTs with branched tetraether bridges (br-GDGTs) derive from bacteria (Weijers et al., 2006). However, in this context it should be mentioned that the br-GDGTs biomarkers only represent part of the bacterial community (Weijers et al., 2006; Schouten et al., 2013), while the archaeal markers cover most of the past archaeal community (Pancost et al., 2001; Koga and Morii, 2006).

The feedback between climate warming and microbial greenhouse gas generation from thawing permafrost is a topic of intensive modern scientific debate (Zimov et al., 2006; Koven et al., 2011; Schuur et al., 2015). Especially the contribution of OM from thawing permafrost deposits of different ages to the climate carbon feedback cycle is still an open question. Therefore, the aims of this study are (1) to compare the stored potential for microbial greenhouse gas production in permafrost deposits from different glacial/ interglacial and stadial/ interstadial periods, and (2) to assign the substrate potential of different permafrost units to characteristic OM parameters and palaeoenvironmental deposition conditions. Furthermore, the 
[revised manuscript text omitted]
 not only to determine the abundance but also to learn more about the quality of the OM with regard to its potential degradability. For instance, terrestrial OM (more aromatic rich) is considered to be more recalcitrant than aquatic OM (more aliphatic rich) (Hedges et al., 2000). Although permafrost OM is mainly of terrestrial origin, also here the quality of the OM is determined by its different terrestrial sources and, therefore, structural composition as well as by its alteration due to early diagenetic degradation processes during its deposition in the past (White, 2013). In the following chapter we apply pyrolysis techniques (Rock-Eval pyrolysis and open-system pyrolysis GC-FID) on the OM to get a deeper insight into the structural composition and to establish a new tool for OM quality assessment as introduced in Stapel et al. (2016).

The permafrost deposits on Bol´shoy Lyakhovsky Island are dominated by terrestrial OM as indicated by the results of the open system-pyrolysis (Fig. 3a). This is supported by the BIT index-values ranging between 0.9 to 1 (table S1), which is based on the ratio of br-GDGTs and crenarchaeol and is close to one in soil OM (Hopmans et al., 2004). Samples from the active layer and the Late Pleistocene (LP) glacial period (comprising MIS 3 and MIS 4, Table 1) reveal in general a terrestrial OM source, however mixed with different proportions of aliphatic-rich OM. This is indicated by the results of the Rock-Eval (increased HI values; Fig 2c) and open-system pyrolysis (OM type II; Fig. 3a). The origin of this aliphatic-rich OM could be algae material, which is usually rich in aliphatic structural units (Kolattukudy, 1980). Thus, the samples with increased HI values might indicate OM accumulation during intervals of increased soil-moisture favourable for algae growth. Periods of increased soil moisture during the LP glacial period was already indicated by Sher et al. (2005). The highest accumulation of OM was found in the interstadial deposits of MIS 3 (core sections 3-1 and 3-2), with TOC values typical of Yedoma deposits (Schirrmeister et al., 2013). The measured TOC/TN values of 5 to 12 are within the range of terrestrial permafrost deposits reported for the NE Siberian Arctic (Wetterich et al., 2009; Schirrmeister et al., 2011a; Strauss et al., 2015). Usually, the TOC/TN ratio describes the amount of sedimentary OM that originates from aquatic vs. terrestrial sources and is commonly used to characterize the dominant origin of the OM (Meyers and Teranes, 2002). However, the TOC/TN signal can be overprinted by different processes during OM decomposition such as microbial consumption (preferred respiration of carbon vs. nitrogen) and pedogenic processes resulting in lower TOC/TN values for stronger decomposed OM (Carter and Gregorich, 2008). The HI is used as indicator for OM quality in terms of microbial degradability (Talbot and Livingstone, 1989; Stapel et al., 2016), since OM with a higher HI is considered to contain a

higher proportion of better degradable aliphatic molecular structures, whereas OM with a low HI contains a higher proportion of less degradable aromatic structures (Hedges et al., 2000). Thus, the low TOC and TOC/TN values (< 5), in addition to the low HI in the Eemian samples (MIS 5e, Table 1) may point to less favourable conditions for OM accumulation and/or an increased degree of OM decomposition and therefore to a reduced OM quality. This would be in line

5  with the warmer and drier climate of the Eemian period in NE Siberia (Andreev et al., 2004; Wetterich et al., 2014; Wetterich et al., 2016), which might have supported intense aerobic microbial degradation of OM due to dyer soil conditions (Andreev et al., 2009). The last interglacial was characterized by higher summer temperatures compared to the LP glacial period (Bond et al., 2001; Shackleton et al., 2003; Kienast et al., 2008, 2011) and accompanied by permafrost thawing, draining, thermokarst formation and thermal erosion (Table 2) (Andreev et al., 2009).

10  The same conclusions can be drawn from the component-specific analysis of the OM by open system-pyrolysis GC (Fig. 3). Here, the Eemian deposits (E1, E2, E3) show a more pronounced terrestrial OM type III character due to their higher content of aromatic components and lower content of aliphatic components compared to the Holocene (MIS 1) deposits (except of sample H1; Fig. 3a). Furthermore, by examining their aliphatic compositions in more detail (Fig. 3b), the differences between the Eemian and Holocene interglacial deposits (both are interpreted to have comparable vegetation (Kienast et al.,

15  2008)) are expressed by the higher aliphatic character and higher HI values of the Holocene deposits, which likely indicates less decomposed OM and higher OM quality in the Holocene compared to the Eemian deposits.

The LP glacial period (Yedoma deposits) was influenced by climate variations, which resulted into alternating wetter or drier environmental conditions in NE Siberia (Andreev et al., 2009). The generally cold climate and anaerobic soil conditions during the LP glacial period slowed rates of soil OM decomposition (Dutta et al., 2006) and increased the accumulation of

20  OM (Andreev et al., 2011; Schirrmeister et al., 2011a; Wetterich et al., 2014). The Yedoma deposits (comprising MIS 3 and partly MIS 4) composed of terrestrial OM with a higher aliphatic proportion (Figs. 3a and 3b) are indicative for a depositional environment with increased soil-moisture during deposition. Here, a higher input of aliphatic-rich presumably aquatic OM (e.g. algae material) is accompanied by an increased accumulation of TOC and higher values of HI (LP1, LP2, LP4, LP5, LP7, LP8; Fig. 2). In contrast, those LP glacial period samples with lower TOC and HI values (LP3, LP6; Fig. 2)

25  reveal a minor aliphatic character, which likely reflect a change to a drier depositional environment with less water-saturated soils at the respective time interval.

The effect of different depositional setting on the OM quality is not really clear yet. Our study suggests that OM deposited in soils from LP glacial time reveal a higher OM quality than the Holocene and Eemian OM deposited in thermokarst lake environments. However, since both Holocene and Eemian samples were deposited in comparable settings (thermokarst

30  lakes), the significant difference in the OM parameters observed here indicate that environmental conditions might have a stronger impact on the OM characteristics than the depositional settings. Thus, more investigation on different depositional settings not only with depth and sediment age but also on a regional scale has to be conducted to improve our insights into the different factors determining the quality of OM in permafrost regions.

Overall, the results from the open system-pyrolysis suggest a terrestrial OM source for all investigated samples with varying proportions of aliphatic-rich OM affected by shifts in soil moisture as observed for the Yedoma deposits, and presumably higher rates of decomposed OM during dryer conditions as observed for the Eemian deposits. Furthermore, since HI values appear to resemble the varying aliphatic character of the OM in the permafrost deposits, HI seems to be an appropriate parameter to assess the quality of the OM supporting results presented in Stapel et al. (2016).

**5.2 Signals of present and past microbial communities in permafrost deposits**

In order to investigate whether the freeze-locked OM already stimulated a microbial community during deposition in the past, biomarkers for past microbial communities were examined. Since past microbial biomarkers could also be a product of microbial degradation of a presently living microbial community, the Bol´shoy Lyakhovsky samples were also screened with regard to microbial life markers to compare both biomarker records. As life markers we used phospholipids with ester bound fatty acids (PLFAs), since these bacterial cell membrane components are rapidly degraded after cell death (Logemann et al., 2011). In contrast, intact polar lipids (IPL) with ether bond moieties (e.g. archaeol) have only a restricted potential to act as life markers for archaea due to their significantly higher stability (Logemann et al., 2011). Thus, since microbial communities generally contain of bacteria and archaea we used the PLFAs here as a general indicator for intervals of increased microbial life.

PLFA life markers indicate the occurrence of a living bacterial community in the investigated deposits from Bol´shoy Lyakhovsky Island. While the PLFA signals are low in the permafrost deposits, all active layers contain higher amounts of PLFAs indicating a larger bacterial community in these surface layers. According to Knoblauch et al. (2013), permafrost surface layers contain both fresh and old OM (within the permafrost), which can stimulate microbial activity. The high concentrations of PLFAs in the active layers suggest that not only the abundance of microbial life is increased in these layers, but also the microbial activity. Living microbial cells in permafrost deposits are strongly decreased compared to the active layers and it has been hypothesized that these are most likely living successors of the microbial community incorporated into the sediments during time of deposition (Bischoff et al., 2013). Different studies have shown that these cells can be re-activated upon permafrost thaw, after which they are able to produce greenhouse gases (e.g. Knoblauch et al., 2013; Schuur et al., 2015; Treat et al., 2015; Walz et al., 2017).

Glycerol dialkyl glycerol tetraethers (GDGTs) and archaeol represent past microbial biomass (Stapel et al., 2016). GDGTs and archaeol are the cores of former membrane lipids, which are already partly degraded as indicated by the loss of their head group moieties. However, the core lipids are very stable over geological time scales (Pease et al., 1998; Schouten et al., 2013) and can be found in many different habitats (Bischoff et al., 2013; Schouten et al., 2013). Past bacterial (br-GDGTs (Weijers et al., 2006)) and archaeal (iso-GDGTs and archaeol (Koga et al., 1993; Pancost et al., 2001)) markers provide information on the abundance of a past microbial community and indirectly might say something about microbial activity during time of deposition. Iso-GDGT-0 (no cyclopentyl-rings in the tetraether alkyl chains) and archaeol are used as markers for methanogenic communities in permafrost regions (Pancost et al., 2011; Bischoff et al., 2014), although their relative

proportion vary within different methanogenic genera (Koga and Mori, 2006). PLFA life marker profiles only indicate abundant microbial life for the active layers and do not correlate with the past markers. Thus, the data suggest that in the permafrost sequence the past marker represent a paleo-signal (Stapel et al., 2016).

The results show that intervals with increased concentrations of past microbial markers often correspond to increased OM concentration (TOC) and quality (HI and higher aliphatic character). This can especially be observed in the Yedoma deposits of core sections MIS 3-1, 3-2 and in the upper part of 4-1 and 4-2 (Fig. 2c: LP1, LP2, LP4 and LP8). Thus, the data in this suggest (chapter 5.1) that the OM with higher quality seems to have stimulated an abundant microbial life during deposition in the past. Both bacterial and archaeal past markers are increased in these intervals (Figs 2e,f), whereas the archaeal markers suggest the presence of methanogenic communities and presumably greenhouse gas production in the past. Also if relating the past microbial markers to the TOC content of the sediments the same trends can be observed, indicating stimulated microbial communities at these depth intervals. Based on that, we suggest that the amount and quality of OM are responsible for the detected past bacterial and archaeal abundance in the Yedoma deposits (Fig. 2e,f).

In a future warmer climate, a slight increase in permafrost temperatures has not only an influence on the soil-moisture content but also on the abundance and diversity of the microbial community (Wagner et al., 2007). Thus, intervals of increased past-marker concentrations reveal time intervals of increased soil-moisture levels, which are thought to be linked to warmer surrounding temperatures (Stapel et al., 2016), especially during the MIS 3 and 4. This link was already observed by Bischoff et al. (2013). Here, comparable high concentrations of archaeol (up to approximately 40 ng/g sediment) and iso-GDGTs (up to approximately 20 ng/g sediment) were detected in Yedoma sediments, which were deposited during warmer and wetter environmental conditions in the Late Pleistocene. Similar average concentrations of iso-GDGT-0 + archaeol are also detected in this study for the MIS 3 and MIS 4 deposits. On the other hand, the relatively high concentrations of archaeol (up to approximately 80 ng/g sediment) in the Holocene sequence of the permafrost deposits from Kurungnakh Island (Bischoff et al., 2013) are not detected within the Holocene deposits of this study. This might be due a less aquatic influence on the Holocene deposits at Bol´shoy Lyakhovsky Island than on the permafrost deposits at Kurungnakh Island (Bischoff et al., 2013). Thus, these data reflect that increased soil moisture during deposition resulted in excellent living conditions for anaerobic bacteria (Weijers et al., 2006) and archaea (Wagner et al., 2007), especially during the MIS 3. According to Wetterich et al (2014), the MIS interstadial optimum occurred between 48 to 38 ka BP on Bol´shoy Lyakhovsky Island and is characterized by warmer temperature conditions with tundra environments with water-saturated active layers (Meyer et al., 2002; Hubberten et al., 2004; Andreev et al., 2011).

The Holocene (MIS 1) and Eemian (MIS 5e) deposits in this study were deposited in a different geomorphological environment than the Yedoma deposits referred to as thermokarst lakes (Andreev et al., 2004; 2009). According to Peterse et al. (2014), significantly higher concentrations of br-GDGTs (approximately 8800-47600 ng/g sediment) are found in flooded or water-filled permafrost depressions (e.g. thermokarst lake sediments) than in frozen Yedoma deposits (161 ng/g sediment). Nevertheless, the results of this study show similar high or even lower concentrations of br-GDGTs and iso-GDGTs (table S1) as well as archaeol in the deposits of MIS 1 and 5 compared to the LP glacial permafrost deposits, and an

average BIT index of 1 for all investigated deposits. Thus, although the geomorphological environment in the MIS 1 and 5 was described as thermokarst lakes (Andreev et a., 2004; 2009), the data indicate that these thermokarst lakes were probably only of minor durability and therefore did not affect significantly on the concentration of once living microorganisms (past marker).

**5.3 Microbial substrate potential for greenhouse gas generation**

Degradation of permafrost OM becoming bioavailable again in the course of ongoing permafrost thaw can finally lead to the production and release of greenhouse gases such as methane and carbon dioxide with their impact on the global climate cycle. To assess the potential of the OM from different depositional ages to provide substrates for the production of greenhouse gases, acetate is used as an appropriate substrate for microbial metabolism (Ivarson and Stevenson, 1964; Sørensen and Paul, 1971; Sansone and Martens, 1981; Balba and Nedwell, 1982). Acetate is the terminal electron acceptor for methanogens in cold-temperate environments (Chin and Conrad, 1995; Wagner and Pfeiffer, 1997), especially for acetoclastic methanogens (Thauer, 1998) and methanogenic archaea which are ubiquitous in anoxic environments and in permafrost sediments (Kobabe et al., 2004).

In this study two acetate pools are investigated: The free-acetate pool within the pore water, representing an easily accessible substrate source for microbial metabolism; and the bound-acetate fraction, which is still linked to the OM and constitutes a future substrate source upon degradation (Glombitza et al., 2009b). Overall, the concentrations of bound acetate (Fig. 2h) in the investigated samples correlate well with the amount of TOC and quality of OM. This indicates that larger reservoir pools for future microbial turnover exist at depths with increased TOC and HI (Fig. 2, Table 2). Deposits from the MIS 1, 3 and 4 possess increased bound acetate mean values, whereas the largest future substrate reservoir is located within MIS 3 (~ 48.9 mg/l), followed by MIS 4 (~ 33.26 mg/l) and MIS 1 (~30.05 mg/l) (Table 2). In contrast, the bound-acetate concentration in the Eemian (MIS 5e) deposits suggests a depleted and possibly already altered bound-substrate pool as the concentration is considerably lower (~ 9.98 mg/l) than in all other deposits.

The very low concentrations of free acetate and elevated concentrations of PLFA life markers detected in the investigated active layer samples suggest a higher microbial consumption of free acetate by an active microbial community (Lee et al., 2012; Knoblauch et al., 2013; Stapel et al., 2016). This activity is stimulated by e.g. warmer temperatures (thawing conditions) and the input of the fresh and old OM during the thawing period (Liebner et al., 2008). The thaw of permafrost due to global warming and the subsequent increase of active layer thickness result into the release of old organic carbon previously frozen in the permafrost, which is shown be particularly sensitive to temperature-induced microbial decomposition (Knorr et al., 2005; Davidson and Janssens, 2006), and therefore is considered as an important substrate source. In terms of the present study, the highest PLFA concentration was detected in the active layer above the MIS 3 deposits. This may reflect the high potential of the MIS 3 Yedoma deposits to serve as a substrate provider for a living microbial community upon thaw. In contrast, local environmental differences may also affect the PLFA concentration. For example, the core containing MIS 3 deposits was drilled on a stable tundra surface (core L14-02), while the other cores were

either drilled in a geomorphological dynamic terrace position experiencing thermo-erosion (Schirrmeister et al., 2011; Grosse et al., 2011) influenced by a seasonal input of sediment and water, or in a drained and refrozen Holocene thermokarst basin (core L14-05), which is characterized by lower ice contents and shallower active layers (Schwamborn and Wetterich, 2015). However, the increased PLFA concentrations in all active layers indicate to a certain extent that the permafrost deposits at least from MIS 3, 4 and 1 can serve as good substrate providers when thawed. For MIS 5e this could not be evaluated due to the lack of MIS 5e deposits with an active layer on top.

In contrast to the bound acetate concentrations, the free-acetate substrate pools and TOC content do not correlate well within each individual core (in all cores: $R^2 < 0.5$). The reason for this might be that free-acetate pool in permafrost pore waters is not only the result of acetate released from the OM, but also can be influenced by other factors e.g. lateral and vertical diffusion promoted by capillary pressure (Parlange, 1971), thawing and freezing processes as well as microbial production and consumption. However, positive relations between acetate (free and bound), TOC and HI are observed at several depth intervals mainly within the MIS 3 and 4 deposits (e.g. core L14-02 at 1.5 and 4.2 m, core L14-03 at 2.5 and 5.8 m). Here, the mean concentration (~ 93.6 and 82.1 mg/l) of free acetate is at least two to three times higher than that identified in the interglacial periods MIS 1 (24.1 mg/l) or MIS 5e (46.1 mg/l). The minor free-acetate pool in the Holocene deposits may be the result of the OM composition or of intense microbial consumption during OM deposition in the past as has been proposed for the modern active layer. The latter could be supported by deeper and prolonged thaw of active layers with increased active microbial acetate consumption (Xue et al., 2016) caused by the onset of the early Holocene, when warming resulted in unstable environmental conditions, especially during the Holocene Optimum (Andreev et al., 2004; Wagner et al., 2007; Wetterich et al., 2008). As such, the low concentrations of free acetate in the Eemian deposits may also be the result of increased microbial consumption due to warmer environmental conditions. Moreover, the Eemian is another period linked to permafrost thaw, which again likely altered the free-substrate concentrations in the sediment due to the lateral transport of water and/or sediment (Andreev et al., 2009). Based on the presented results obtained in this study it can be noted that sediments from the interglacial periods contain reduced amounts of free acetate in comparison to the LP glacial period investigated.

Changes in size of the free and bound acetate reservoirs are caused by either local microbial consumption or by lateral and vertical transport, probably also resulting in microbial acetate consumption on a larger scale (e.g. in a nearby located soil or lake). Although the free-acetate pool in the MIS 1 and MIS 5e deposits is similar low, the bound-acetate concentrations in the MIS 1 deposits imply that there exists a considerable future-substrate reservoir compared to the MIS 5e deposits (Table 2). On the other hand, in the investigated MIS 3 and 4 deposits both substrate pools (free and bound) are characterized by higher acetate concentrations compared to the deposits from the MIS 1 and 5 (Table 2). The deposits from the interstadial Yedoma period (MIS 3) in particular possess a larger substrate reservoir than found in other deposits linked to increased amount and quality of OM, as has been observed previously in a study of Buor Khaya Peninsula permafrost, 350 km SW from Bol'shoy Lyakhovsky Island (Stapel et al., 2016). Considering the sizeable thickness of the LP Yedoma deposit on Bol'shoy Lyakhovsky Island (20 m, Schennen et al., 2016) and across Siberia (10-60 m, Dutta et al., 2006), as well as its

extension across Russia (about 1028264 km$^2$, Grosse et al., 2013), it is hypothesised that a significant substrate pool may become accessible for future microbial greenhouse gas generation as permafrost thaws. On a broader perspective, ongoing warming in the Arctic will increase active layer thickness making substrates from deeper and older OM available for microbial decomposition, and enhancing the production and release of greenhouse gases (Schuur et al., 2008). Although the

5    complexity of potential positive (e.g. Schuur et al. (2008)) and negative (e.g. Flanagan and Syed (2011)) feedback loops between climate warming and the carbon cycle in the Arctic is still uncertain, the permafrost substrate potential for future greenhouse gas production plays a key role concerning shifts in the microbial community composition, vegetation, hydrogeology and soil thermal regime. The results of this study suggest that OM deposited during the interstadial and glacial periods contain a larger substrate potential than that found in the interglacial deposits and that parameters deduced from OM

10    pyrolysis seem to represent appropriate tools to reflect quality differences of the permafrost OM of different ages.

[revised manuscript text omitted]
 Epoch | Age MIS | Palaeoenvironment | OM quality | Substrate potential present | Substrate potential future | Cores |
|---|---|---|---|---|---|---|
| Holocene (interglacial) | 1 | - climatic warming[1]
- moisture increased thawing, thermokarst[2]
- unstable environmental conditions[2]
- dissected landscape influences by local hydrology[3] | - | - - | + | L14-05 |
| Late Pleistocene Glacial Period (glacial) — interstadial | 3 | - increased temperature and soil moisture[4] | + + | - | + + | L14-02 |
|  |  | - warm/moderate and dry climate[5] | + + | + + | + + |  |
|  |  | -optimum: warm and dry (48 to 38 ka BP)[5]
- warmer summers, open vegetation[2] | - | + | + |  |
| Late Pleistocene Glacial Period (glacial) — stadial | 4 | - cold and dry climate[5] | + | + | + | L14-03 |
|  |  | - harsh climate conditions[2] | - |  |  |  |
|  |  | - thin snow cover, low precipitation[2] | + |  |  |  |
| Eemian (interglacial) | 5e | - warmer climate, open-grass tundra similar to modern[2]
- permafrost thawing[2]
- optimum: 4-5 °C higher summer temperatures than modern, shrub tundra[2] | - - | - | - - | L14-04 |

**Table S1:** Concentration of identified branched glycerol dialkyl glycerol tetraethers (GDGTs) (GDGT-Ia, GDGT-Ib, GDGT-II, GDGT-III), isoprenoid GDGTs (GDGT-0, GDGT-1, GDGT-2, crenarchaeol), archaeol as well as the calculated branched and isoprenoid tetraether (BIT) index. For GDGT structures please refer to Schouten et al. [2013].

| Core name | mean depth [m] | Archaeol [ng/g Sed] | GDGT-0 [ng/g Sed] | GDGT-1 [ng/g Sed] | GDGT-2 [ng/g Sed] | Crenarchaeol [ng/g Sed] | GDGT-Ia [ng/g Sed] | GDGT-Ib [ng/g Sed] | GDGT-II [ng/g Sed] | GDGT-III [ng/g Sed] | BIT |
|---|---|---|---|---|---|---|---|---|---|---|---|
| L14-05 | 0.1 | 17.7 | 3.1 | 0.6 | 0.6 | 0.0 | 125.7 | 8.2 | 391.0 | 309.2 | 1.0 |
| | 0.5 | 8.9 | 1.0 | 0.2 | 0.1 | 0.0 | 2.6 | 0.5 | 11.5 | 15.2 | 1.0 |
| | 2.2 | 41.1 | 13.9 | 2.9 | 2.2 | 5.1 | 39.5 | 5.9 | 124.7 | 180.2 | 1.0 |
| | 2.7 | 7.5 | 2.1 | 0.3 | 0.2 | 0.5 | 2.2 | 0.5 | 7.5 | 10.1 | 1.0 |
| | 3.3 | 10.0 | 4.7 | 0.7 | 0.5 | 1.5 | 9.2 | 1.4 | 29.5 | 38.8 | 1.0 |
| | 4.7 | 22.0 | 12.6 | 2.5 | 2.7 | 5.2 | 21.7 | 4.0 | 77.8 | 121.7 | 1.0 |
| | 5.5 | 5.4 | 15.0 | 3.1 | 3.5 | 6.5 | 20.1 | 3.6 | 68.2 | 68.6 | 1.0 |
| | 7.1 | 81.1 | 133.4 | 5.5 | 4.6 | 0.0 | 269.5 | 11.0 | 871.8 | 918.1 | 1.0 |
| L14-02 | 0.2 | 3.9 | 0.8 | 0.0 | 0.0 | 0.5 | 12.9 | 0.5 | 24.2 | 15.4 | 1.0 |
| | 0.5 | 5.9 | 0.3 | 0.0 | 0.0 | 0.1 | 8.2 | 1.3 | 16.2 | 8.5 | 1.0 |
| | 1.6 | 12.4 | 3.8 | 1.1 | 0.8 | 1.0 | 92.1 | 14.4 | 278.8 | 206.0 | 1.0 |
| | 3.1 | 23.2 | 8.6 | 1.7 | 1.5 | 2.3 | 26.2 | 4.3 | 70.5 | 90.8 | 1.0 |
| | 4.2 | 10.2 | 0.9 | 0.5 | 0.7 | 1.4 | 17.1 | 3.1 | 68.2 | 59.3 | 1.0 |
| | 5.2 | 2.7 | 0.9 | 0.3 | 0.3 | 0.8 | 4.2 | 1.0 | 14.6 | 12.2 | 0.9 |
| | 7.0 | 1.4 | 0.5 | 0.2 | 0.2 | 0.4 | 0.8 | 0.1 | 1.6 | 1.2 | 0.9 |
| | 7.9 | 1.1 | 1.5 | 0.4 | 0.4 | 1.0 | 0.9 | 0.1 | 2.0 | 1.7 | 0.9 |
| | 9.0 | 1.5 | 1.9 | 0.5 | 0.5 | 0.3 | 2.6 | 0.3 | 5.3 | 5.4 | 0.9 |
| | 10.2 | 1.0 | 0.3 | 0.0 | 0.0 | 0.3 | 4.6 | 0.2 | 9.3 | 6.8 | 1.0 |
| L14-03 | 0.2 | 2.0 | 1.4 | 0.6 | 0.6 | 1.8 | 4.0 | 0.4 | 12.1 | 10.6 | 0.9 |
| | 0.6 | 4.6 | 0.4 | 0.0 | 0.0 | 0.1 | 18.5 | 0.7 | 42.1 | 26.0 | 1.0 |
| | 1.4 | 38.3 | 2.8 | 0.6 | 0.5 | 0.0 | 19.9 | 1.4 | 83.7 | 103.1 | 1.0 |
| | 2.0 | 28.8 | 2.8 | 0.6 | 0.6 | 0.2 | 21.6 | 1.2 | 60.5 | 65.7 | 1.0 |
| | 2.6 | 5.6 | 1.0 | 0.3 | 0.4 | 0.6 | 11.0 | 0.6 | 24.6 | 12.1 | 1.0 |
| | 3.4 | 3.0 | 1.1 | 0.3 | 0.0 | 0.4 | 3.9 | 0.3 | 8.7 | 7.5 | 1.0 |
| | 4.8 | 10.1 | 1.8 | 0.5 | 0.3 | 0.4 | 21.0 | 2.0 | 55.4 | 49.6 | 1.0 |
| | 5.8 | 4.2 | 1.3 | 0.4 | 0.0 | 0.6 | 10.3 | 1.3 | 29.4 | 33.1 | 1.0 |
| | 6.4 | 3.5 | 1.4 | 0.3 | 0.3 | 0.5 | 6.7 | 0.6 | 13.3 | 9.6 | 1.0 |
| | 7.3 | 2.8 | 1.3 | 0.3 | 0.3 | 0.6 | 5.9 | 0.5 | 11.8 | 8.9 | 1.0 |
| | 8.8 | 3.3 | 1.9 | 0.5 | 0.5 | 1.1 | 5.4 | 0.5 | 10.9 | 8.5 | 1.0 |
| | 10.1 | 2.6 | 1.0 | 0.2 | 0.2 | 0.4 | 2.8 | 0.3 | 6.2 | 5.1 | 1.0 |
| | 11.3 | 2.0 | 0.8 | 0.2 | 0.2 | 0.3 | 3.0 | 0.2 | 5.9 | 4.6 | 1.0 |
| | 12.4 | 3.7 | 1.5 | 0.0 | 0.0 | 0.5 | 7.4 | 0.0 | 12.6 | 8.1 | 1.0 |
| | 13.8 | 0.6 | 0.0 | 0.0 | 0.0 | 0.0 | 0.0 | 0.0 | 0.0 | 0.0 | 1.0 |
| L14-04 | 0.2 | 5.2 | 1.0 | 0.4 | 0.3 | 0.6 | 34.5 | 1.0 | 82.5 | 78.9 | 1.0 |
| | 2.4 | 7.0 | 1.0 | 0.4 | 0.3 | 0.1 | 55.6 | 3.9 | 133.7 | 100.8 | 1.0 |
| | 4.0 | 17.4 | 1.2 | 0.4 | 0.4 | 0.2 | 29.0 | 2.2 | 76.5 | 67.3 | 1.0 |
| | 5.3 | 15.8 | 1.5 | 0.2 | 0.2 | 0.4 | 2.8 | 0.5 | 9.8 | 21.1 | 1.0 |
| | 6.3 | 2.9 | 1.2 | 0.3 | 0.3 | 0.6 | 3.8 | 0.4 | 8.3 | 10.0 | 1.0 |
| | 7.2 | 2.5 | 1.0 | 0.2 | 0.2 | 0.4 | 3.9 | 0.4 | 8.4 | 9.1 | 1.0 |
| | 7.5 | 3.4 | 1.9 | 0.5 | 0.5 | 1.0 | 5.6 | 0.5 | 10.9 | 10.2 | 1.0 |

**"Substrate potential of last interglacial to Holocene permafrost organic matter for future microbial greenhouse gas production"**

By Janina G. Stapel et al.

5    We thank the three reviewers for their thoughtful and very constructive comments and suggestions on our manuscript, which helped to improve the clarity and the quality of the paper. Below, we addressed all comments (point-to-point) listed by the reviewers and changed our manuscript accordingly.

**Reviewer #1**

10   **To Introduction:**

*1) The link between the general introduction on permafrost thaw, its consequences, and the scope of this study can use some improvement. For example, Bol'shoy Lyakhovsky Island is suddenly mentioned on P2, L15. Later it appears to be the study site, but it needs some more context in the introduction. Similar for the 'Eemian deposits in this study...' (P2, L30). Also include acetate, and the difference between free and bound acetate.*

15   We improved the first part of the general introduction by rephrasing, condensing and/or combining sentences (P1, L29 - P2, L14). We now better outlined the relation between global warming, permafrost thawing, OM availability, greenhouse gas production and it feedback on global warming and permafrost thaw (P2, L8-14). We describe why Bol´shoy Lyakhovsky Island was selected as study site (P2, L30-31) and why Eemian deposits are of specific interest (P2, L31-34). We added some sentences on acetate as a quality indicator for greenhouse gas production and the difference between free and bound

20   acetate in the introduction (P3, L7-14). Finally, we improve the context of our scientific question (P3, L26-27).

**To Stratigraphy:**

*2) The composite core consists of different lithologies, i.e. lacustrine (MIS1and MIS5), floodplain deposits (MIS4), and also contains cryostructures. I miss a discussion on how different sources may influence OM parameters, lipid abundance and distribution, or acetate availability? For example, GDGTs in lacustrine (MIS1 and MIS5) and floodplain deposits (MIS4)*

25   *may have a mixed soil and aquatic origin. This may influence your results, especially when considering that an earlier study has shown that Siberian thermokarst lakes (probably comparable to the MIS5 deposits in this study?) can contain >200 times the concentration of branched GDGTs compared to Yedoma (Peterse et al., 2014, JGR-B). Similarly, isoGDGTs, archaeol, and PLFA concentrations may be influenced too.*

Our pyrolysis data indicate that the OM is mainly of terrestrial origin with some variations in the aliphatic character. Mainly

30   samples deposited during the late Pleistocene glacial period show this higher aliphatic character. We already discussed these differences in the aliphatic character as a result of different environmental conditions (higher soil moisture favorable for algae growth vs dry conditions, anaerobic soil conditions/reduced OM decomposition, cold vs warmer conditions etc.) rather than of different depositional settings (see chapter 5.1), which seems sometimes confusing. For instance, Holocene as well as Eemian samples were both deposited in a kind of thermokarst lakes, but they show significant differences in the amount and

35   quality of the OM. Thus, the influence of the depositional settings on the OM parameters applied here is not clear yet and

environmental impact on OM production and degradation might play a more important role. We added a few sentences on this into the manuscript (P11, L27-33).

In our study we cannot really see that the Holocene GDGTs or archaeol concentrations (deposited in thermokarst lakes) are significantly more abundant than in the LP glacial deposits. The data are in the same range in all time intervals. However, we added a sentence on this into the manuscript (P13, L33 – P12, L1). PLFAs are generally high in active layers and significantly lower in permafrost sequences. Thus, the PLFA signal is mainly determined by the thaw-front in summer and not so much from the depositional environment.

*3) Another way to check sources could be to calculate BIT index values (Hopmans et al., 2004, EPSL). Peterse et al., 2014 (JGR-B) found that Yedoma has a significantly lower BIT index than in soils. Similar changes should be visible throughout the composite core studied here.*

We calculated the BIT index (according to Hopmans et al., 2004), integrated it into the text (P6, L22-23 and P10, L16-17 and P14, L1) and provided the data in the table in the supplement (S1). The BIT index is usually 1 and only decrease in four MIS 3 samples (and in one AL MIS 4) to 0.91 and, therefore, it supports the overwhelming terrestrial soil character. Our study found that the BIT in the Yedoma deposits (MIS 3 and 4) did not much differ from the BIT values indicated in the active layers or in MIS 1 and 5e. The results presented by Peterse et al. (2014), indicate a much lower BIT value of 0.82 for Yedoma deposits from the Duvannyi Yar cliff. According to Strauss et al. (2012), the Yedoma at Duvanny Yar is of polygenetic origin formed on a floodplain of the Kolyma River, where alluvial and fluvial processes were the controlling processes. On the other hand, the Yedoma deposits on Bol´shoy Lyakhovsky Island were not influenced by dynamic fluvial processes during deposition (except for the lowermost 40 cm of core L14-03 where gravel in the sediments might indicate stronger fluvial influence during deposition). Furthermore, the investigated permafrost deposits are not floodplain deposits and therefore are less influenced by alluvial processes (Andreev et al., 2009). That explains why - other than in Peterse et al., 2014 - we observe a higher BIT index in the Yedoma deposits at Bol´shoy Lyakhovsky Island and do not see relevant differences in the BIT index by comparing the individual deposits.

**To Methods:**

*4) P4, L30: were samples decalcified prior to determining TOC? Otherwise total carbon is reported instead of TOC. P5, L5: was there any pre-treatment of the sample material prior to pyrolysis?*

Samples for TOC analyses were decalcified before the measurement. We added this information into the text (P5, L16). For both Rock-Eval pyrolysis and open-system pyrolysis sample material was freeze-dried and ground (as indicated in the text; P5, L19-20). There was no additional pre-treatment of the samples before the measurement. Free-biomolecules were thermally removed before pyrolysis of the macromolecular organic matrix (P5, L22-23).

**To Discussion:**

*5) P9, L3: How/in what figure/parameter is the contribution of aquatic OM reflected? Similar for the input of aquatic OM during MIS3 (P9, L32).*

Generally, based on our results the deposited OM in all investigated cores is of terrestrial origin. However, the results from Rock-Eval analysis (Hydrogen Index) indicate that the samples have varying proportion of a hydrogen rich component (Fig.

3a). Open-system pyrolysis shows that the samples with a higher HI reveal a higher aliphatic character (Fig. 3b). In contrast to aromatic compounds aliphatic compounds are assumed to be better degradable, thus, samples with higher HI (more aliphatic rich) seem to have a better OM quality in terms of biodegradability. This material also contains higher concentrations of potential substrates (e.g. acetate). A source of more aliphatic-rich OM could be algae, living in surface ponds or water saturated soils. Periods of increased soil-moisture for the Yedoma deposits were already indicated by Sher et al. (2005). Thus, higher HI point to a higher aliphatic character in permafrost OM which may reflect a higher proportion of aquatic OM. Figure 3a (after Eglinton et al., 1990) classifies the OM by different types and provides hints on the origin of the OM; whereby OM type I and II are characteristic for aquatic or marine OM. Especially, Yedoma samples (Fig. 3a: e.g. AL, LP1 and LP2) show a stronger proportion of hydrogen rich OM (type II organic matter). The higher aliphatic character seems to be related to increased soil-moisture favorable for algae growth during time of deposition or as it is observed today in the active layer during the thawing period. Figure 3b assigns the higher aliphatic character to the LP glacial samples. We added some further information on the parameters and figures indicating "aquatic OM" (P10, L20-23) to the discussion.

*6) Please clarify. Again P12, L15-16: what indicates the link to moist depositional conditions?* We deleted "moist depositional conditions" from the text here, since it was not important at this position. As described under point 5) above the interpretation on moist depositional environment derived from the increased aliphatic character and from literature information for the LP glacial period.

*7) P.9, L27: I can't follow this sentence. Check grammar/order of words. Also: if glacial conditions would slow down degradation, how does it influence its production?*

We rephrased this sentence as suggested.

*8) P.10, L15 and following: How can you be certain that GDGTs reflect past microbial biomass? In this study, only the core lipids are analyzed, whereas part of the GDGT pool may present as IPL, and thus derive from living biomass. Other studies have reported an IPL contribution of >30% to the GDGT pool in OC-rich soils (e.g. Peterse et al., 2011 Org Geochem).*

IPL with an ether bond moiety are less suitable to act as life markers due to their significantly increased stability (Logemann et al., 2011). Due to the similar structural moieties something similar can be suggested for the br-GDGTs and iso-GDGTs. In contrast IPLs with ester moieties are known to rapidly decrease after cell death and thus, since microbial communities usually consist of both bacteria and archaea, we use the PLFAs as life markers for the microbial community in general here (P12, L 10-15). Since life marker and past marker do not match we interpret that past marker signal is a paleo-signal.

*9) Furthermore, the manuscript refers to both microbial biomass and methanogens, of which the latter is of course more specific. Please go through the manuscript and check which level of specificity is relevant.*

Methanogens are more relevant when discussing acetate as a substrate for greenhouse gas production. We went through the manuscript to be more precise.

*10) Fig. 2: Is there a reason why archaeol is plotted together with isoGDGTs? They do not necessarily share the same source. Instead, it would be more logical to plot archaeol next to e.g. GDGT-0/cren, which are both indicators of methanogenesis. These data can then also be used to compare with the acetate data.*

5 The reason was to have one parameter for the past archaeal biomass and to safe some vertical space for the figure. Iso-GDGTs are mainly dominated by iso-GDGT-0 (see attached table S1) and both iso-GDGT-0 and archaeol are used for the presence of methanogens in permafrost regions (Bischoff et al., 2014; Pancost et al., 2011), although their ratio differs throughout different methanogenic genera (Koga and Mori, 2006). We added some sentences on this (P12, L29 - P13, L1). Thus, we would like to keep this combined archaeal parameter in Fig. 2, but provide all GDGT data in the supplement table

10 S1.

*11) L 20: Can you explain how exactly GDGT concentration data provides information on the activity of microbial biomass in the past?*

Sure the GDGTs are not a direct measure of microbial activity. However, we can assume that if microorganisms can be

15 found that they have been active. Thus, the presence of past markers concomitantly suggests the activity of these past microbial communities (when they formed the active layer community), (P12, L29-32).

*12) L23: Given that GDGT concentrations and TOC seem to covary (i.e. both higher in Yedoma?), it makes sense to normalize GDGT concentrations on TOC to distinguish between high GDGT concentrations due to high TOC content and*

20 *actual elevated microbial biomass. Do the trends and conclusions still hold?*

Yes they do. By normalizing the GDGT and archaeol concentrations to TOC, the same trends within the depth profiles of the cores are visible. Depths of increased TOC and OM quality (e.g. core L14-02 at about 1.5 m with a HI of 246 mg HC/ g TOC and TOC of 4.7 wt%) correspond with increased concentrations of br-GDGTs. On the opposite, depths of decreased TOC but increased HI (e.g. core L14-03 at about 1.4 m with a HI of 254 mg HC/ g TOC and TOC of 1.9 wt%) reveal

25 increased br-GDGT and archaeol concentrations indicating that the concentration of living microorganisms in the past not only depends on the amount of TOC but also on the quality of the OM (HI values). We added a sentence on this (P13, L9-11).

*13) The choice of 'excellent' as description for substrate (in abstract and P10, L31) seems odd. In my opinion something can*

30 *turn out to be, or has proven to be an excellent substrate, but you cannot select something as an excellent substrate if it isn't compared to anything else.*

Acetate is well-known to be intensively used as a substrate for microbial metabolism. Thus, based on this background knowledge acetate was classified as being "an excellent substrate" especially for methanogenesis (thus, compared to what we know from literature). However, we replaced "excellent" by "appropriate" (P1, L16 and P14, L9).

*14) The discussion on microbial activity in the active layer leads to the conclusion that MIS1, 3, 4 provide most substrate upon thaw. However, it is important to mention that active layer sediments included in this study, and that there are no active layer samples from MIS2 and MIS5 permafrost are included. There should be a few words on how representative this sample selection is.*

Reading this comment, we agree that this sentence can be interpreted in this way. However, what we wanted to say is that there are already "natural" examples that old material can stimulate microbial life again. We are aware that we cannot say anything about the potential of MIS 5e material, since there was no active layer for the MIS 5e available in the field. We added the following sentences: (P7, L3-4) "… Due to the stratigraphic settings at the study site on Bol´shoy Lyakhovsky Island, active layers containing OM from MIS 2 and MIS 5e could not be obtained in the field." and another sentence on this on P15, L5-6.

*15) I furthermore miss the link between substrate availability and OM quality or composition and microbial biomarker abundance. I also miss a comparison with data from the literature.*

*I have already mentioned Peterse et al 2014, JGR-B (Branched glycerol dialkyl glycerol tetraether in Arctic lake sediments: sources and implications for paleothermometry at high latitudes), but there are more biomarker papers on Siberian (or Arctic in general) permafrost soils, and there must be on microbial community composition and OM properties, too. To name a few: Bischoff et al 2013, GBC. Response of methanogenic archaea to Late Pleistocene and Holocene climate changes in the Siberian Arctic. Knoblauch et al., 2013, GCB, Predicting long-term carbon mineralization and trace gas production from thawing permafrost of Northeast Siberia. Blaud et al., 2015, Res in Microb. Arctic soil microbial diversity in a changing world. And papers citing those.*

We now wrote a paragraph on the relation of OM quality or composition and microbial abundance (P13, L4-8) and a sentence on OM quality and substrate availability (P14, L16-18). Furthermore we added a paragraph were we compare our results with literature data (P13, L16- P14, L4).

*16) P11, L31: Can you support the drawn similarities between TOC and free acetate with statistics? Is their relation stronger during MIS3 and 4 compared to during MIS1 and 5?*

The statistical similarity of the bound acetate fraction with TOC (average $R^2$ for all cores= 0.7-0.8) is much better than for the free acetate (for all cores $R^2 < 0.5$), which was already observed in Stapel et al., 2016. We think that the free acetate might be much more influenced by external factors (adsorption, release, consumption (e.g. in active layer), diffusion and maybe transport). The factors are not clear yet. We wrote a sentence about the statistics between free acetate and TOC (P15, L8-11).

*17) P12, L2-5: check the grammar of this sentence, I can't follow the reasoning. I think 'favoured' needs to be replaced with 'a result of'....caused by the onset of the Holocene, when a warming climate caused unstable environmental conditions ....*

We corrected the sentence according to the reviewer's suggestions.

*18) P12, L5-9: This sentence also seems to lack punctuation marks, verbs, logical order of words. Please check.*

We rewrote and splitted this sentence.

5    **Specific comments:**

*P1, Line 15: delete excellent*; deleted.

*P3, L5: abbreviation (PLFA) seems out of place* here; deleted.

*P4, L30: replace grounding by grinding*; done (P5, L16).

*P5, L2: past tense of grind is ground, not grounded;* replaced (P5, L20).

10    *P5, L9, replace were by was;* done (P6, L2).

*P5, L15: replace grounded by ground*; replaced (P6, L8).

*P.5, L17: polarity fractions, or fractions of increasing polarity*; done (P6, L10).

*P5, L20: what does PL stand for?*; replaced by polar lipid fraction (P6, L13).

*P10, L23: increased*; sentence was deleted.

15    *P11, L31: similarities ....are...*; sentence was rewrote (P15, L11-12).

*P11, L34:...the OM composition...*; done (P15, L15).

*P12, L5: reason;* sentences was rephrased (P15, L19-22).

*P12, L10: replace 'implied' by 'caused'*; done (P15, L27).

*P12, L16: 'moist' or 'more moist'. 'Moister' does not exist*; sentence was rephrased (P15, L30-32).

20    *P12, L20:...gas generation is accessible within permafrost.*; replaced by might become accessible with increasing permafrost thaw (P16, L1-2).

*P12, L20: delete 'an'*; done.

**_Terminology:_**

*P.11, L20: what is old freeze-look permafrost*; deleted due to rephrasing of the paragraph.

25    *P. 11, L 27: what is a thermos terrace?* It has to be "thermo terrace" which is also known as "thermo-erosional valley" (Schirrmeister et al., 2011). We replaced it and added the reference to the text (P15, L1).

**Reviewer #2**

**General comments**

*1) I recommend to be careful with conclusions on organic matter degradability based on organic matter chemistry (see e.g. Schmidt et al., 2011, Nature).*

5   In our understanding Schmidt et al. 2011 claimed that the persistence of soil OM primarily not depend on the molecular properties of the OM itself but on the physicochemical and biological properties of the surrounding environment. In permafrost regions the main factor controlling the ecosystem is the cold temperature, which causes reduced OM decomposition and therefore an increased OM accumulation. Thus, permafrost can contain OM which is highly vulnerable to microbial degradation as it was not fully degraded due to the low environmental temperature conditions (short periods of
10  microbial activity during summer season). Although microbial degradation might be impeded by the conditions in permafrost areas, microorganisms will not degrade OM randomly upon thaw. Specific structural moieties are better degradable (less energy demanding) than others for the microbial community involved in the degradation processes (aromatics vs. aliphatics). These structural differences in the OM depend on the OM source and the level of microbial degradation before freeze-locking. In the current paper we addressed whether we can (i) trace these structural differences in
15  the deposited OM by pyrolysis and (ii) characterize a stored potential substrate pool for microbial turnover. Thus, we apply these two parameters as quality indicators for the stored OM to act as a substrate provider for microbial degradation upon permafrost thaw. The detected microbial abundance and activity in the active layers on top of the permafrost (Fig. 2) as well as incubation experiments of old permafrost material indicate that old OM indeed can act as a substrate provider. Thus, our approach is more to discuss the OM quality in terms of potential substrate provision as outlined in the final conclusions.

*2) I do not understand the importance of past and present microbial biomass for this research question. There is no doubt about the presence of a living, active microbial decomposer community in the active layer, and the data are not interpreted to more detail. I would also like to point out that I am not aware of studies testing how fast PLFAs are degraded in continuously frozen soils, and that we therefore do not know for sure if PLFAs in permafrost really represent the living*
25  *microbial community.*

  In order to investigate whether the freeze-locked OM stimulated already a microbial community during its deposition in the past, biomarkers for past microbial communities were examined. Since past microbial biomarkers could also be a product of microbial degradation by the presently living microbial community, the Bol´shoy Lyakhovsky samples were also screened with regard to microbial life markers to compare both biomarker records (P12, L7-10). PLFA life marker profiles indicate
30  abundant microbial life in the active layers compared to the permafrost deposits and do not correlate with the past markers profiles. Thus, the data suggest that in the permafrost sequence the past marker represent a paleo-signal (P13, L1-3).

  The significant difference in PLFA concentration between the active layer and the permafrost deposits suggest that PLFAs also in permafrost environments can be used as a life marker. Other studies have shown that microorganisms can survive in deep permafrost sediments (e.g. Gilichinsky and Wagener, 1995; Rivkina et al., 2004; Bischoff et al., 2013) and incubation
35  experiments have measured microbial produced $CO_2$ after thawing permafrost sediments (e.g. Knoblauch et al., 2013; Walz et al., 2017), indicating living microbial cells in frozen permafrost sediments, which can be "re-activated" with permafrost thaw and then consume the stored OM (We added this to the text: P12, L19-25). Thus, the low numbers of PLFA in the permafrost seems indeed represent living microorganisms in the permafrost sequence. For comparison also see Stapel et al., 2016. Overall, PLFA life marker profiles only indicate abundant microbial life for the active layers and do not correlate to

the past markers in the permafrost sequence. Thus, the data suggest that in the permafrost sequence the past marker represent a paleo-signal (see chapter 5.2).

*3) As the authors themselves acknowledge (page 11, lines 12-14), acetate concentrations say little about acetate availability as this depends on the production rates of acetate from organic matter. I am therefore not sure about the value of this parameter in this context.*

This was misunderstood; free acetate is an easily consumed substrate for microbial metabolism. Thus, free acetate concentration is a good tool to assess the potential of the OM to provide substrates for microbial turnover. The same is valid for the bound acetate concentrations indicating the future potential of the OM for substrates release upon degradation.

In the respective sentence we wanted to explain why the free acetate concentration might be more different from the TOC values than the bound acetate concentration. We rephrased this part and shifted it to create a better understandable context (P15, 8-11).

*4) I would appreciate more details on the applied methods that are also not contained in the cited previous publication (Stapel et al., 2016). In particular, what compounds were detected with pyrolysis-GC-MS and how were they evaluated to generate Figure 3?*

As external standard we used *n*-butane and the pyrolysate products were identified with the aid of reference chromatograms. The peak areas were integrated and calculated using the AGILENT ChemStation software. For the Eglinton-diagram (Eglinton et al,. 1990) o-xylene, 2,3-dimethylthiophene and *n*-nonene and for the Horsfield-diagram Horsfield et al. (1989) $C_1$-$C_5$ gases, $C_6$-$C_{14}$-*n*-alkanes and *n*-alkenes as well as $C_{15}$ and longer *n*-alkanes and n-alkenes were integrated (P5, L24-29).

*5) What PLFAs were detected and used to quantify total PLFAs? Were only bacterial or also fungal markers considered?*

Further information on the quantification of the PLFAs was added in the sub-chapter "microbial lipid biomarker analysis". Bacterial PLFAs from 14:0 to 21:0 with corresponding *iso*- and *anteiso*-FAs as well as br- and unsaturated-FAs were considered (P6, L13-15).

*6) The authors further present some interesting correlations between individual parameters, and the statistical approach should be described in the methods section. The underlying correlation matrix could also be presented in a separate table in the manuscript to give the reader a better overview.*

We added an additional sub-chapter (3.4 Statistical approaches: P6, L24-27) to describe how the statistical parameters were calculated. All shown data will be available on *https://www.pangaea.de* for free download; therefore we decided not to include extra tables to present the underlying correlation matrix for every single calculation.

*7) The authors use European terminology to describe the glacial cycles that is technically not correct for Siberia. I do not object in general since the European terms are well known and the authors also use Marine Isotope Stages to identify these periods, but I suggest to add at least a comment on the terminology to the text.*

We totally agree to also add the Russian terminology into the text and into table 1. The Russian terms were integrated in the chapters "study area and material" (P4, L12) and "core descriptions" (P4, L26 and P5, L9 an L13).

*8) If I understood correctly, the authors imply that old organic matter from deep, continuously frozen permafrost deposits might move upwards into the active layer and stimulate microbial activity there (e.g., page 10, lines 10-13; page 11, lines 20-24; Conclusions). While it is correct that an influx of additional organic carbon can stimulate microbial activity in soils ("priming effect"), I do not see by what mechanism organic compounds could move upwards from frozen into non-frozen parts of the soil.*

This was also a misunderstanding. What is meant here is that old OM gets into the active layer with increasing active layer depth (deepening of thaw front) and therefore is incorporated into the active microbial carbon-cycle again during the thawing period. We rewrote the respective paragraphs making clear that the OM becomes available again due to permafrost thaw. (P14, L 26-28).

*9) The manuscript contains many grammatical mistakes and would overall profit from some language polishing (some sentences are very long and difficult to understand).*

We revise the manuscript thoroughly.

*10) As my last general comment, I want to mention that I think the authors do a good job in keeping overview of the different depositional ages across the different cores. This is not an easy task.*

We thank the reviewer for the positive feedback.

**Technical corrections**

*Page 1, line 19: What do you mean with "representing at least a future substrate potential upon release during OM degradation"?*

The investigated bound acetate is a not directly available substrate pool and can only be available upon liberation from the OM via geochemical or microbial alteration of the OM. We added "(present substrate pool)" and "(future substrate pool upon degradation)" into the abstract (P1, L15-16). Additionally, to improve the understanding of the idea of this parameter sentences were added into the introduction (P3, L7-14).

*Page 1, lines 28-30: I find it difficult to follow the causalities here, please rephrase*; - paragraph was rephrased (P1, L29 - P2, L14).

*Page 2, line 1: The upper 0-3 m*; - corrected (P1, L30).

*Page 2, lines 2-4: I suppose you mean that the freeze-locked OM might thaw and/or be converted into CO2 or CH4, potentially inducing a positive feedback to global warming. Since the consequences of permafrost thaw are described in more detail later in the paragraph anyway, I suggest deleting this sentence. If you want to keep the sentence, please be more concrete.*

We removed this sentence part, but explained the climate carbon feedback later in P2 L10-14.

*Page 2, line 6: What do you mean with "drastic changes in the ecosystem"*; - replaced by "changes in vegetation" (P2, L3-4).

*Page 2, lines 8-10: The sentence is not clear to me. The term "re-mobilization" usually refers to the export of previously frozen OM or nutrients into aquatic systems (e.g., in the cited Vonk et al. reference), but this is not the cause for increasing decomposition rates or accessibility of OM for microbial degradation. Rather, permafrost thaw leads to increased microbial activity and consequently increased decomposition rates, as well as to increased export into aquatic systems.* We were not aware that the term "re-mobilization" can only be used in the way the reviewer claimed. Here, we wanted to say that freeze-locked OM and nutrients are again taking part in the carbon cycling upon permafrost thaw. To avoid confusion we replaced this term by "Thawing of permafrost promotes the accessibility of the formally preserved OM and nutrients for microbial turnover again, which results in increased microbial activity and consequently in increased OM decomposition rates (Dutta et al., 2006; Schmidt et al., 2011)."(P2, L8-10)

*Page 2, line 10: "preserved"*; - corrected (P2, L8).

*Page 2, line 10: Delete "the"*; - deleted.

*Page 2, lines 12-14: This sentence mainly repeats the statements in the sentences before, I suggest to include the experimental information there.*

We rephrased the paragraph to avoid reparations (P1, L29 - P2, L14).

*Page 2, line 28: Delete "of"*; - deleted.

*Page 3, lines 9-10: Reference missing.* We added the references "Weijers et al., 2006; Schouten et al., 2013" and "Pancost et al., 2001; Koga and Morii" (P3, L23 and L24-25).

*Page 3, lines 12-13: What do you mean with "feedback effects on permafrost deposits"?* We rephrased the whole sentence: "… the contribution of thawing permafrost deposits of different ages to the carbon-climate cycle is still an open question" (P3, L26-27).

10  *Page 3, lines 15-16: I don't understand, please rephrase*; - done (P3, L30-31).

*Page 4, line 30: "Grinding" instead of "grounding". See also page 5, lines 2 and 15*; - replaced (P5, L16, L20 and P6, L8).

*Page 4, lines 30-31: Were samples acidified before TOC analysis?*

15  Yes they were. We added this information to the text (P5, L16).

*Page 5, line 9: Change "were" to "was"*; - done (P6, L2).

*Page 5, line 23: Change to either "using medium-pressure liquid chromatography" or "using a medium-pressure liquid*
20  *chromatograph"*; - done (P6, L18).

*Page 6, line 1: I suggest using "includes" instead of "consists"*; - replaced (P7, L2).

*Page 6, line 10: Change "marker" to "markers"*; - corrected (P7, L12).

*Page 6, line 19: Do the PLFA and TOC concentrations also correlate?*

There are some similarities between PLFA profile and TOC (especially MIS 1, MIS 5e) but they show no overall correlation. We added this information to the text (P7, L22-23).

*Page 6, line 23: Change "is" to "are"*; - replaced (P7, L27).

*Page 6, line 23: "Significantly": Was this statistically tested, and if yes, how? Otherwise, I would use another word.* We replaced "significantly" by "much higher" here (P7, L27).

*Page 7, line 2: What do you mean with a partial correlation between past bacterial and archaeal markers?*

10  Past bacterial and archaeal biomarker show higher abundances in the same depth interval, but the curves do not directly correlate (P8, L6-7).

*Page 7, line 6: Should be plural, "concentrations"*; - replaced (P8, L12).

15  *Page 7, line 7: "Rise"*; - corrected (P8, L12).

*Page 7, line 23: Do p-value and R2 refer to correlations of both bacteria and archaea with TOC?* The p-value and $R^2$ referred only to the correlation of the br-GDGTs with TOC. We rephrased and extended the sentence to make that more clear. (P8 , L28-29)

*Page 8, line 8: "Scatters"*; - corrected (P9, L16).

*Page 8, line 23: I suggest using "weakest" instead of "smallest". Same for line 26 ("strongest" instead of "highest")*; - replaced (P9, L32 and P10, L2).

*Page 8, line 30: What do you mean with "assigned"? How does OM degradability depend on the amount of OM? And how do you distinguish OM composition and OM quality?*

We added a paragraph at the beginning of each discussion chapter to introduce into the following discussion (chapter 5.1, 5.2, 5.3). Thus, the respective part was rephrased becoming part of the starting paragraph: "When permafrost thaws,

formerly freeze-locked OM becomes bioavailable again. In this context, it is of utmost interest for the assessment of the impact of this OM on future climate evolution not only to determine the abundance but also to learn more about the quality of the OM with regard to its potential degradability. For instance, terrestrial OM (more aromatic rich) is considered to be more recalcitrant than aquatic OM (more aliphatic rich) (Hedges et al., 2000). Thus, the quality of the OM in permafrost deposits is determined by its source and, therefore, structural composition as well as by its alteration due to early diagenetic degradation processes during its deposition in the past (White, 2013). In the following chapter we apply pyrolysis techniques (Rock-Eval pyrolysis and open-system pyrolysis GC-FID) of the OM to get a deeper insight into the structural composition and to establish a new tool for OM quality assessment as introduced in Stapel et al. (2016)." (P10, L6-14).

*Page 9, line 10: The main mechanism by which TOC/TN decreases during OM decomposition is the faster loss of C than N due to microbial respiration.*

We agree with the reviewer and added this to the sentence (P10, L28-31).

*Page 9, line 13: Change to "...while OM with low HI contains..."*; - corrected (P11, L1).

*Page 9, line 19: Please add a reference here;-* we added "Andreev et al., 2009 " as reference (P11, L9).

*Page 9, lines 27-29: Are you referring to the last interstadial here? I also noticed that Table 2 suggests both a dry climate and moist soils during that period, this seems rather strange. There is also something wrong with the grammar in the first part of the sentence.*

In this sentence we are referring to the Late Pleistocene (LP) glacial period in general, which is characterized by cold-climate conditions with anaerobic soil conditions. These slowed down OM decomposition rates and increased the accumulation of OM during this period. In this study, we present permafrost deposits from a stadial and interstadial period within the LP glacial. Subordinated different climate conditions (e.g. dry and wet) during the overall cold-climate conditions during the LP glacial period characterize this interstadial and stadial period. We rephrased the sentence (P11, L17-20).

*Page 9, lines 30-31: "Moisture increased depositional settings": please rephrase; -* done (P11, L22).

*Page 9, line 33: Change "minor" to "lower"*; - replaced (P11, L24).

*Page 10, line 11: The Fontaine paper is not about permafrost; -* we removed this reference and replaced it by Knoblauch et al., 2013. The sentence was rewritten (P12, L19-21).

*Page 10, lines 11-12: I do not understand. + Page 10, line 12: Which is not surprising considering that the active layer is seasonally thawed and the deeper permafrost is continuously frozen.*

This was deleted here and shifted now to chapter 5.3, where it was rephrased to: "However, the increased PLFA concentrations in all active layers indicate to a certain extent that the permafrost deposits at least from MIS 3, 4 and 1 can serve as good substrate providers when thawed. For MIS 5e this could not be evaluated due to the lack of MIS 5e deposits with an active layer on top." (P15, L4-6)

*Page 10, lines 13-14: PLFAs inform about microbial biomass, not activity. + Page 10, line 20: Please add a reference for GDGTs as indicators of microbial activity. Also, the data presented show TOC concentrations, not accumulation rates.*

That is true, but high abundance of PLFAs in a surface near seasonally thawed deposit highly suggest that the living microbes are somehow active, especially if comparing with the PLFA data from the deeper permafrost deposits. We re-wrote the whole paragraph: "According to Knoblauch et al. (2013), permafrost surface layers contain both fresh organic material and old OM (within the permafrost), which can stimulate microbial activity. The high concentrations of PLFAs in the active layer suggest that not only the abundance of microbial life is increased in this layer, but also the microbial activity. (P12, L18-21).

Also GDGTs provide no direct measure on the activity, but the high abundance of these past markers suggest that they have been active in the past during time of deposition. We rephrased this sentence to: "…and indirectly their abundance might say something about their activity during time of deposition." (P12, L29-32.)

*Page 10, line 23: "increased"*; - we rewrote the sentence (P13, L7-9).

*Page 10, lines 23-25: The observed coincidence of high GDGT and OM concentrations does not necessarily imply certain environmental conditions. Microbial biomass is often correlated with OM concentrations since most microorganisms use OM as substrates.*

By normalizing the GDGT and archaeol concentrations to TOC, the same trends within the depth profiles of the cores are visible (P13, L9-11). Depths of increased TOC and OM quality (e.g. core L14-02 at about 1.5 m with a HI of 246 mg HC/ g TOC and TOC of 4.7 wt%) correlate with increased concentrations of br-GDGTs. On the opposite, depths of decreased TOC but increased HI (e.g. core L14-03 at about 1.4 m with a HI of 254 mg HC/ g TOC and TOC of 1.9 wt%) reveal increased br-GDGT and archaeol concentrations indicating that the concentrations of living microorganisms in the past not only depend on the amount of TOC but also on the quality of the OM (HI values). As the HI indicates a higher aliphatic character representing increased soil-moister conditions (Stapel et al., 2016), conclusions on changes in the past soil moisture can be derived. We rewrote the paragraph about the past microbial markers.

*Page 10, line 32: I suppose you mean microbial metabolism, not turnover*; - replaced (P14, L9).

*Page 11, line 15: What do you mean with "soil biogeochemistry composition"?* We rephrased this sentence to: "The reason for this might be that free acetate pool in permafrost pore waters is not only the result of acetate released from the OM, but also can be influenced by other factors e.g. lateral and vertical diffusion promoted by capillary pressure (Parlange, 1971), thawing and freezing processes as well as microbial production and consumption." (P15, L8-11).

*Page 11, lines 16-17: Speculation.*

The sentence was rephrased to: "The very low concentrations of free acetate and elevated concentrations of PLFA life markers detected in the investigated active layer samples suggest a higher microbial consumption of free acetate by an active microbial community (Lee et al., 2012; Knoblauch et al., 2013; Stapel et al., 2016)." (P14, L23-25).

*Page 11, line 19: It is true that input of fresh OM by plants might additionally stimulate the microbial community in active layers, but I would expect that the main reason for the higher microbial biomass is the fact that the active layer is thawed in summer (i.e., provides liquid water).*

Sure the reason for the activity in the surface layer is the fact that this part is thawing during summer, but we think also the kind of OM is important for microbial degradation (what is more easily degradable for a microorganisms). We rephrased the paragraph and considered the reviewer´s comment (P14, L25-26).

*Page 11, lines 20-21: What do you mean with the incorporation of frozen permafrost carbon into the active layer?*

Due to increasing active layer thickness, more old freeze-locked permafrost carbon is integrated in the microbial degradation processes in the active layer during the thawing period. We extended the sentence (P14, L26-30).

*I also think it should be "freeze-locked";* - deleted due to rephrasing of the paragraph.

*Page 11, line 25: Change to "...differences also affect the PLFA concentration";* - replaced (P15, L32).

*Page 11, line 26: "thermos"?*

It has to be "thermo terrace" which is also known as "thermo-erosional valley" (Schirrmeister et al., 2011). We replaced it and added the reference to the text (P15, L1).

*Page 11, line 27: Change to "continuous"*; - changed to "seasonal" (P15, L2).

*Page 11, line 31: Change "is" to "are"*; - sentence was deleted due to rephrasing of the paragraph (P15, L7-24).

*Page 11, line 34: What do you mean with "OM composition deposited"?*; - - sentence was deleted due to rephrasing of the paragraph (P15, L7-24).

*Page 12, line 1: Microbial consumption in the active layer is an ongoing process and not restricted to the "time of*
10 *deposition".*

This was a misunderstanding. It was meant when the Holocene deposits were part of the active layer in the past. We rephrased this to: "The minor free-acetate pool in the Holocene deposits may be the result of the OM composition or of intense microbial consumption during OM deposition in the past as has been proposed for the modern active layer (P15, L14-16).

*Page 12, line 2: What do you mean with "stronger pronounced"? I'm afraid I cannot follow the entire sentence.*; - we rephrased the sentence and replaced "stronger pronounced" by "deeper and longer thawed" (P15, L16).

*Page 12, line 5: Singular: "reason".*

20 We rephrased and splitted this sentence (P15, L8).

*Page 12, line 10: I am not sure "implied" is the right word*; - replaced by "caused" (P15, L25).

*Page 12, line 11: What do you mean with "microbial acetate consumption on a regional scale"?* Permafrost thaw enables
25 lateral and vertical transportations of water, OM and sediment which can redistribute substrates. These redistributed substrates will probably be microbially consumed on/in another place (e.g. in a near located soil or lake). In other words, on a more regional scale (this study site only represents a small spot in the Siberian Arctic) lateral and vertical transportation of substrates probably will end up in microbial consumption. We added further information into the text (P15, L26-27).

*Page 12, line 20: I suggest changing to "...a significant substrate pool for future microbial greenhouse gas generation might become accessible within thawing permafrost"*; - rephrased (P16, L1-2).

*Page 12, lines 22-26: I think this sentence is incomplete. I certainly cannot follow the grammar.* We rewrote the sentences (P16, L4-8).

*Page 12, line 27: "appears"*; - corrected (P16, L4).

*Page 12, line 32: "a strong impact"*; - corrected (P16, L6).

*Page 13, line 8: "guided"*; - corrected (P16, L24).

*Page 27, line 6: "Eglinton"*; - corrected (P31, L6).

*Page 29: How were the plus and minus signs assigned?*

To visualize the OM quality and the substrate potential a relative scaling based on the results of this study was applied (P33, L8).

**Reviewer #3**

**Major comments**

*1) Methods section – there is no explicit mention of the technique used to measure acetates, or it's not mentioned clearly. Since this is the main purpose of the paper it should be obvious what has been done to measure the acetate compounds.*

5    The method is described in chapter "3.2 Low molecular weight organic acids (LMWOAs) analyses" (P6, L1-6).

*2) A lot of the molecular concentrations are reported as per gram sediment, but this leads to depth profiles that mostly correlate with OC content. Reporting molecular concentrations per gram carbon may lead to more interesting comparisons along and between cores.*

10   Relation to gTOC provides information on the abundance of a parameter relative to TOC. As outlined above (reviewer 1, comment 12) there was not much change in the observed trends (P13, L9-11). Since we want to show the total abundance of microorganisms and substrates we would like to keep the gSed relation.

*3) Permafrost soils and Yedoma can have very different biomarker compositions. For example, GDGTs are being used as*
15   *microbial biomarkers, but Sparkes et al., Biogeosciences, 2015 showed that GDGT concentrations are low in Yedoma sediments. Bacteriohopanepolyols may be better tracers of microbial activity in this region (see for example Bischoff et al., Biogeosciences, 2016). When linking timescales to substrate potential, the different sediment types within each core need to be shown in figures and discussed as well, since there could be a combination of climatic and sedimentological controls on OM quality and substrate potential.*

20   Well, bacteriohopanepolyols have not been investigated. The appeal of using GDGTs and archaeol is that you get information on bacteria and archaea, which is not the case for bacteriohopanepolyols. Also we are aware that GDGTs and archaeol do not represent all microorganisms, we at least get some insights of bacterial and archaeal variations over time.

As outlined in our response to comment 2 (reviewer 1) the depositional effect on the OM quality is not clear yet. However, we added this into the discussion (P11, L26-34).

*4) P9 Line 1 – it is asserted that the permafrost deposits are dominated by terrestrial OM. Since GDGTs were measured, the BIT index could be used to confirm this.*

See comment 3 reviewer 1 above.

30   *5) P1 Line 23 – The GDGTs seem to correlate with TOC in the core sections referenced here. Relative increases in these molecules may support increased bacterial productivity, but if the biomarkers are changing with TOC then it may just represent variations in preservation. Once more, other markers for microbial activity would add value to the study.*

GDGTs are already degraded biomarkers from intact polar lipids. The core lipids (GDGTs) are regarded as relatively stable. Thus, we suggest that they represent the past abundance of microbial communities, although preservation aspects cannot fully be ruled out.

5 **Minor comments**

*P5 Line 24 - Was an internal GDGT standard used?*

An external archaeol standard is used for quantification. We added this to the method chapter. (P6, L21-22)

*P5 Line 24 – The GDGT biomarker molecules being measured are not defined.*

10 We added a supplement table where all GDGTs are listed (table S1), (P6, L23).

*P9 Line 15 – Absence of a particular biomarker does not necessarily mean that it has decomposed; it may never have been present.*

This sentence was rephrased to: "Thus, the low TOC and TOC/TN values (< 5), in addition to the low HI in the Eemian
15 samples (MIS 5e, Table 1) may point to less favorable conditions for OM accumulation and/or an increased degree of OM decomposition and therefore to a reduced OM quality." (P11, L2-4)

*P10 Line 10-15. This section is hard to understand, rephrasing may help*

This was rephrased and shifted (see comments above) to: "However, the increased PLFA concentrations in all active layers
20 indicate to a certain extent that the permafrost deposits at least from MIS 3, 4 and 1 can serve as good substrate providers when thawed. For MIS 5e this could not be evaluated due to the lack of MIS 5e deposits with an active layer on top." (P15, L4-6)

**Typos**

*P2 Line 10 – preserved*; - changed (P2, L8).

*P2 Line 11 – gases*; - changed (P2, L12).

*P3 Line 32 – no comma needed after intervals*; deleted (P4, L17).

*P4 Line 30, and elsewhere – grinding rather than grounding; ground rather than grounded*; corrected (P5, L16 and 20, and P6, L8).

*P6 Line 21 – sentence does not make sense, especially "whereas"*; sentence was deleted.

*P6 Line 23 – is should be are*; done (P7, L27).

10   *P8 Line 4 – commas required after 'are' and 'sediment'*; we added a comma after "sediment" (P9, L12).

*P9 Line 23 – "nonene" rather than "nonen"?*; - deleted here, but corrected at P31, L8.

*P9 Line 27 – 'have' does not make sense*; sentence was rephrased (P11, L26-29).

*P11 Line 20 – freeze-locked*; - deleted due to rephrasing of the paragraph.

*P12 Line 11 – comma required after 'transportation'*; comma added (P15, L26).

20   *Figure 3 caption – Eglinton not Eglington*; - corrected (P31, L6).

---

## Author Response (AR2)

**Journal:** *bg-2017-89*

**Author:** *Janina Stapel et al.*

**Title:** *Substrate potential of Eemian to Holocene permafrost organic matter for future microbial greenhouse gas production*

*Revised parts in the manuscript are highlighted by a yellow background!*

**To the comments of reviewer 2**

**Based on the comments of 3 reviewers, Stapel et al have revised their manuscript on substrate potential of permafrost OM for greenhouse gas production. The current version the manuscript has improved compared to the first version, but still needs further improvement in order to be acceptable for publication in Biogeosciences. Next to the interpretation (see detailed comments below), also the grammar will need a more thorough check. The long sentences, grammatical mistakes, and needless repetitions distract from bringing the main message across.**

*We separated and simplified long sentences, removed needless repetitions and streamlined the overall paper. Due to the streamlining process parts of the manuscript were substantially revised or removed.*

**Based on reviewers comments (in random order):**

**- OM quality**

**There is no clear definition of OM 'quality 'in the introduction, and to me it is not clear what is exactly meant by 'quality', and what makes the quality of OM of good or bad. To me it would make more sense if the terms OM composition and/or OM properties are being used. Or even OM biodegradability. Since OM 'quality' is what the paper is about, this needs to be clarified.**

**A better definition may also prevent strange constructions like on P3L6-9: To access (better: obtain?) information on quality in terms of biodegradability of freeze locked OM, we examined characteristic OM parameters (amount and quality) and LMWOAs. In other words: you assess quality in order to examine quality....**

*We used the "OM quality" term since it was used in the literature before. However, following the suggestions of the reviewers we removed the term throughout the whole manuscript and directly defined in the text in which context the OM is assessed.*

**- Since biomarker concentrations and TOC follow the same trends, both reviewer 1 and 3 suggest to normalize biomarker data on TOC. Doing this would help make the case that downcore trends in biomarker concentrations are real, and not related to changes in TOC. In their rebuttal, the authors indicate that biomarker trends remain after normalizing. To prevent future readers asking the same question, I strongly suggest to present TOC normalized data in the next version.**

*We now added S-Fig. 2 with biomarker data related to gTOC to the supplements. The data show comparable trends with exception of two samples with very low TOC contents (core L14-05 and*

*coreL14-04). Additionally, a second table with the GDGT data related to gTOC is added to the supplements.*

**- source of OM (terrestrial vs lacustrine)**

**As already pointed out by reviewers 1 and 3 there are still contradictions in the description of the source of OM in the permafrost deposits. The origin is claimed to be terrestrial, but site descriptions clearly mention that the depositional environment was lacustrine. Also, all aliphatic-rich sediments are attributed an algal source. This needs to be discussed or stated more carefully.**

*The Holocene and Eemian samples were deposited in a thermokarst lake environment. However, for this OM the pyrolysis data indicate a clear terrestrial (Typ III) OM. It is the Yedoma OM which shows a tendency towards Typ II OM. Thus, overall the samples contain mainly terrestrial organic matter but most of the Yedoma samples show an increased proportion of aliphatic structural moieties. This causes most likely this shift into the direction to a more Typ II OM. We now stated this more carefully in the manuscript. Lines 314-324*

**- past vs present biomass (reviewer 1 comments 8 and 11)**

**I get that you choose to use PLFAs as a marker for living biomass, but I still don't see how that automatically makes GDGTs markers for past biomass. If you want to use GDGTs as such, you will have to provide a reference, or present this as a finding of this study rather than introducing them this way. Be careful with the ether-ester bond interpretation for the qualification of fossil/living-derived lipids, as this concerns headgroups, and not bonds within membrane lipids.**

**The extraction technique used here only extracts core lipid GDGTs. The vast majority of the GDGT pool in soils (also modern ones) is present as core lipid, so their presence as such does not tell you anything about the vitality of the microbial community, in past nor present. Instead you would have to look at IPL-GDGTs, i.e. with a phospho- or glyco-headgroup attached, and then I do agree that the phospho-GDGTs (attached with an ester bond) would indeed be a better marker for living organisms than glyco-GDGTs (attached with an ether bond).**

*– I am not sure whether I really understand what you are claiming here. Intact phospholipid esters (with fatty acid side chains) are used as life markers, since they rapidly degrade after cell death (White et al., 1979 and Logemann et al., 2011). Especially due to hydrolysis of the ester linked fatty acid side chains. In contrast intact phospholipid or glycolipid ethers (diethers or tetraethers) are more stable and degrade not along the ether bond side chain but finally at the head group. Logemann et al., 2011 indicated that it seems to make no difference, whether we have a phosphoester bound head group or a glycosidic bound head group. Both seem to have the same degradation rates (Logemann et al., 2011). Finally, the head group cleavage leads then to the accumulation of diether (archaeol) or tetraether (GDGTs) core lipids in the sediments after degradation. Thus, that bacterial phospholipids with ester linked side chains are less stable cannot be a matter of head group stability but of side chain stability. Since intact ether lipids are more stable, they only have restricted potential to act as life markers (Logemann et al., 2011) and thus, we did not use or measure these components. Instead we used the PLFAs as a general indicator for present microbial life here. At least this is my state of*

*knowledge. If there are new publications stating the opposite, I would be grateful if you could provide me these references.*

*We interpret br-GDGTs and iso-GDGTs as well as archaeol as degradation products of intact membrane lipids (loss of head groups). These biomarkers are widely used as past biomarkers for paleo-climatic and -environmental interpretation also in permafrost environments (Peterse et al., 2011; Schouten et al. 2013; Weijers et al., 2006). However, it is clear that the loss of the head group makes them not automatically to past markers. They could also be the remnants of an actual living microbial community in the deeper parts of the permafrost successions. That is the reason why we compare the GDGT and archaeol data sets with the PLFA depth profiles (see lines 367-369). The fact that abundant microbial life in permafrost regions is restricted to the surface active layer and that the depth profiles are different makes us confident, that these ether biomarkers represent the past microbial community. Line 370-374 and 385-396*

**- Comment 10 of reviewer 1: Space wise I can see why you choose to plot isoGDGT-0 and archaeol in the same panel. However, since they do not necessarily share the same source, please plot them as separate lines, so it is clear when they follow the same trend and when they do not.**

*– We separated iso-GDGT-0 and archaeol now in different plots in Fig .2.*

**- Introduction:**

**The transition to the paragraph with the study area (P2L30) is still very sudden. From the previous section it is not clear what the open question actually is that will be addressed in this study. This only starts to make sense after reading the actual aim in the very last paragraph on P3L26. I suggest to move this paragraph up, prior to the site introduction. The permafrost feedback that is the whole driver of this study can also be better introduced, as it is now only mentioned on the side (P2L13).**

*– We revised and restructured the introduction chapter. The aim is pointed out in the abstract. Then we start a general introduction into the background of the current study leading again to the aim of the study at the end of the introduction. I guess this is the usual structure for the introduction part in a paper. We shifted the part on Bol´shoy Lyakhovsky Island to the "Study area and materials chapter" and we added a sentence on the carbon climate feedback cycle. Lines 46-47*

**- Results:**

**The results are presented as values, rather than what they really are (e.g. concentrations). For example, P7L8: TOC values instead of TOC concentration. Check this in the whole section.**

*– Changed to "concentrations or contents" in most cases. The Hydrogen index are usually used with the term "values".*

**- Statistics:** I am happy to see that the authors have included statistical support for their statements.

**Textual (a selection, the text needs editing):**

P1L16:…acetate are used. *– Abstract was revised.*

P1L16: replace highest by largest (pool). *– Changed. Line 13*

P1L17: …pools in permafrost are present in the layers that cover MIS 3 and MIS 4. *– Changed. Line 14*

P1L18: …deposits from MIS5e contain only a small pool of substrate. *– Changed as suggested. Line 15*

P2L8: replace formerly preserved OM by accumulated OM *– Changed to "accumulated and freeze-locked OM". Line 36*

P2L14: replace former by Previous *– Changed. Line 38*

P2L15: remove comma after in *– Revised. Line 39*

P2L31: 'The last interglacial deposits have been interpreted as Eemian' -> of course they are, otherwise they would not be from the last interglacial. *– Sentence part was removed due to restructuring.*

P4L24: replace which are distinct between by which occur between *– Changed. Line 121*

P4L28: how can a site drain and then freeze over? *– After lake drainage the sediments left behind froze over again/ becoming permafrost again. We added a reference. Line 125*

P5L16: indicate how samples were decalcified (with HCl?) *– Yes, information added. Line 144*

P6L4: indicate that free acetate was measured in the pore water *– Added. Line 162*

P8L7: curves do not correlate but have R2 of 0.8, p=0.02? Seems like a nice correlation to me though…*– Text revised. Lines 233-235*

P8L16: How can you have an average concentration in a range? *– Revised. Line 244*

P11L30: if you use significant provide p-value *– This part was rewritten. Lines 347-360*

P12L7: how can OM stimulate a microbial community? I think you need to rephrase. *– We write now: "stimulated a diverse bacterial and archaeal community". Line 363*

P12L12: archaeol without a headgroup is not an IPL! See earlier comment on IPL vs CLs and present vs past biomarkers. *– This was related to "ether bond moieties". We rephrased this part. Line 372*

**To the comments of reviewer 1:**

I find the manuscript improved, but also that some of my concerns have been only partly addressed (previous comments 1, 2, 4, 8):

**(1) I agree with the authors that chemical composition of organic matter can influence its degradability. However, we know that chemical composition is not the only factor involved. Since the data on chemical composition of organic matter presented here are used to make conclusions about its degradability, I think a comment on other parameters that can have an impact is necessary, and that the data need to be discussed more carefully. Apart from temperature, I am for instance thinking of oxic versus anoxic conditions and association with soil minerals.**

*Of course OM degradation is the result of different factors such as OM composition, environmental*

*conditions and microbial controls. However, the focus of this paper is clearly placed on the evaluation of specific compositional characteristics of the OM of different ages. Nevertheless, we discussed already the accumulation and deposition of permafrost OM within the context of the environmental conditions such as soil moisture including aerobic and anaerobic soil conditions and temperature in the paper (see discussion). Thus, this criticism is only partly justified. In the revised version we now broaden this discussion starting the Discussion chapter with a general introducing paragraph on the factors influencing OM accumulation and degradation (Lines 298-305). Furthermore, environmental and microbial controls are considered at several positions in the discussion.*

**(2) I still find the motivation for looking at past and present microbial biomarkers poorly justified. For instance, on page 12, lines 7-8, the authors write: "In order to investigate whether the freeze-locked OM already stimulated a microbial community during deposition in the past, biomarkers for past microbial communities were examined." Why would anybody assume otherwise? Microorganisms exist in almost every environment. There is no reason to assume that there were no microbial communities in the Pleistocene.**

*Yes, this was maybe a bit misleading. Our intention was to show whether specific kind of OM stimulated already an abundant microbial community for the production of greenhouse gases in the past. In addition to methanogenic archaea this would include also bacterial microorganisms needed to degrade the OM by fermentative processes to provide the substrate pool for methanogens. However, in the current version we now strengthened the focus on the biomarkers representing methanogenic archaea. We made this point clearer in the Abstract (Lines 18-20) and in Chapter 5.2 (Lines 363-367).*

**(4) Please add a list with the specific compounds considered for total PLFAs. There is a lot of variation between different studies.**

*We added a list of PLFAs found in the active layer and mention that the PLFA diversity decreases with depth (Lines 200-205).*

**(8) While my technical comments have been addressed, there are plenty of new grammatical errors and unclear sentences (see below). Please check your language and consult a professional service if necessary.**

*We significantly revised the manuscript concerning unclear sentence structures.*

**Further general comments:**

**(10) I find the discussion of soil moisture conditions during deposition very speculative (e.g., page 13, lines 13-28).**

*Generally, literature data show that the Yedoma successions have been deposited during slightly warmer and wetter conditions leading to higher soil moisture with anaerobic soil conditions. These conditions were favorable for the accumulation of presumably less degraded OM, which we tried to link to the OM characteristics on Bol´shoy Lyakhovsky Island (higher TOC, higher HI, higher aliphatic character, higher acetate/substrate concentrations). Using these relations we speculate a bit on the conditions for OM accumulation during the Holocene and especially during the Eemian. Nevertheless, we strongly revised and restructured the discussion and reduced the speculative parts.*

**Technical corrections**
Page 2, line 7: "resulted in". *– Changed. Line 35*
Page 2, line 9: Delete "again". *– Removed and sentence revise. Line 37*
Page 2, line 12: "enhanced". I'm also not sure what you mean with "microbial production". Activity?

Growth? – We removed "the" to change sentence to: "microbial production and release of greenhouse gases"

Page 2, lines 13-14: I suggest a separate sentence for the global change feedback. - Changed to "This enhanced release is expected to have strong feedback on global warming and further permafrost degradation." Line 44

Page 2, line 15: Delete "in". – Removed. Line 39

Page 2, line 18: "at today's coasts and islands". You could also delete the insert. – Removed. Line 48

Page 2, line 19: Split the sentence. E.g., "These deposits provide a unique paleo-environmental archive … ". – Changed as suggested. Line 49

Page 2, line 21: What do you mean with "here"? – Removed. Line 51

Page 2, lines 30-31: This sentence is incomprehensible. I suggest: "We selected Bol'shoy Lyakhovsky Island in the Laptev Sea (NE Siberia) as our study area, since it provides an excellent opportunity to investigate permafrost OM deposited from the last interglacial to the Holocene."– Revised and shifted to Study area. Lines 93-119

Page 2, lines 33-34: I suggest: "These Eemian depostis form a paleo-equivalent to the Holocene and are otherwise rather difficult to assess." – Changed as suggested. Lines 103-104

Page 3, line 2: "Oyogos Yar". – Corrected. Line 104

Page 4, lines 26-28: Please add a reference. – Reference added. Line 125

Page 5, line 28: I think the reference should be in brackets. – Changed. Line 156

Page 5, line 29: With C1-C5 gases, do you mean alkanes? – This is the usual assignment, but we added now "alkane gases". Line 157

Page 6, line 4: What other anions? This is the only time this is mentioned. If other compounds were quantified originally but are not presented in this paper, I would not mention them. – Removed! Line 161

Page 6, line 23: Change "were" to "was". – Changed. Line 180

Page 6, line 25: What do you mean with "optical correlations"? – Changed. Line 184

Page 6, line 29: Add "communities" after "bacterial". – Changed to "past bacterial and archaeal communities". Lines 188-189

Page 6, line 30: The bracket after "archaeal communities" is incomprehensible. – Changed to (isoGDGT-0 and archaeol). Line 189

Page 7, line 22: Add a "the" to the "TOC curve". – Added. Line 216

Page 8, line 7: How is R2=0.8, p=0.021 not a significant correlation? – Changed. Line 234

Page 9, line 16: Remove "of". – *Removed. Line 278*

Page 10, line 6: Remove "again". – *We cannot see what is wrong with "again", because it was already bioavailable in the past when it was part of the active layer. Line 298*

Page 10, line 9: Remove "rich" in both cases. – This part was removed due to the streamlining process.

Page 10, lines 8-14: See my general comments on organic matter chemistry. I do not question that chemical composition is important, but I think a more balanced view is necessary. Further, I don't find it justified to claim that pyrolysis is established here as a new tool. It has been frequently used before and the ending of the sentence ("… as introduced in Stapel et al….") is in fact a contradiction to that claim. The sentence in lines 10-12 is also not clear.

*– We revised this introductory paragraph of the discussion chapter presenting now a more balanced view on OM accumulation and degradation. However, the focus of this paper still are the different characteristics of the OM of different ages. Lines 298-305*

Page 10, lines 20-21: This interpretation implies that there is strong algal growth now in the active layer. Is there any evidence for this from previous studies that use different approaches?

*– Soil algae material can be one explanation. We revised this part to make this clearer and added a reference. Lines 322-324*

Page 10, line 30: Nitrogen is not respired, please re-phrase. Page 10, lines 28-31: A common mechanism that can lead to low TOC/TN values is soils is the accumulation of inorganic N (binding of

ammonium to clay minerals).

*– Due to streamlining the Discussion this part was completely removed.*

Page 11, line 6: Dryer. *– Changed. Line 338*
Page 11, line 17: Change "into" to "in". *– Changed. Line 327*
Page 11, line 27: Change "setting" to "settings". *– Part was rewritten. Lines 347-360*
Page 11, line 28: "Reveals". *– Changed.*
Page 12, line 3: What do you mean with "higher rates of decomposed OM"? *– The whole paragraph was removed in order to avoid repetitions and to streamline the manuscript.*
Page 12, line 14: "contain bacteria and archaea". *– Changed to "consist of both". Line*
Page 12, lines 18-19: Do you mean labile carbon in the permafrost? *– Changed to "newly produced and old". Line 379*
Page 13, line 1: "varies between". *– Changed. Line 393*
Page 13, line 3: "markers". *– Changed. Line 395*
Page 13, lines 6-7: There is some mixup in the text. *– This paragraph was rewritten. Lines 401-402*
Page 13, line 9: I would change "greenhouse gas production" to "methane production". *– Changed as suggested. Line 401*
Page 13, lines 9-11: What do you mean? I thought the whole paragraph is about correlations between TOC and past microbial markers. *– This sentence was introduced because the reviewers ask for relating biomarker data to gTOC. We added know a table and a figure into the supplements with biomarker data related to gTOC.  Lines 403-404*
Page 13, lines 13-14: Please add a "might" or "is expected to" or similar. *- Changed as suggested. Line 405*
Page 13, line 30: Please clarify which periods the thermokarst lakes refer to. *– Holocene soil OM is suggested to be the largest source of brGDGTs supplied to the thermokarst lakes in the Kolyma region (Peterse et al., 2014). Also this paragraph was streamlined and revised. Line 351 and line 422.*
Page 14, line 3: Change to "… did not significantly affect the concentration …" *– This part was removed.*

Page 14, lines 6-7: Please re-phrase. This sentence is incomprehensible. *– Sentence was completely removed.*

Page 14, line 26: I don't find anything on the input of old and new carbon and a stimulation of microbial activity in the cited paper. *– I cannot reconstruct how this could happen. Maybe it is an artefact of a former revision of the text, where the text does not fit anymore to the reference. We removed this reference here.*

Page 14, line 27: "result in". *– This part was rephrased. Lines 443-446*

Page 14, line 30 to page 15, line 6: Do you mean that when the active layer is deepened and reaches the now frozen MIS 3, 4, and 1 deposits, we expect a microbial biomass similar to the current active layer, and microbial consumption of the contained acetate? Can you write this more clearly? *– No, the active layer of the core containing MIS 3 deposits (core L14-02) contains MIS 3 OM which is thawed already. The fact that this material shows the highest PLFA concentrations indicate that the MIS 3 OM can act as a good substrate provider for microbial life. We revised this part. Lines 448-458*
Page 15, line 8: Article missing ("… that the free-acetate pool …") *– Changed. Line 460*
Page 15, line 17: I would replace "caused by the onset of the early Holocene" by "at the onset of the early Holocene". I suppose this is what you mean. *– Changed to "starting with". Line 469*
Page 16, lines 7-8: Please specific what you mean with "concerning" here. *– This paragraph was removed during the streamlining process.*
Page 3, lines 15-16: I don't understand, please rephrase. *– Rephrased. Lines 72-74*

---

## Author Response (AR3)

**bg-2017-89**

**Title: Substrate potential of last interglacial to Holocene permafrost organic matter for future microbial greenhouse gas production**

**Authors:** Janina G. Stapel, Georg Schwamborn, Lutz Schirrmeister, Brian Horsfield, Kai Mangelsdorf

**Comments to Reviewer 1:**

The structure of the introduction has improved and is much clearer now. Also many of the grammatical errors have been corrected, although a few new ones have emerged in the revised parts of the manuscript. I still have a few comments that should be addressed prior to publication in BGS:

P1, L12: …,which form…. – Changed as suggested. Line 12.

P1, L17: …pyrolysis experiments on the… – Changed as suggested. Line 17.

P2, L49: These deposits provide…. – Changed as suggested. Line 49.

P3, L73: replace 'applied' by 'performed'. Or even better, adjust the whole sentence to: In addition, microbial biomarkers such as….are analyzed to examine… – Second option is considered. Line 73.

P4L107: add the IR-OSL age. – Age data added. Line 108.

P4:125: This line confused me earlier. I don't think soils freeze over. Lakes do, but soils/sediments just freeze. Delete 'over' (but check with a native speaker if you can). – "over " is removed. Line 126.

L201-205: Can you add how PLFAs were identified? In particular the position of branches/unsaturations/presence of rings (rather than unsaturation). – PLFAs were identified by means of mass spectrum interpretation and comparison with a standard mix comprising saturated, unsaturated, branched and cyclo-propyl-fatty acid methyl esters. Lines 173-175.

L205: 'The sum of all brGDGTs (…) varies between xx and xx, while concentrations of the….' – Changed as suggested. Line 208.

L234: …but reach their maximum at a different depth. – Changed as suggested. Lines 237-238.

L298-305: This paragraph should go to the introduction. – Elements of this paragraph are already part of the introduction. Thus, we would like to keep this paragraph at this position as a starter for the following discussion. Lines 302-309.

L312: add the TOC concentration between brackets. I have no idea what a 'typical concentration' is for a Yedoma deposit… – It was simply meant, that the TOC values are in the same range as observed in other studies investigating Yedoma deposits. This sentence was rewritten and we added "up to 4.9% TOC". Lines 315-317.

L316: Since you do not present any results that directly support the thermokarst environment during the Holocene and Eemian, add a reference after the statement that Holocene and Eemian sediments are deposited in a thermokarst environment. This seems quite important given your sub-conclusion in the last paragraph of this section. – We added a reference to this sentence. Line 320.

L316: The BIT index is not properly introduced, nor are BIT index values presented in the results section. At least mention brGDGTs as soil end-member and crenarchaeol (currently not introduced at all) as marine end-member, being produced by marine algae. Consequently, soils are characterized by high BIT index values, and marine sediments by low BIT index values.

As for your specific setting: BIT index values in lake sediments are not well understood (see e.g. Blaga et al 2009 J. Paleolimn. or Blaga et al 2011 GCA), and can easily reach values op to 1 as well, as the BIT index can be influenced by Thaumarchaea and (unknown) aquatic brGDGT-producers (e.g. Tierney et al., 2010 GCA, Weber et al., 2015 GCA). Hence, the BIT index is not a very reliable tool to differentiate between soils and thermokarst/lake material. The only statement I would be comfortable making is that BIT index values in your permafrost material are high, and thus have a terrestrial origin (as opposed to marine). – The BIT Index was only used to underline the terrestrial character or origin of the OM. However, the pyrolysis data also can stand alone. Thus, the BIT data are not really important in this context and the reviewer is right that they do not bring additional information into the story. Thus, we removed these data from the manuscript, but keep them in the supplement table for readers which might be interested in those data. Removed from Lines 320 ff.

L320ff: Try to avoid using samples, but instead mention the type of sediment. In this case active layer material. This makes it easier to follow what deposit is being discussed. – We do not agree here, since samples are defined as "TOC-rich samples from active layer and LP (MIS 3 and 4)" in line 321 and then we refer to this definition by writing "these samples". Every time the samples set is changed samples are clearly described/defined again for instance "LP glacial period sample with lower TOC (line 334)" or "Eemian samples" (line 337). In our opinion adding always sample descriptions would create many repetitions and make the text only longer and less simple to read.

L338: replace dryer by drier. – Changed as suggested. Line 340.

L349: Peterse et al 2014?? – Changed as suggested. Line 352.

L365: replace 'as to' by 'on'. – Changed as suggested. Line 367.

L380: decreased compared to what? Other layers? Or over time? – We added "compared to the active layer. Line 382.

I don't understand the next sentence. Life marker signals represent living community… – It is meant that the permafrost preserves the past microbial community from time of deposition. We added "in the permafrost section" in line 383" and "Thus, the permafrost preserved the microbial community of the past." in line 384-385.

L410: 'warmer conditions' or 'higher temperatures'. – We removed "temperature". Line 413.

L413: Stapel et al 2017 is not listed in the references. – Changed to 2016. Line 416.

L417: …suggest lower soil-moisture content in the active… – Changed as suggested. Line 420.

L418: An explanation for the lower soil-moisture content is that all cores… – Changed as suggested. Line 421.

L424-425: this is exactly the reason why I asked earlier to normalize biomarker data to TOC, as this makes a fairer comparison than comparing g biomarker/g sediment. It would make sense to use the TOC-normalized brGDGT concentrations here too. – We added br-GDGT/gTOC data. Lines 426-428.

Also, these last few lines seem a bit out of place here, and should better fit with the discussion on the sources of OC in the different permafrost layers. – These lines are related to the microbial biomarker abundances in our and the Kolyma study by Peterse et al. (2014), thus we would like to keep them at the current position in the chapter where we discuss the microbial biomarker signals.

L470: As a consequence, or consequently. – Changed as suggested. Line 474.

L482: Thus, in contrast to simply using the… – Changed as suggested. Line 486.

L490: these – Changed as suggested. Line 494.

**Comments to Reviewer 2**

*General comments*

I find the manuscript much improved. Except for a few typos, I have no further objections.

*Technical corrections*

Line 43: "can lead to enhanced microbial production" – Changed as suggested! Line 43.

Line 49: "these deposits provide" – Changed as suggested! Line 49.

Line 55: "thermo-erosional" (I also had to fight with auto-correct here.) – Changed as suggested! Line 55.

Line 99: "thermo-erosion" – Changed as suggested! Line 99.

Line 331: Replace "its" with "their" – Changed as suggested! Line 333.

Line 490: "These deposits" – Changed as suggested! Line 494.

Additionally we revised Tables 1 and 2. In Table 1 the column order was changed starting now from left to right with the cores followed by the age assignment. In Table 2 the column "OM quality" was removed as an artefact of a previous version. Also the table captions were revised.

Table and Figure assignment in the text was checked and revised.

Supplement Fig. S1 was revised now including also PLFAs in µg/gTOC. Supplement Tables were revised.

---

## Author Response (AR4)

**Associate Editor Decision: Publish subject to technical corrections** (06 Feb 2018) by Jack Middelburg

Comments to the Author:

I have read your paper in detail and I am sorry to tell you that I am a little disappointed about the presentation quality since this is version 4. The science is fine but the writing needs improvement. I nevertheless have decided to accept your paper pending further technical corrections. Congratulations, but please make these corrections and try to improve your writing further. After that and editorial office corrections your paper will likely make a nice contribution to the literature.

We revised the manuscript following your suggestions below. Furthermore, we revised the writing of the whole text and have the final manuscript checked by a native speaker. The text was changed to American English.

Technical corrections:

• All through please check your use of past vs. present tense. For instance, in the abstract many sentences should have been written in past tense. – We checked and revised the tense throughout the manuscript. We used past tense for the abstract. In the Introduction background facts are presented in present tense. Past was used when referring to past events. In the Methods chapter we used the past tense to describe what was done. In the result chapter we also used past tense for the results obtained. In the discussion chapter past tense is applied to summarize results combined with present tense to explain the significance of results or to interpret results. In the Conclusion we used present tense. Text was changed to American English.

• The use of marine isotope stages and Holocene, Eemian is somewhat confusing (some section use MIS, others Eemian, different authors?) and please introduce MIS before first use, both in abstract and in main text. – In the results chapter we used only MIS, because this classification of the sample material was easier for the description of the results. We now allocated also the respective time period names, used in the introduction and discussion, to the different marine isotope stages for better orientation (headers of chapter 4, Lines 211-276). Additionally, in the Introduction and Discussion, period names are linked again to the marine isotope stages (Line 39 and Line 319). MIS was introduced now in abstract (Line 14) and main text (line 39), when mentioned for the first time.

• Line 62: are suggested to (by who??). – Changed to "might have strong effect". Line 64.

• Line 70: On the one and on the other hand always go together – Changed as suggested. Line 70.

• Line 73: were analysed/analysed – Changed as suggested. Line 75.

• Line 141: deposits were deposited: reformulate – Changed to "Sediments … were deposited". Line 141.

• Line 145: grinding, samples – Changed as suggested. Line 145.

• Line 162: rpm have no meaning, use g-force – Converted with a radius of 130 mm and 2500 rpm. Line 162.

• Line 163: were measured in pore-water samples – Changed as suggested. Line 163.

• All through, use construction like from x to y and vary between x and y. Do not use a mixture.

(there are many of these misuses) – We revised this throughout the manuscript.

• Line 208: vary between 849.1 and 27.2… – Comma was removed. Line 206.

• Line 238: all past markers were high (results so in past tense and use high rather than strongly increased, because increased from what). – Changed to: "In the MIS 3-1 section all past markers were significantly higher … compared to the other MIS 3 deposits. Line 237-238.

• Line 243: reformulate: are indicated – Revised to "The sample from MIS 3-1 showed a free… Line 242.

• Line 344: .. 3) for the Eemian deposits (…) showed a more…. – Changed as suggested. Line 343.

• Line 367: on how the – Changed as suggested. Line 366.

• Line 488: OM with high HI.. – Changed as suggested. Line 484.

• Concentration are reported in mg/L for acetate. Is this acetate or mg C/L. I guess the former as most organic geochemist report, but many biogeochemist report on C basis. Inform the reader. – We added "and measured as mg/l acetate." in Line 163.